# Dual RL: Unification and New Methods for Reinforcement and Imitation Learning

**Harshit Sikchi[1], Qinqing Zheng [2], Amy Zhang[1,2], Scott Niekum[3]**
[1] The University of Texas at Austin, [2] FAIR, Meta AI
[3] University of Massachusetts Amherst
`{hsikchi}@utexas.edu`

## Abstract

The goal of reinforcement learning (RL) is to find a policy that maximizes the expected cumulative return. It has been shown that this objective can be represented as an optimization problem of state-action visitation distribution under linear constraints. The dual problem of this formulation, which we refer to as *dual RL*, is unconstrained and easier to optimize. In this work, we first cast several state-of-the-art offline RL and offline imitation learning (IL) algorithms as instances of dual RL approaches with shared structures. Such unification allows us to identify the root cause of the shortcomings of prior methods. For offline IL, our analysis shows that prior methods are based on a restrictive coverage assumption that greatly limits their performance in practice. To fix this limitation, we propose a new discriminator-free method ReCOIL that learns to imitate from arbitrary off-policy data to obtain near-expert performance. For offline RL, our analysis frames a recent offline RL method XQL in the dual framework, and we further propose a new method $f$-DVL that provides alternative choices to the Gumbel regression loss that fixes the known training instability issue of XQL. The performance improvements by both of our proposed methods, ReCOIL and $f$-DVL, in IL and RL are validated on an extensive suite of simulated robot locomotion and manipulation tasks. **Project page (code and videos)**: hari-sikchi.github.io/dual-rl/

## 1 Introduction

A number of deep Reinforcement Learning (RL) algorithms optimize a regularized policy learning objective using approximate dynamic programming (ADP) [Bertsekas and Tsitsiklis, 1995]. Popular off-policy temporal difference algorithms spanning both imitation learning [Kostrikov et al., 2018, Ni et al., 2021] and RL [Haarnoja et al., 2018, Janner et al., 2019, Sikchi et al., 2022b, Hafner et al., 2023] exemplify this class. As we discuss in Section 3, one way to develop a principled off-policy algorithm is to ensure unbiased estimation of the on-policy policy gradient using off-policy data [Nachum and Dai, 2020]. Unfortunately, many classical off-policy algorithms do not guarantee this property, resulting in issues like training instability and over-estimation of the value function [Fu et al., 2019, Fujimoto et al., 2018, Baird, 1995]. To obtain high learning performance, these algorithms require that most data be nearly on-policy, otherwise require special algorithmic treatments (e.g., importance sampling [Precup et al., 2001], layer normalization [Ball et al., 2023], prioritized sampling [Vecerik et al., 2017]) to avoid the aforementioned issues. Recently, there have been developments leading to new off-policy algorithms with improved performance for RL [Kumar et al., 2020, Garg et al., 2021, Kostrikov et al., 2021] and IL [Zhu et al., 2020, Ma et al., 2022, Garg et al., 2021, Florence et al., 2022]. These methods are derived via a variety of mathematical tools and attribute their success to different aspects. It remains an open question if we can inspect these algorithms under a unified framework to understand their limitations, and subsequently propose better methods.

In this work, we consider a specific formulation for RL that writes the performance of a policy as a convex program with linear constraints [Manne, 1960]. This convex program can be converted into unconstrained forms using Lagrangian duality, which is more amenable for numerical optimization. We refer to the class of approaches that admit the dual formulations as *Dual RL*. Dual RL approaches naturally provide unbiased estimation of the on-policy policy gradient using off-policy data, in a principled way. They avoid explicit importance sampling that leads to high variance and ensures training stability and convergence [Tsitsiklis and Van Roy, 1996]. Related approaches in this space have often been referred to as DICE (DIstribution Correction Estimation) methods in previous literature [Nachum et al., 2019, Kostrikov et al., 2019, Lee et al., 2021, Ma et al., 2022, Zhang et al.,

| | Method | dual-Q/V | Gradient | Objective | Off-Policy Data |
|---|---|---|---|---|---|
| RL | AlgaeDICE, GenDICE, *CQL* | $Q$ | semi | regularized RL | Arbitrary |
| | OptiDICE | $V$ | full | regularized RL | Arbitrary |
| | *XQL*, REPS, $f$-**DVL** | $V$ | semi | regularized RL | Arbitrary |
| | VIP, GoFAR | $V$ | full | regularized RL | Arbitrary |
| | Logistic Q-learning | $QV^1$ | full | regularized RL | ✗ |
| IL | *IQLearn, IBC* | $Q$ | semi | $D_f(\rho^\pi \| \rho^E)$ | Expert-only |
| | *OPOLO, OPIRL* | $Q$ | semi | $D_{kl}(\rho^\pi \| \rho^E)$ | Arbitrary |
| | SMODICE | $V$ | full | $D_{kl}(\rho^\pi \| \rho^E)$ | Arbitrary |
| | DemoDICE, LobsDICE | $V$ | full | $D_{kl}(\rho^\pi \| \rho^E) + \alpha D_{kl}(\rho^\pi \| \rho^R)$ | Arbitrary |
| | P$^2$IL | $QV^1$ | full | $D_C(\rho^\pi \| \rho^E)^1$ | ✗ |
| | **ReCOIL-Q** | $Q$ | full | $D_f(\rho^\pi_{mix} \| \rho^{E,R}_{mix})$ | Arbitrary |
| | **ReCOIL-V** | $V$ | full | $D_f(\rho^\pi_{mix} \| \rho^{E,R}_{mix})$ | Arbitrary |

Table 1: A number of recent works can be studied together under the unified umbrella of **dual-RL**. These methods are instantiations of dual-RL with a choice of update strategy, objective, constraints, and their ability to handle off-policy data. **Bold** names correspond to the methods proposed in the paper and *Italic* names correspond to methods that aren't yet known to be dual-approaches.

2020]. We note that the linear programming formulation for the RL objective has been used and studied in Manne [1960], Denardo [1970], de Ghellinck and Eppen [1967], Borkar [1988], Malek et al. [2014] and the general duality framework for regularized RL was first introduced in Nachum and Dai [2020].

Our *first* contribution is to extend the work of Nachum and Dai [2020] and show that many recent algorithms in deep RL and IL can all be viewed as different instantiations of dual problems for regularized policy optimization, see Table 1 for the complete list. These algorithms have been motivated from a variety of perspectives and differing derivations. For example, XQL [Garg et al., 2023] focuses on introducing Gumbel regression into RL, CQL [Kumar et al., 2020] aims at learning a pessimistic $Q$ function, IQLearn [Garg et al., 2021] and OPOLO [Zhu et al., 2020] use the change of variables for IL, and IBC [Florence et al., 2022] uses a contrastive loss for imitation learning, but as we show all can actually be derived from the dual formulation.

*Second*, the dual unification in IL reveals an important shortcoming of prior methods that learn to imitate the expert by leveraging arbitrary off-policy data. Prior work [Ma et al., 2022, Zhu et al., 2020, Kim et al., 2022b] imposes a coverage assumption (the suboptimal data covers the visitations of the expert data) and learn a density ratio between suboptimal and expert data via a discriminator to use it for downstream learning. In an offline setting, with limited data and coverage, learning a density ratio between suboptimal and expert can be challenging, and the inaccuracies of the discriminator can compound in downstream RL, negatively affecting resulting policy performance. We show that by a simple modification to the dual formulation, we can get away from this limitation. In Section 5, we present ReCOIL, a simple, theoretically principled, and discriminator-free imitation learning method from arbitrary off-policy data. We empirically demonstrate the failure of previous IL methods based on the coverage assumption in a number of MuJoCo environments and show substantial performance improvements of ReCOIL in Section 7.

*Third*, the presented unification also provides us with a useful tool to examine the limitation for a recent offline RL method, XQL [Garg et al., 2023]. XQL's success was originally attributed to better modeling of Bellman errors using Gumbel regression. On the other hand, XQL also suffers from training instability, also caused by Gumbel regression. By situating the implicit policy improvement algorithms like XQL in the dual RL framework, in Section 6 we are able to propose a family of implicit algorithms $f$-Dual V Learning ($f$-DVL), which successfully addresses the training instability issue. The empirical experiments on the D4RL benchmarks establish the superior performance of $f$-DVL, see Section 7.

## 2  RELATED WORK

**Off-Policy Methods for IL**  Imitation learning has benefited greatly from using off-policy data to improve learning performance [Kostrikov et al., 2018, Agarwal et al., 2020, Zhu et al., 2020, Ni et al., 2021, Sikchi et al., 2022a]. Often, replacing the on-policy expectation common in most Inverse RL formulations [Ziebart et al., 2008, Swamy et al., 2021] by expectation under off-policy samples, which is unprincipled, has led to gains in sample efficiency [Kostrikov et al., 2018]. Previous works have proposed a solution in the dual RL space for principled off-policy imitation but is based

---

[1]These methods use a different regularizer. More details in Appendix C.4.

on a restrictive coverage assumption [Ma et al., 2022, Zhu et al., 2020, Kim et al., 2022b] which requires estimating a density ratio using a discriminator and further limit themselves to matching a particular $f$-divergence. In this work, we eliminate this assumption and allow for generalization to all $f$-divergences, presenting a principled off-policy discriminator-free approach to imitation.

**Off-Policy Methods for RL** Off-policy RL methods promise a way to utilize data collected by arbitrary behavior policies to aid in learning an optimal policy and thus are advantageous over on-policy methods. This promise falls short, as previous off-policy algorithms are plagued with a number of issues such as overestimation of the value function, training instability, and various biases [Thrun and Schwartz, 1993, Fu et al., 2019, Fujimoto et al., 2018, Kumar et al., 2019]. A common cause for a number of these issues is *distribution mismatch*. As we shall discuss later, the RL objective requires on-policy samples but is often estimated by off-policy samples in practice. Prior works have proposed fixing the distribution mismatch by using importance weights [Precup, 2000], which can lead to high variance policy gradients or ignoring the distribution mismatch completely [Haarnoja et al., 2018, Fujimoto et al., 2018, Sun et al., 2022, Ji et al., 2023, Chen et al., 2021b]. Unfortunately, these approaches do not carry over well to the offline setting. For example, when deploying the policy online, the overestimation bias can be corrected by the environment feedback, which is infeasible for offline RL. A number of solutions exist for controlling overestimation in prior work—$f$-divergence regularization to the training distribution [Wu et al., 2019, Fujimoto et al., 2019], support regularization [Singh et al., 2022], implicit maximization [Kostrikov et al., 2021], Gumbel regression [Garg et al., 2023] and learning a Q function that penalizes OOD actions [Kumar et al., 2020]. Dual-RL methods consider a regularized RL setting suitable for offline RL as well as fixing the distribution mismatch issue in a principled way.

## 3 PRELIMINARIES

We consider an infinite horizon discounted Markov Decision Process denoted by the tuple $\mathcal{M} = (\mathcal{S}, \mathcal{A}, p, r, \gamma, d_0)$, where $\mathcal{S}$ is the state space, $\mathcal{A}$ is the action space, $p$ is the transition probability function, $r : \mathcal{S} \times \mathcal{A} \to \mathbb{R}$ is the reward function, $\gamma \in (0, 1)$ is the discount factor, and $d_0$ is the distribution of initial state $s_0$. Let $\Delta(\mathcal{A})$ denote the probability simplex supported on $\mathcal{A}$. The goal of RL is to find a policy $\pi : \mathcal{S} \to \Delta(\mathcal{A})$ that maximizes the expected return: $\mathbb{E}_\pi\left[\sum_{t=0}^\infty \gamma^t r(s_t, a_t)\right]$, where we use $\mathbb{E}_\pi$ to denote the expectation under the distribution induced by $a_t \sim \pi(\cdot|s_t), s_{t+1} \sim p(\cdot|s_t, a_t)$. We also define the discounted state-action visitation distribution $d^\pi(s, a) = (1 - \gamma)\pi(a|s)\sum_{t=0}^\infty \gamma^t P(s_t = s|\pi)$. The unique stationary policy that induces a visitation $d(s, a)$ is given by $\pi(a|s) = d(s, a)/\sum_a d(s, a)$. We will use $d^O$ and $d^E$ to denote the visitation distributions of the behavior policy of the offline dataset and the expert policy, respectively. $f$-divergences are denoted by $D_f$ and measure the distance between two distributions. For a more formal overview of the above concepts, refer to Appendix B.1.

**Value Functions and Bellman Operators** Let $V^\pi : \mathcal{S} \to \mathbb{R}$ be the state value function of $\pi$. $V^\pi(s)$ is the expected return when starting from $s$ and following $\pi$: $V^\pi(s) = \mathbb{E}_\pi\left[\sum_{t=0}^\infty \gamma^t r(s_t, a_t)|s_0 = s\right]$. Similarly, let $Q^\pi : \mathcal{S} \times \mathcal{A} \to \mathbb{R}$ be the state-action value function of $\pi$, such that $Q^\pi(s, a) = \mathbb{E}_\pi\left[\sum_{t=0}^\infty \gamma^t r(s_t, a_t)|s_0 = s, a_0 = a\right]$. Let $V^*$ and $Q^*$ denote the value functions corresponding to an optimal policy $\pi^*$. Let $\mathcal{T}_r^\pi$ be the Bellman operator with policy $\pi$ and reward function $r$ such that $\mathcal{T}_r^\pi Q(s, a) = r(s, a) + \gamma\mathbb{E}_{s'\sim p(\cdot|s,a), a'\sim\pi(\cdot|s')}[Q(s', a')]$. We also define the Bellman operator for the state value function $\mathcal{T}_r V(s, a) = r(s, a) + \gamma\mathbb{E}_{s'\sim p(\cdot|s,a)}[V(s')]$.

### 3.1 REINFORCEMENT LEARNING VIA LAGRANGIAN DUALITY

RL optimizes the expected return of a policy. We consider the linear programming formulation of the expected return [Manne, 1960], to which we can apply Lagrangian duality or Fenchel-Rockfeller duality to obtain corresponding constraint-free problems. We now review this framework, first introduced by Nachum and Dai [2020][1]. Consider the regularized policy learning problem

$$\max_\pi J(\pi) = \mathbb{E}_{d^\pi(s,a)}[r(s, a)] - \alpha D_f(d^\pi(s, a) \,||\, d^O(s, a)), \tag{1}$$

where $D_f(d^\pi(s, a) \,||\, d^O(s, a))$ is a conservatism regularizer that encourages the visitation distribution of $\pi$ to stay close to some distribution $d^O$, and $\alpha$ is a temperature parameter that balances the expected return and the conservatism. An interesting fact is that $J(\pi)$ can be rewritten as a convex problem that searches for a visitation distribution that satisfies the *Bellman-flow* constraints. We refer to this

---

[1]We use Lagrangian duality instead of Fenchel-Rockfeller duality for ease of exposition.

form as `primal-Q`:

`primal-Q` $\quad \max_\pi J(\pi) = \max_\pi \left[ \max_d \mathbb{E}_{d(s,a)}[r(s,a)] - \alpha D_f(d(s,a) \,||\, d^O(s,a)) \right.$

$$\text{s.t } d(s,a) = (1-\gamma)d_0(s).\pi(a|s) + \gamma \sum_{s',a'} d(s',a')p(s|s',a')\pi(a|s), \ \forall s \in \mathcal{S}, a \in \mathcal{A} \Big]. \tag{2}$$

We can convert this to an unconstrained problem with dual variables $Q(s,a)$ defined for all $s,a \in \mathcal{S} \times \mathcal{A}$ by applying Lagrangian duality and the convex conjugate, giving us the `dual-Q` formulation:

`dual-Q` $\quad \max_\pi \min_Q (1-\gamma)\mathbb{E}_{s\sim d_0, a\sim\pi(s)}[Q(s,a)] + \alpha\mathbb{E}_{(s,a)\sim d^O}[f^*([\mathcal{T}_r^\pi Q(s,a) - Q(s,a)]/\alpha)], \quad$ (3)

where $f^*$ is the convex conjugate of $f$. In fact, one can note that Problem (2) is overconstrained—the constraints already determine the unique solution $d^\pi$, rendering the inner maximization w.r.t $d$ unnecessary. Therefore, we can relax the constraints to obtain another problem with the same optimal solution $\pi^*$ and $d^*$, which we call `primal-V` below:

`primal-V` $\quad \max_{d\geq 0} \mathbb{E}_{d(s,a)}[r(s,a)] - \alpha D_f(d(s,a) \,||\, d^O(s,a))$

$$\text{s.t } \sum_{a\in\mathcal{A}} d(s,a) = (1-\gamma)d_0(s) + \gamma \sum_{(s',a')\in\mathcal{S}\times\mathcal{A}} d(s',a')p(s|s',a'), \ \forall s \in \mathcal{S}. \tag{4}$$

Similarly, we consider the Lagrangian dual of (4), with dual variables $V(s)$ defined for all $s \in S$:

`dual-V` $\quad \min_V (1-\gamma)\mathbb{E}_{s\sim d_0}[V(s)] + \alpha\mathbb{E}_{(s,a)\sim d^O}\left[f_p^*([\mathcal{T}V(s,a) - V(s)]/\alpha)\right],$ (5)

where $f_p^*$ is a variant of $f^*$ defined as $f_p^*(x) = \max(0, f'^{-1}(x))(x) - f(\max(0, f'^{-1}(x)))$. Such modification is to cope with the nonnegativity constraint $d(s,a) \geq 0$ in `primal-V`. Note that in both cases for `dual-Q` and `dual-V`, the optimal solution is the same as their primal formulations due to strong convexity. See Appendix B.1 for a detailed review, connections between Fenchel and Lagrangian duality, and discussion of computing $\pi^*$ from $V^*$ for the `dual-V` formulation.

*Remarks.* The dual formulations have a few appealing properties. (a) They allow us to transform constrained distribution-matching problems into unconstrained forms w.r.t previously logged data. (b) One can show that the gradient of `dual-Q` w.r.t $\pi$, when $Q$ is optimized for the inner problem, is the on-policy policy gradient computed by off-policy data [Nachum and Dai, 2020]. This property is key to relieving the instability or divergence issue in many off-policy learning algorithms [Thrun and Schwartz, 1993, Fu et al., 2019, Fujimoto et al., 2018].

## 4 A UNIFIED PERSPECTIVE ON RL AND IL THROUGH DUALITY

In this section, we discuss how a number of recent RL and IL algorithms can be cast as dual-RL methods. We restrict ourselves to demonstrating this equivalence on a subset of methods in *offline* RL and IL settings from Table 1 whose shortcomings we study via the unified viewpoint. In later sections, we present approaches for addressing these shortcomings in both RL and IL. For the interested reader, a complete discussion on Table 1, particularly how algorithms like implicit behavior cloning, CQL and OPOLO can be cast as dual-RL, can be found in Appendix C. We further discuss the extension of dual-RL formulation to the online setting in Appendix E. All proofs are deferred to the appendix.

### 4.1 DUAL FORMULATION FOR EXISTING IMITATION LEARNING ALGORITHMS

We first consider the standard imitation learning setup where the agent is given a set of expert demonstrations, i.e. state-action trajectories, and does not have access to environment reward. We consider two possible offline IL settings — 1) only expert demonstrations are available and 2) we additionally have access to suboptimal transitions from the environment. Intuitively, these suboptimal transitions should aid in better matching the expert behavior.

**a. Offline IL with Expert Data Only** Imitation learning, or occupancy matching [Ghasemipour et al., 2020] is a direct consequence of the regularized RL problem (Eq. 1) when the reward is set to be 0 uniformly across the state-action space and the regularization distribution and $d^O$ are set to be the expert visitation distribution $d^E$. The corresponding `dual-Q` under these conditions simplifies to:

`dual-Q` $\quad \max_\pi \min_Q (1-\gamma)\mathbb{E}_{d_0(s),\pi(a|s)}[Q(s,a)] + \alpha\mathbb{E}_{s,a\sim d^E}[f^*([\mathcal{T}_0^\pi Q(s,a) - Q(s,a)]/\alpha)].$ (6)

Interestingly, this reduction directly leads us to IQLearn [Garg et al., 2021], which was derived using a change of variables in the form of an inverse backup operator.

**Proposition 1.** *IQLearn [Garg et al., 2021] is an instance of* `dual-Q` *using the semi-gradient [2] update rule with a (soft) Bellman operator, where* $r(s,a) = 0 \ \forall s \in \mathcal{S}, a \in \mathcal{A}, d^O = d^E$.

---

[2] For an overview of semi-gradient vs full-gradient methods please refer to Appendix B.1.7.

**b. Offline IL with Additional Suboptimal Data** Unfortunately, the dual-RL formulations above offer no way to naturally incorporate additional suboptimal data $d^S$. To remedy this, prior methods have relied on careful selection of the $f$-divergence and a *coverage assumption* to craft an off-policy objective [Zhu et al., 2020, Hoshino et al., 2022, Ma et al., 2022, Kim et al., 2022b,a]. More precisely, under the *coverage assumption* that the suboptimal data visitation covers the expert visitation ($d^S > 0$ wherever $d^E > 0$) [Ma et al., 2022], and with the KL divergence, we obtain the following simplification for the imitation objective:

$$D_{\text{KL}}(d(s,a) \,||\, d^E(s,a)) = \mathbb{E}_{s,a \sim d(s,a)}\left[\log \frac{d(s,a)}{d^E(s,a)}\right] = \mathbb{E}_{s,a \sim d(s,a)}\left[\log \frac{d(s,a)}{d^S(s,a)} + \log \frac{d^S(s,a)}{d^E(s,a)}\right]$$

$$= \mathbb{E}_{s,a \sim d(s,a)}\left[\log \frac{d^S(s,a)}{d^E(s,a)}\right] + D_{\text{KL}}(d(s,a) \,||\, d^S(s,a)).$$

The final objective now resembles `primal-Q` when $r(s,a) = -\log \frac{d^S(s,a)}{d^E(s,a)}$ and correspondingly we obtain the following `dual-Q` problem using Eq. 3:

$$\texttt{dual-Q} \quad \max_{\pi(a|s)} \min_{Q(s,a)} (1-\gamma)\mathbb{E}_{\rho_0(s),\pi(a|s)}[Q(s,a)] + \mathbb{E}_{s,a \sim d^S}\left[f^*(\mathcal{T}^\pi_{r^{imit}}Q(s,a) - Q(s,a))\right], \quad (7)$$

where $\mathcal{T}_{r^{\text{imit}}}$ denote Bellman operator under the *pseudo-reward* function $r^{\text{imit}}(s,a) = -\log \frac{d^S(s,a)}{d^E(s,a)}$. This objective also allows us to cast IL method OPOLO [Zhu et al., 2020] in the `dual-Q` framework (see Appendix C.3). The pseudo-reward is a logarithmic density ratio learned using a discriminator and is later used for policy learning by optimizing Equation 7. Density ratio learning is difficult in a limited data regime as well as when the expert and suboptimal data share low coverage, and errors in learned discriminators can cascade for RL training and deteriorate the performance of output policy. We show how a simple modification to the imitation objective can allow us to relax the coverage assumption and propose a discriminator-free IL method that learns performant policies from arbitrary suboptimal data, see Section 5.

## 4.2 Dual Formulation for existing reinforcement learning algorithms

Now that we have seen how IL can be understood as a special case of the full regularized RL objective, we consider the full objective in Eq. (1). Regularized policy learning, in its various forms [Nachum et al., 2019, Wu et al., 2019], is a natural objective for offline RL algorithms, preventing the policy from incorrectly deviating out-of-distribution by regularizing against the offline data visitation. However, implicit policy improvement (XQL [Garg et al., 2023], IQL [Kostrikov et al., 2021]), one of the most successful classes of offline RL methods which uses in-distribution samples to pessimistically estimate the greedy improvement to the $Q$-function, has evaded connections to regularized policy optimization. Proposition 2 shows, perhaps surprisingly, that XQL can be cast as a dual of regularized policy learning, concretely as a `dual-V` problem.

**Proposition 2.** *XQL is an instance of* `dual-V` *under the semi-gradient update rule, where the $f$-divergence is the reverse Kullback-Liebler divergence, and $d^O$ is the offline visitation distribution.*

The success of XQL was attributed to the property that Gumbel distribution better models the Bellman errors [Garg et al., 2023]. Despite its decent performance, XQL is prone to training instability (see e.g., Figure 2), since the Gumbel loss is an exponential function that can produce large gradients during training. Situating XQL in the dual-RL framework allows us to propose a solution to the training instability problem, a new insight we discuss in Section 6.

**A consequence of unification in RL:** Offline RL can be broadly categorized in three approaches: 1) regularized policy learning, 2) pessimistic value learning e.g. CQL [Kumar et al., 2020], ATAC [Cheng et al., 2022] and 3) implicit policy improvement algorithms (e.g. XQL). The latter two frameworks have seemingly been exceptions to the regularized policy learning formulation (e.g. Eq. (1)). In Proposition 4, we show that with an appropriate choice of $f$-divergence, CQL and ATAC can be cast as a `dual-Q` problem. Overall, our results (Proposition 4 and Proposition 2) are the first, to our knowledge, to bring together the latter two approaches, pessimistic value learning and implicit policy improvement as dual approaches to regularized policy learning.

## 5 ReCOIL: Imitation Learning from Arbitrary Experience

As demonstrated in Section 4.1, previous off-policy IL methods often rely on the coverage assumption and train a discriminator between the demonstration and the offline data to obtain a pseudo-reward $r^{\text{imit}}$. We propose **RE**laxed **C**overage for **O**ff-policy **I**mitation **L**earning (ReCOIL), an off-policy IL algorithm that relaxes the coverage assumption and eliminates the need for the discriminator. To

achieve this, we consider an alternative way to leverage suboptimal data for imitation: matching two mixture distributions $d_{\text{mix}}^S := \beta d(s,a) + (1-\beta)d^S(s,a)$ and $d_{\text{mix}}^{E,S} := \beta d^E(s,a) + (1-\beta)d^S(s,a)$, where $\beta \in (0,1)$ is a fixed hyperparameter. We consider the following problem in `primal-Q` form:

$$\texttt{primal-Q} \qquad \max_{d(s,a)} -D_f(d_{\text{mix}}^S(s,a) \,||\, d_{\text{mix}}^{E,S}(s,a))$$

s.t $\forall s \in \mathcal{S}, a \in \mathcal{A}, \quad d(s,a) = (1-\gamma)d_0(s)\pi(a|s) + \gamma \sum_{(s',a') \in \mathcal{S} \times \mathcal{A}} d(s',a')p(s|s',a')\pi(a|s).$ (8)

This is a valid imitation learning formulation [Ghasemipour et al., 2020] since the global maximum of the objective is attained at $d = d^E$, irrespective of the suboptimal data distribution $d^S$. The primal formulation (Eq. 8) deters offline learning, as it requires sampling from $d$ to estimate the $f$-divergence. We thus consider its dual formulation that allows us to derive an off-policy objective that only requires samples from the offline data. We term this formulation and associated approach `ReCOIL`.

**Theorem 1.** *(ReCOIL objective) The* `dual-Q` *problem to the mixture distribution matching objective in Eq. 8 is given by:*

$$\max_\pi \min_Q \beta(1-\gamma)\mathbb{E}_{d_0,\pi}[Q(s,a)] + \mathbb{E}_{s,a \sim d_{\text{mix}}^{E,S}}[f^*(\mathcal{T}_0^\pi Q(s,a) - Q(s,a))] - (1-\beta)\mathbb{E}_{s,a \sim d^S}[\mathcal{T}_0^\pi Q(s,a) - Q(s,a)] \quad (9)$$

*and recovers the same optimal policy $\pi^*$ as Eq. 8 since strong duality holds from Slater's conditions.*

In other words, imitation learning can be solved by optimizing the unconstrained problem `ReCOIL` with arbitrary off-policy data, without the coverage assumption. Besides, as opposed to many previous algorithms, `ReCOIL` uses the Bellman operator $\mathcal{T}_0$ which does not need the pseudo-reward $r^{\text{imit}}$, making it discriminator-free. Although the pseudo-reward is not needed for training, `ReCOIL` allows for recovering the reward function using the learned $Q^*$, which corresponds to the intent of the expert. That is, $r(s,a) = Q^*(s,a) - \mathcal{T}_0^\pi(Q^*(s,a))$. Moreover, our method is generic to incorporate any $f$-divergence. We also present the `dual-V` form for `ReCOIL` in Appendix D but defer its investigation for future work.

**A Bellman Consistent Energy-Based Model (EBM) View for ReCOIL** Instantiating `ReCOIL` with $\chi^2$ Divergence, we present a simplified objective (complete derivation in Appendix D.3) to:

$$\max_\pi \min_Q \beta(\mathbb{E}_{d^S,\pi(a|s)}[Q(s,a)] - \mathbb{E}_{d^E(s,a)}[Q(s,a)]) + 0.25\underbrace{\mathbb{E}_{s,a \sim d_{\text{mix}}^{E,S}(s,a)}[(\gamma Q(s',\pi(s')) - Q(s,a))^2]}_{\text{Bellman consistency}}. \quad (10)$$

One can see that `ReCOIL` learns a score function $Q$ whose expected value is low over the suboptimal distribution but high over the expert distribution, while ensuring that $Q$ is Bellman consistent over the mixture. The Bellman consistency is crucial to propagate the information of how to recover when the policy makes a mistake. The $Q$ value can be interpreted as a score as it is not representative of any expected return, and thus `ReCOIL` is an energy-based model with Bellman consistency. Figure 6 in the appendix illustrates this intuition.

**Practical Algorithm** In Algorithm 1 we consider three parameterized functions $Q_\phi(s,a)$, $V_\theta(s)$ and $\pi_\psi$. Furthermore, we rely on Pearson $\chi^2$ divergence (Eq. 10) as it has been shown to lead to stable learning for the

---

**Algorithm 1:** ReCOIL (offline, $\chi^2$)

1: Initialize $Q_\phi$, $V_\theta$, and $\pi_\psi$, mixing ratio $\beta$, conservatism $\tau$, temperature $\alpha$
2: $\mathcal{D}^S = (s,a,s')$ be suboptimal dataset
3: $\mathcal{D}^{\mathcal{E}} = (s,a,s')$ be expert dataset.
4: **for** $t = 1..T$ iterations **do**
5: $\quad$ Train $Q_\phi$ using $\min_\phi \mathcal{L}(\phi)$:
6: $\quad$ Train $V_\theta$ using $\min_\theta \mathcal{J}(\theta)$
7: $\quad$ Update $\pi_\psi$ via $\max_\psi \mathcal{M}(\psi)$:
8: **end for**

---

imitation setting [Garg et al., 2021]. Our practical algorithm uses a *semi-gradient* update that results in minimizing the following loss for $Q_\phi$:

$$\mathcal{L}(\phi) = \beta(\mathbb{E}_{d^S,\pi(a|s)}[Q_\phi(s,a)] - \mathbb{E}_{d^E(s,a)}[Q_\phi(s,a)]) + 0.25\,\mathbb{E}_{s,a \sim d_{\text{mix}}^{E,S}(s,a)}[(\gamma V_\theta(s') - Q_\phi(s,a))^2]. \quad (11)$$

Naively maximizing $Q$ over $\pi$ in Eq. 10 can result in the selection of out-of-distribution actions in the offline setting. To prevent extrapolation error in the offline setting, we rely on an implicit maximizer [Garg et al., 2023] that estimates the maximum over the $Q$-function conservatively with in-distribution samples.

$$\mathcal{J}(\theta) = \mathbb{E}_{s,a \sim d_{\text{mix}}^{E,S}(s,a)}[\exp((Q_\phi(s,a) - V_\theta(s))/\tau) + (Q_\phi(s,a) - V_\theta(s))/\tau]. \quad (12)$$

Finally, the policy is extracted via advantage-weighted regression [Peters and Schaal, 2007]:

$$\mathcal{M}(\psi) = \max_\psi \mathbb{E}_{s,a \sim d_{\text{mix}}^{E,S}(s,a)}[\exp(\alpha(Q_\phi(s,a) - V_\theta(s)))\log(\pi_\psi(a|s))]. \quad (13)$$

## 6 $f$-**DVL** : BETTER IMPLICIT MAXIMIZERS FOR OFFLINE RL

Proposition 2 shows that XQL is a particular `dual-V` problem where the Gumbel loss is the conjugate $f_p^*$ corresponding to reverse KL divergence. This insight allows us to extend XQL by

choosing different $f$-divergences, where the conjugate functions are more amenable to optimization. We further show that the proposed methods enjoy both improved performance and better training stability in Section 7.

Implicit policy improvement algorithms iterate two steps alternately: 1) regress $Q(s, a)$ to $r(s, a) + \gamma V(s')$ for transition $(s, a, s')$ and 2) estimate $V(s) = \max_{a \in A} Q(s, a)$. The learned $Q$, $V$ functions can be used to extract a policy. As for the `dual-V` formulation, see Appendix B.1.6. Step 1) is akin to the *policy evaluation* step of generalized policy iteration (GPI), and step 2) acts like the *policy improvement* step without explicitly learning a policy $\pi(s) = \arg \max_a Q(s, a)$. The crux is to conservatively estimate the maximum of $Q$ in step 2.

Consider a rewriting of `dual-V` with the temperature parameter $\lambda$ and a chosen surrogate function $\bar{f}_p^*$ that extends the domain of $f_p^*$ to $\mathbb{R}$. We discuss the need for a surrogate function below.

$$\min_V (1 - \lambda) \mathbb{E}_{s \sim d^O} [V(s)] + \lambda \mathbb{E}_{(s,a) \sim d^O} \big[ \bar{f}_p^* \big( \bar{Q}(s, a) - V(s) \big) \big], \tag{14}$$

where $\bar{Q}(s, a)$ denotes `stop-gradient`$(r(s, a) + \gamma \sum_{s'} p(s'|s, a) V(s'))$. Let $x$ be a random variable of distribution $D$. Eq (14) can be considered as a special instance of the following problem:

$$\min_v (1 - \lambda) v + \lambda \mathbb{E}_{x \sim D} \big[ \bar{f}_p^* (x - v) \big], \tag{15}$$

where $x$ is analogous to $\bar{Q}$ and $v$ is analogous to $V$. As opposed to handcrafted choices [Kostrikov et al., 2021, Garg et al., 2023], we show through Proposition 3 below that objective (15) naturally gives rise to a family of *implicit maximizers* that estimates $\sup_{x \sim D} x$ as $\lambda \to 1$.

**Proposition 3.** *Let $x$ be a real-valued random variable such that $\Pr(x > x^*) = 0$. Let $v_\lambda$ be the solution of Problem (15). It holds that $v_{\lambda_1} \leqslant v_{\lambda_2}, \forall 0 < \lambda_1 < \lambda_2 < 1$. Further, $\lim_{\lambda \to 1} v_\lambda = x^*$.*

Figure 5 provides an illustration for Proposition 3. We propose a family of maximizers associated with different $f$-divergences and apply them to `dual-V`. We call the resulting methods $f$-`DVL` (Dual-V Learning).

**Practical Considerations and Algorithm** A practical issue for optimizing Eq 14 is that $f_p^*$ is not well defined over the entire domain of $\mathbb{R}$. To remedy this, we consider an extension of $f_p^*$ on $\mathbb{R}$ that leads to the surrogate function denoted by $\bar{f}_p^*$: for (1) Total Variation: $f(x) = \frac{1}{2}|x - 1|$, $\bar{f}_p^*(y) = \max(y, 0)$, (2) Pearson $\chi^2$ divergence: $f(x) = (x - 1)^2$, $\bar{f}_p^*(y) = \max(\frac{1}{4}y^2 + y, 0)$. We defer the derivation of these surrogates to Appendix F.3. Recall that XQL uses the implicit maximizer associated with reverse KL divergence, where $f_p^*$ is exponential. Compared with XQL, our $\bar{f}_p^*$ functions are low-order polynomials and are thus stable for optimization. Algorithm 2 details the steps for $f$-`DVL`.

---

**Algorithm 2:** $f$-`DVL` (Under Stochastic Dynamics)

1: Initialize $Q_\phi, V_\theta, \pi_\psi$, temperature $\alpha$, weight $\lambda$
2: Let $\mathcal{D} = (s, a, r, s')$ be offline dataset
3: **for** $t = 1..T$ iterations **do**
4:     Train $Q_\phi$ by minimizing:
    $\mathbb{E}_{s,a,s' \sim \mathcal{D}} \big[ (Q_\phi(s, a) - (r(s, a) + \gamma V_\theta(s')))^2 \big]$.
5:     Train $V_\theta$ by minimizing Eq 14
    with surrogate $\bar{f}_p^*$
6:     Update $\pi_\psi$ by maximizing:
    $\mathbb{E}_{s,a \sim \mathcal{D}} [e^{\alpha(Q_\phi(s,a) - V_\theta(s))} \log \pi_\psi(s|a)]$.
7: **end for**

---

## 7 EXPERIMENTS

Our experiments aim to answer the following four questions. **IL:** 1) How does `ReCOIL` perform and compare with previous offline IL methods? 2) Can `ReCOIL` accurately estimate the policy visitation distribution $d^\pi$ and the reward function/intent of the expert? **RL:** 3) How does $f$-`DVL` perform and compare with previous offline RL methods? 4) Is the training of $f$-`DVL` more stable than XQL?

In order to circumvent the intricacies associated with exploration and direct our attention towards the intrinsic nature of dual RL formulation, we focus on the offline setting in this section, although the approaches can also be applied to online settings. We consider the locomotion and manipulation tasks from the D4RL benchmark [Fu et al., 2020], and report the results in Section 7.1 and 7.2, respectively. For each algorithm, we train 7 instances with different seeds and report their average return and standard derivation. Complete experiment details can be found in Appendix F.

### 7.1 OFFLINE IL

**Benchmark Comparisons** For every task, our agent is given 1 expert demonstration and a set of suboptimal transitions, both extracted from the D4RL datasets. We follow the construction

Figure 1: (a) [Left] shows an MDP that starts at the leftmost state and transitions to one of the five absorbing states on the right. Under the given expert and replay/offline visitation we study if a prespecified policy's visitation can be inferred whose ground truth visitation is known [Right] shows MSE error plots with policy's ground truth visitation where ReCOIL perfectly infers $d^\pi$ whereas a method that only relies on expert data or the replay data with the coverage assumption fails. Results averaged over 100 seeds. More details in Appendix G.3 (b) Recovered $R$ and $V^*$ on a simple grid-world environment by ReCOIL.

of suboptimal dataset in SMODICE [Ma et al., 2022]. For locomotion tasks, the suboptimal dataset consists of 1 million transitions of the random or medium D4RL datasets and 200 expert demonstrations, which we label as 'random+expert' and 'medium+expert', respectively. We also consider suboptimal datasets mixed with only 30 expert demonstrations, which are called `random+few-expert` and `medium+few-expert` to simulate a more difficult setting. Similarly, we construct datasets for the manipulation tasks. More details in Appendix F.2.

| Suboptimal Dataset | Env | RCE | ORIL | SMODICE | BC (only expert data) | BC (full dataset) | IQ-Learn (offline) | ReCOIL | Expert |
|---|---|---|---|---|---|---|---|---|---|
| random+ expert | hopper | 51.41±38.63 | 73.93±11.06 | **101.61±7.69** | 4.52±1.42 | 5.64±4.83 | 1.85±2.19 | 108.18±3.28 | 111.33 |
| | halfcheetah | 64.19±11.06 | 60.49±3.53 | **80.16±7.30** | 2.2±0.01 | 2.25±0.00 | 4.83±7.99 | **80.20±6.61** | 88.83 |
| | walker2d | 20.90±26.80 | 2.86±3.39 | **105.86±3.47** | 0.86±0.61 | 0.91±0.5 | 0.57±0.09 | 102.16±7.19 | 106.92 |
| | ant | 105.38±14.15 | 73.67±12.69 | **126.78±5.12** | 5.17±5.43 | 30.66±1.35 | 42.23±20.05 | 126.74±4.63 | 130.75 |
| random+ few-expert | hopper | 25.31±18.97 | 42.04±13.76 | 60.11±18.28 | 4.84±3.83 | 3.0±0.54 | 1.37±1.23 | **97.85±17.89** | 111.33 |
| | halfcheetah | 2.99±1.07 | 2.84±5.52 | 2.28±0.62 | -0.93±0.35 | 2.24±0.01 | 1.14±1.94 | **76.92±7.53** | 88.83 |
| | walker2d | 40.49±26.52 | 3.22±3.29 | **107.18±1.87** | 0.98±0.83 | 0.74±0.20 | 0.39±0.27 | 83.23±19.00 | 106.92 |
| | ant | **67.62±15.81** | 25.41±8.58 | -6.10±7.85 | 0.91±3.93 | 35.38±2.66 | 32.99±3.12 | 67.14±8.30 | 130.75 |
| medium+ expert | hopper | 58.71±34.06 | 61.68±7.61 | 49.74±3.62 | 16.09±12.80 | 59.25±3.71 | 12.90±24.00 | **88.51±16.73** | 111.33 |
| | halfcheetah | 65.14±13.82 | 54.66±0.88 | 59.50±0.82 | -1.79±0.22 | 42.45±0.42 | 25.67±20.82 | **81.15±2.84** | 88.83 |
| | walker2d | **96.24±14.04** | 8.19±7.70 | 2.62±0.93 | 2.43±1.82 | 72.76±3.82 | 59.37±30.14 | 108.54±1.81 | 106.92 |
| | ant | **86.14±38.59** | 102.74±6.63 | 104.95±6.43 | 0.86±7.42 | 95.47±10.37 | 37.17±41.15 | 120.36±7.67 | 130.75 |
| medium few-expert | hopper | **66.15±35.16** | 17.40±15.15 | 47.61±7.08 | 7.37±1.13 | 46.87±5.31 | 11.05±20.59 | 50.01±10.36 | 111.33 |
| | halfcheetah | **61.14±18.31** | 43.24±0.75 | 46.45±3.12 | -1.15±0.06 | 42.21±0.06 | 26.27±20.24 | 75.96±4.54 | 88.83 |
| | walker2d | **85.28±34.90** | 6.81±6.76 | 6.00±6.69 | 2.02±0.72 | 70.42±2.86 | 73.30±2.85 | 91.25±17.63 | 106.92 |
| | ant | **67.95±36.78** | 81.53±8.618 | 81.53±8.618 | -10.45±1.63 | 81.63±6.67 | 35.12±50.56 | 110.38±10.96 | 130.75 |
| cloned+expert | pen | 19.60±11.40 | -3.10±0.40 | -3.36±0.71 | 13.95±11.04 | 34.94±11.10 | 2.18±8.75 | **95.04±4.48** | 106.42 |
| | door | 0.08±0.15 | -0.33±0.01 | 0.25±0.54 | -0.22±0.05 | 0.011±0.00 | 0.07±0.02 | **102.75±4.05** | 103.94 |
| | hammer | 1.95±3.89 | 0.25±0.01 | 0.15±0.078 | 2.41±4.48 | 5.45±7.84 | 0.27±0.02 | **95.77±17.90** | 125.71 |
| | relocate | -0.25±0.04 | -0.29±0.01 | 1.75±3.85 | -0.17±0.04 | -0.24±0.01 | -0.1±0.12 | **67.43±14.60** | 118.39 |
| human+expert | pen | 17.81±5.91 | -3.38±2.29 | -2.20±2.40 | 13.83±10.76 | 90.76±25.09 | 14.29±28.82 | **103.72±2.90** | 106.42 |
| | door | -0.05±0.05 | -0.33±0.01 | -0.20±0.11 | -0.03±0.05 | 103.71±1.22 | 5.6±7.29 | **104.70±0.55** | 103.94 |
| | hammer | 5.00±5.64 | 1.89±0.70 | -0.07±0.39 | 0.18±0.14 | 122.61±4.85 | 5.32±1.38 | **125.19±3.29** | 125.71 |
| | relocate | 0.02±0.10 | -0.29±0.01 | -0.16±0.04 | -0.13±0.11 | 81.19±7.73 | -0.04±0.22 | **91.98±2.89** | 118.39 |
| partial+expert | kitchen | 6.875±9.24 | 0.00±0.00 | 39.16±1.17 | 2.5±5.0 | 45.5±1.87 | 0.0±0.0 | **60.0±5.70** | 75.0 |
| mixed+expert | kitchen | 1.66±2.35 | 0.00±0.00 | 42.5±2.04 | 2.2±3.8 | 42.1±1.12 | 0.0±0.0 | **52.0±1.0** | 75.0 |

Table 2: The normalized return obtained by different offline IL methods trained on the D4RL suboptimal datasets with 1 expert trajectory. Methods with avg. perf within the std-dev of the top performing method is highlighted.

We compare ReCOIL against recent offline IL methods RCE [Eysenbach et al., 2021], IQLearn [Garg et al., 2021], SMODICE [Ma et al., 2022], ORIL [Zolna et al., 2020] and behavior cloning. We do not compare to DEMODICE [Kim et al., 2022b] and ValueDICE [Kostrikov et al., 2019] as SMODICE was shown to outperform DEMODICE in Ma et al. [2022] and IQLearn was shown to outperform ValueDICE in [Garg et al., 2021] on the same environments. Both SMODICE and ORIL require learning a discriminator, and SMODICE relies on the coverage assumption. RCE also uses a recursive discriminator to test the proximity of the policy visitations to successful examples. In contrast, ReCOIL is discriminator-free and does not need this coverage assumption. Table 2 shows that ReCOIL strongly outperforms the baselines in most environments. SMODICE exhibits poor performance in cases when the combined offline dataset has few expert samples (`random+few-expert`) or where the discriminator can easily overfit (high-dimensional environments like dextrous manipulation).

**Estimation of the Policy Visitation Distribution and Reward Recovery** Correctly estimating a given policy's visitation distribution $d^\pi$ is key to testing its closeness to the expert visitation. $d^\pi$ can be computed via Eq (49) (appendix). Figure 1a and Figure 11 show that ReCOIL can estimate $d^\pi$ more accurately than SMODICE [Ma et al., 2022] which relies on coverage assumption and IQLearn [Garg et al., 2021] which only utilizes expert data. This empirically validates our hypothesis that a learned discriminator with low coverage can lead to poor performance of the downstream policy. Further, Figure 1b shows the reward function recovered by ReCOIL for a simple grid-world task. For Hopper and Walker, we respectively observe a Pearson correlation of **0.98** and **0.92** between the recovered reward with the ground truth. See more details in Appendix G.3 and G.10.

| Dataset | BC | 10%BC | DT | TD3+BC | CQL | IQL | XQL(r) | $f$-DVL ($\chi^2$) | $f$-DVL (TV) |
|---|---|---|---|---|---|---|---|---|---|
| halfcheetah-medium-v2 | 42.6 | 42.5 | 42.6 | **48.3** | 44.0 | **47.4** | 47.4 | **47.7** | 47.5 |
| hopper-medium-v2 | 52.9 | 56.9 | 67.6 | 59.3 | 58.5 | 66.3 | **68.5** | 63.0 | 64.1 |
| walker2d-medium-v2 | 75.3 | 75.0 | 74.0 | **83.7** | 72.5 | 78.3 | 81.4 | 80.0 | 81.5 |
| halfcheetah-medium-replay-v2 | 36.6 | 40.6 | 36.6 | **44.6** | **45.5** | 44.2 | 44.1 | 42.9 | **44.7** |
| hopper-medium-replay-v2 | 18.1 | 75.9 | 82.7 | 60.9 | 95.0 | 94.7 | 95.1 | 90.7 | **98.0** |
| walker2d-medium-replay-v2 | 26.0 | 62.5 | 66.6 | **81.8** | 77.2 | 73.9 | 58.0 | 52.1 | 68.7 |
| halfcheetah-medium-expert-v2 | 55.2 | **92.9** | 86.8 | 90.7 | 91.6 | 86.7 | 90.8 | 89.3 | 91.2 |
| hopper-medium-expert-v2 | 52.5 | **110.9** | 107.6 | 98.0 | 105.4 | 91.5 | 94.0 | 105.8 | 93.3 |
| walker2d-medium-expert-v2 | 107.5 | 109.0 | 108.1 | **110.1** | 108.8 | **109.6** | **110.1** | **110.1** | **109.6** |
| antmaze-umaze-v0 | 54.6 | 62.8 | 59.2 | 78.6 | 74.0 | **87.5** | 47.7 | 83.7 | **87.7** |
| antmaze-umaze-diverse-v0 | 45.6 | 50.2 | 53.0 | 71.4 | **84.0** | 62.2 | 51.7 | 50.4 | 48.4 |
| antmaze-medium-play-v0 | 0.0 | 5.4 | 0.0 | 10.6 | 61.2 | **71.2** | 31.2 | 56.7 | **71.0** |
| antmaze-medium-diverse-v0 | 0.0 | 9.8 | 0.0 | 3.0 | 53.7 | **70.0** | 0.0 | 48.2 | 60.2 |
| antmaze-large-play-v0 | 0.0 | 0.0 | 0.0 | 0.2 | 15.8 | 39.6 | 10.7 | 36.0 | **41.7** |
| antmaze-large-diverse-v0 | 0.0 | 6.0 | 0.0 | 0.0 | 14.9 | **47.5** | 31.28 | 44.5 | 39.3 |
| kitchen-complete-v0 | 65.0 | - | - | - | 43.8 | 62.5 | 56.7 | **67.5** | 61.3 |
| kitchen-partial-v0 | 38.0 | - | - | - | 49.8 | 46.3 | 48.6 | 58.8 | **70.0** |
| kitchen-mixed-v0 | 51.5 | - | - | - | 51.0 | 51.0 | 40.4 | **53.75** | 52.5 |

Table 3: The normalized return of offline RL methods on D4RL tasks. XQL(r) denotes the results obtained under the standard evaluation protocol. Results aggregated over 7 seeds. Highlighted results are within one performance point of the best-performing algorithm.

## 7.2 OFFLINE RL

**Benchmark Comparison** Table 3 shows that $f$-DVL outperforms XQL and other prior offline RL methods [Chen et al., 2021a, Kumar et al., 2019, 2020, Kostrikov et al., 2021, Fujimoto and Gu, 2021] on a broad range of continuous control tasks. We note an inconsistency between our reproduced XQL results and the results reported in the original paper: their results were reported by taking the best average return during training as opposed to the standard practice of taking the average of the last iterate performance across different seeds at 1 million gradient steps. Such inconsistency can be validated by comparing their training plots and reported results (Fig 11 and Table 1 in Garg et al. [2023]). XQL(r) shows the results for XQL under the standard evaluation protocol.

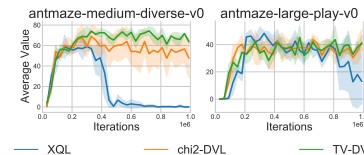

Figure 2: XQL training diverges due to the numerical instability of its loss function. $f$-DVL fixes this problem by using more well-behaved $f$-divergences.

**Training Stability** As pointed out by the authors, the exponential loss function of XQL causes numerical instabilities during optimization. As discussed in Section 6, this is a by-product of reverse KL divergence. Fig. 2 confirms that this is fixed by $f$-DVL by using other $f$-divergences with more stable loss functions. Additionally, Fig. 13 demonstrates that $f$-DVL is competitive to XQL and SAC in the online setting as well. See Appendix F for additional experimental details.

## 7.3 ADDITIONAL EXPERIMENTS

We conduct additional experiments in Appendix G. We further demonstrate a) when incorporating off-policy data in online training, traditional ADP-based methods can suffer from the over-estimation of value functions, and the performance gain is limited, whereas dual-RL methods can leverage the same data to achieve better performance (Figure 3 and Appendix G.1); b) the reward functions learned by ReCOIL are of high quality (Appendix G.10); c) the hyperparameter ablation for $f$-DVL (Appendix G.8) and qualitative results for ReCOIL (Appendix G.4).

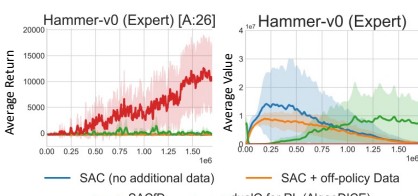

Figure 3: Augmenting SAC with expert data at the start of training destabilizes value function learning (r), but dual-RL approaches can make effective use of the additional data to learn performant policy (l).

## 8 CONCLUSION

Our work unifies a significant number of recent developments in RL and IL. The insights gleaned from the unification allow us to identify key gaps and subsequently improve performance and training stability in imitation learning and reinforcement learning. Leveraging this unification, we propose 1) a family of *stable offline RL methods* $f$-DVL relying on implicit value function maximization, 2) ReCOIL, a general *off-policy IL method* to learn from arbitrary suboptimal data while being discriminator-free. We show that $f$-DVL and ReCOIL both outperform previous methods in online/offline RL and offline IL domains, respectively. We demonstrate that Dual-RL algorithms have great potential for developing performant algorithms and warrant further study. We direct interested readers to read our Appendix containing more insights into Dual-RL and potential directions for improvements and new algorithms.

ACKNOWLEDGEMENTS

We thank Ahmed Touati, Ben Eysenbach, Siddhant Agarwal, and ICLR reviewers for valuable feedback on this work. This work has taken place in the Safe, Correct, and Aligned Learning and Robotics Lab (SCALAR) at The University of Massachusetts Amherst and Machine Intelligence through Decision-making and Interaction (MIDI) Lab at The University of Texas at Austin. SCALAR research is supported in part by the NSF (IIS-2323384), AFOSR (FA9550-20-1-0077), and ARO (78372-CS, W911NF-19-2-0333), and the Center for AI Safety (CAIS). This research was also sponsored by the Army Research Office under Cooperative Agreement Number W911NF-19-2-0333. HS and AZ are funded in part by a sponsored research agreement with Cisco Systems Inc. The views and conclusions contained in this document are those of the authors and should not be interpreted as representing the official policies, either expressed or implied, of the Army Research Office or the U.S. Government. The U.S. Government is authorized to reproduce and distribute reprints for Government purposes notwithstanding any copyright notation herein.

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

# Appendix

## Table of Contents

## A   LIMITATIONS AND NEGATIVE SOCIETAL IMPACTS

**Limitations**:   Notably, none of the supervised learning approaches, like Upside-Down RL [Schmidhuber, 2019], DT [Chen et al., 2021a, Zheng et al., 2022] and RvS [Emmons et al., 2021], can be cast as instances of this framework. The main reason is that there is no return optimization term in the supervised learning loss for those approaches. Second, our framework is constrained to the regularized RL setting where the regularization coefficient $\alpha > 0$ (see Eq. 1 ). Finally, our framework considers a static regularization distribution $d^O$, and we believe that regularization with dynamically changing distributions, like those in on-policy methods (PPO [Schulman et al., 2017], TRPO [Schulman et al., 2015], etc) requires a more careful theoretical treatment. Another limitation of the paper is the assumption that the expert demonstrations used in the imitation learning process are always of high quality and provide the desired behavior. In practice, obtaining high-quality demonstrations can be challenging, especially in complex environments where the behavior of the expert is not always clear. The performance of the proposed approach could be limited in cases where the expert demonstrations are of poor quality or where the behavior of the expert does not correspond to the desired behavior.

**Negative Societal Impacts**: As machine learning algorithms continue to grow in sophistication, it is important to consider the potential risks and harms associated with their use. One such area of concern is imitation learning, which involves training a model to imitate a desired behavior by providing it with examples of that behavior. However, this approach can be problematic if the demonstration data includes harmful behaviors, whether intentional or not. Even in cases where the demonstration data is of high quality and desirable behavior is learned, the algorithm may still fall short of providing sufficient guarantees of performance. In high-stakes domains, the use of such algorithms without appropriate safety checks on learned behaviors could lead to serious consequences. As such, it is crucial to carefully consider the potential risks and benefits of imitation learning, and to develop strategies for ensuring safe and effective use of these algorithms in real-world application

## B   DUAL REINFORCEMENT LEARNING

### B.1   A REVIEW OF DUAL-RL

In this section, we aim to give a self-contained review for Dual Reinforcement Learning. For a more thorough read, refer to [Nachum and Dai, 2020]. Our proofs will take a different approach than [Nachum and Dai, 2020] which we believe leads to easier exposition and understanding.

#### B.1.1   CONVEX CONJUGATES AND $f$-DIVERGENCE

We first review the basics of duality in reinforcement learning. Let $f : \mathbb{R}_+ \to \mathbb{R}$ be a convex function. The convex conjugate $f^* : \mathbb{R}_+ \to \mathbb{R}$ of $f$ is defined by:

$$f^*(y) = \sup_{x \in \mathbb{R}_+}[xy - f(x)]. \tag{16}$$

The convex conjugates have the important property that $f^*$ is also convex and the convex conjugate of $f^*$ retrieves back the original function $f$. We also note an important relation regarding $f$ and $f^*$: $(f^*)' = (f')^{-1}$, where the $'$ notation denotes first derivative.

Going forward, we would be dealing extensively with $f$-divergences.   Informally, $f$-divergences [Rényi, 1961] are a measure of distance between two probability distributions. Here's a more formal definition:

Let $P$ and $Q$ be two probability distributions over a space $\mathcal{Z}$ such that $P$ is absolutely continuous with respect to $Q$ [3]. For a function $f : \mathbb{R}_+ \to \mathbb{R}$ that is a convex lower semi-continuous and $f(1) = 0$,

---

[3]Let $z$ denote the random variable. For any measurable set $Z \subseteq \mathcal{Z}$, $Q(z \in Z) = 0$ implies $P(z \in Z) = 0$.

| Divergence Name | Generator $f(x)$ | Conjugate $f*(y)$ |
|---|---|---|
| Reverse KL | $x \log x$ | $e^{(y-1)}$ |
| Squared Hellinger | $(\sqrt{x} - 1)^2$ | $\frac{y}{1-y}$ |
| Pearson $\chi^2$ | $(x - 1)^2$ | $y + \frac{y^2}{4}$ |
| Total Variation | $\frac{1}{2}|x - 1|$ | $y$ if $y \in \left[-\frac{1}{2}, \frac{1}{2}\right]$ otherwise $\infty$ [4] |
| Jensen-Shannon | $-(x + 1) \log(\frac{x+1}{2}) + x \log x$ | $-\log(2 - e^y)$ |

Table 4: List of common $f$-divergences.

the $f$-divergence of $P$ from $Q$ is

$$D_f(P \,||\, Q) = \mathbb{E}_{z \sim Q}\left[f\left(\frac{P(z)}{Q(z)}\right)\right]. \tag{17}$$

Table 4 lists some common $f$-divergences with their generator functions $f$ and the conjugate functions $f*$.

### B.1.2 AN OVERVIEW OF REINFORCEMENT LEARNING VIA LAGRANGIAN DUALITY

In this section, we give a more detailed review than we are able to in the main text due to space constraints. We consider RL problems with their average return considered in the form of a convex program with linear constraints [Manne, 1960], to which we apply Lagrangian duality to obtain corresponding constraint-free problems. This framework was first introduced in the work of Nachum and Dai [2020], which obtains the same formulations as ours via Fenchel-Rockfeller duality. Here we use Lagrangian duality for its simplicity and popularity.

Consider the following regularized policy learning problem

$$\max_\pi J(\pi) = \mathbb{E}_{d^\pi(s,a)}[r(s,a)] - \alpha D_f(d^\pi(s,a) \,||\, d^O(s,a)), \tag{18}$$

where $D_f(d^\pi(s,a) \,||\, d^O(s,a))$ is a conservatism regularizer that encourages the visitation distribution of $\pi$ to stay close to some distribution $d^O$, and $\alpha$ is a temperature parameter that balances the expected return and the conservatism.

An interesting fact is that $J(\pi)$ can be rewritten as a convex problem that searches for an *achievable* visitation distribution that satisfies the *Bellman-flow* constraints:

$$J(\pi) = \max_d \; \mathbb{E}_{d(s,a)}[r(s,a)] - \alpha D_f(d(s,a) \,||\, d^O(s,a)) \tag{19}$$
$$\text{s.t } d(s,a) = (1-\gamma)d_0(s).\pi(a|s) + \gamma \sum_{s',a'} d(s',a')p(s|s',a')\pi(a|s), \; \forall s \in \mathcal{S}, a \in \mathcal{A}.$$

Applying Lagrangian duality and convex conjugate (16) to this problem, we can convert it to an unconstrained problem with dual variables $Q(s,a)$ defined for all $s, a \in \mathcal{S} \times \mathcal{A}$:

$$\min_Q (1-\gamma)\mathbb{E}_{s \sim d_0, a \sim \pi(s)}[Q(s,a)] + \alpha \mathbb{E}_{(s,a) \sim d^O}[f^*([\mathcal{T}_r^\pi Q(s,a) - Q(s,a)]/\alpha)], \tag{20}$$

where $f*$ is the convex conjugate of $f$. We defer the derivation to the next section. As problem (19) is convex, strong duality holds and problems (19) and (20) have the same optimal objective value up to a constant scaling[5]. We refer to the nested policy learning problem where $J(\pi)$ is of form (19) as `primal-Q` and the joint problem with scaled $J(\pi)$ of form (20) as `dual-Q`.

`primal-Q` $\max_\pi [J(\pi)$ in the form Eq. (2)], $\tag{21}$

`dual-Q` $\max_\pi \min_Q (1-\gamma)\mathbb{E}_{s \sim d_0, a \sim \pi(s)}[Q(s,a)] + \alpha \mathbb{E}_{(s,a) \sim d^O}[f^*([\mathcal{T}_r^\pi Q(s,a) - Q(s,a)]/\alpha)].$

$$\tag{22}$$

In fact, problem (19) is overconstrained – the maximization w.r.t $d$ is unnecessary, as for a fixed $\pi$ the $|\mathcal{S}| \times |\mathcal{A}|$ equality constraints already uniquely determine a solution $d^\pi$ [Puterman, 2014]. Let $\pi*, d*$ be the optimal policy and corresponding visitation distribution. In fact, we can relax the constraints to get another problem [Agarwal et al., 2019] with the same optimal solution $d*$, which we call

---

[4]For our derivations, we will consider smooth extensions of TV conjugate function

[5]We scaled the dual problem by $1/\alpha$ for derivation simplicity.

`primal-V` below:

`primal-V` $\quad \max_{d \geqslant 0} \mathbb{E}_{d(s,a)}[r(s,a)] - \alpha D_f(d(s,a) \,||\, d^O(s,a))$

$\text{s.t} \sum_{a \in \mathcal{A}} d(s,a) = (1-\gamma)d_0(s) + \gamma \sum_{(s',a') \in \mathcal{S} \times \mathcal{A}} d(s',a')p(s|s',a'), \; \forall s \in \mathcal{S}.$
$$\tag{23}$$

Comparing with problem (19), the constraints are relaxed and there is no policy $\pi$ in this formulation. In fact, as opposed to `primal-Q`, which needs to solve nested inner problems, `primal-V` solves a single problem to obtain $d^*$, from which we can recover $\pi^*$ via Eq. (24)[6]:

$$\pi(a|s) = d^\pi(s,a) / \sum_{a \in \mathcal{A}} d^\pi(s,a). \tag{24}$$

Similarly, we consider the Lagrangian dual of (23), with dual variables $V(s)$ defined for all $s \in S$:

`dual-V` $\quad \min_V (1-\gamma)\mathbb{E}_{s \sim d_0}[V(s)] + \alpha \mathbb{E}_{(s,a) \sim d^O}\left[f_p^*\left([\mathcal{T}V(s,a) - V(s)]/\alpha\right)\right], \tag{25}$

where $f_p^*$ is a variant of $f^*$ defined in Eq. (47). Such modification is to cope with the nonnegativity constraint $d(s,a) \geqslant 0$ in `primal-V`. This constraint is ignored in `primal-Q` because the constraints of the inner problem (19) already uniquely identify the solution. See Appendix B.1.4 for the derivation. As before, strong duality holds here (up to a factor of $1/\alpha$), and we can compute the optimal policy $\pi^*$ after obtaining $V^*$. We discuss this in detail in Appendix B.1.6.

*Remark 1.* The above formulations generalizes to the popular MaxEnt RL framework, where the objective $J(\pi)$ contains an extra policy entropy regularizer. One only needs to replace the Bellman operator $\mathcal{T}_r^\pi$ by its soft variant: $\mathcal{T}_{r,\text{soft}}^\pi Q(s,a) = r(s,a) + \gamma\mathbb{E}_{s',a'}[Q(s',a') - \log\pi(a'|s')]$.

*Remark 2.* We derive the dual problems via the Lagrangian duality. Taking the `primal-Q` problem as an example, the key step which bridges its Lagrangian dual problem $\min_Q \max_d L(Q,d)$ and the final formulation `dual-Q` is that the maximizer $d^*$ of the inner problem has a closed form solution. Equivalently, we can rewrite the inner problem $\max_d L(Q,d)$ via the convex conjugate (32), which eliminates the variable $d$. The Fenchel-Rockerfeller duality provides an alternative way to directly reach the same formulation, where one first rewrites the linear constraints as part of the objective using the Dirac delta function [Nachum and Dai, 2020].

*Remark 3.* The dual formulations have a few appealing properties. (a) They allow us to transform constrained distribution-matching problems, w.r.t previously logged data, into unconstrained forms. (b) One can show that the gradient of `dual-Q` w.r.t $\pi$, when $Q$ is optimized for the inner problem, is the on-policy policy gradient computed by off-policy data. This key property relieves the instability or divergence issue in off-policy learning. (c) The dual framework can be extended to the max-entropy RL setting, where $J(\pi)$ consists of additional entropy regularization, by replacing Bellman-operator with their soft Bellman counterparts [Haarnoja et al., 2017].

### B.1.3 DERIVING `DUAL-Q`

We again consider the RL problem as a maximization of a convex program for estimating policy performance $J(\pi)$ by considering optimization over *achievable* state-action visitations (i.e $\max_\pi J(\pi)$):

$$\max_\pi \left[ \max_{d \geqslant 0} \mathbb{E}_{d(s,a)}[r(s,a)] - \alpha D_f(d(s,a) \,||\, d^O(s,a)) \right. \tag{26}$$

$$\left. \text{s.t } d(s,a) = (1-\gamma)d_0(s).\pi(a|s) + \gamma \sum_{s',a'} d(s',a')p(s|s',a')\pi(a|s) \right], \tag{27}$$

where $\alpha$ allows us to weigh policy improvement against conservatism from staying close to the state-action distribution $d^O$. We will assume that $d^O$ has non-zero coverage over $\mathcal{S} \times \mathcal{A}$ for the derivation below.

A careful reader may notice that the inner problem is overconstrained and overparameterized. The solution to the inner maximization problem with respect to $d$ is uniquely determined by the $|\mathcal{S}| \times |\mathcal{A}|$ linear constraints, and the nonnegativity constraint $d \geqslant 0$ is not necessary. Moreover, given a fixed policy $\pi$, the solution of the inner problem is its visitation distribution $d^\pi$.

---

[6]Eq. (24) can be easily computed for discrete actions, yet it is difficult for continuous actions. While our analysis focuses on the tabular case, we discuss two methods for recovering $\pi^*$ for continuous actions in Appendix B.1.6.

The constraints of the inner problem are known as the *Bellman flow equations* that an achievable stationary state-action distribution must satisfy. The next question is how can we solve it? Here is where Lagrangian duality comes into play. First, we form the Lagrangian dual of our original optimization problem, transforming our constrained optimization into an unconstrained form. This introduces additional optimization variables - the Lagrange multipliers $Q$.

As mentioned before, we can discard the nonnegativity constraint $d \geqslant 0$ as the other constraints imply a unique solution for $d$. Focusing on the inner optimization problem, we optimize the Lagrangian dual problem:

$$\min_{Q(s,a)} \max_d \mathbb{E}_{s,a\sim d(s,a)}[r(s,a)] - \alpha D_f(d(s,a) \,||\, d^O(s,a))$$

$$+ \sum_{s,a} Q(s,a) \left( (1-\gamma)d_0(s).\pi(a|s) + \gamma \sum_{s',a'} d(s',a')p(s|s',a')\pi(a|s) - d(s,a) \right),$$

where $Q(s,a)$ are the Lagrange multipliers associated with the equality constraints. We can now do some simple algebraic manipulation to further simplify it:

$$\min_{Q(s,a)} \max_d \mathbb{E}_{s,a\sim d(s,a)}[r(s,a)] - \alpha D_f(d(s,a) \,||\, d^O(s,a))$$

$$+ \sum_{s,a} Q(s,a) \left( (1-\gamma)d_0(s).\pi(a|s) + \gamma \sum_{s',a'} d(s',a')p(s|s',a')\pi(a|s) - d(s,a) \right) \tag{28}$$

$$= \min_{Q(s,a)} \max_d (1-\gamma)\mathbb{E}_{d_0(s),\pi(a|s)}[Q(s,a)]$$

$$+ \mathbb{E}_{s,a\sim d}\left[ r(s,a) + \gamma \sum_{s',a'} p(s'|s,a)\pi(a'|s')Q(s',a') - Q(s,a) \right] - \alpha D_f(d(s,a) \,||\, d^O(s,a)), \tag{29}$$

where we swap the maximum and minimum in the last step as strong duality holds for this problem. This is equivalent to solving the following scaled objective (scaled by $1/\alpha$).

$$\min_{Q(s,a)} \max_d \frac{(1-\gamma)}{\alpha}\mathbb{E}_{d_0(s),\pi(a|s)}[Q(s,a)]$$

$$+ \mathbb{E}_{s,a\sim d}\left[ (r(s,a) + \gamma \sum_{s',a'} p(s'|s,a)\pi(a'|s')Q(s',a') - Q(s,a))/\alpha \right] - D_f(d(s,a) \,||\, d^O(s,a)) \tag{30}$$

$$= \min_{Q(s,a)} \frac{(1-\gamma)}{\alpha}\mathbb{E}_{d_0(s),\pi(a|s)}[Q(s,a)]$$

$$+ \mathbb{E}_{s,a\sim d^O}\left[ f^*((r(s,a) + \gamma \sum_{s',a'} p(s'|s,a)\pi(a'|s')Q(s',a') - Q(s,a))/\alpha) \right], \tag{31}$$

where we applied the convex conjugate (Eq. (16)) in the last step. To see this more clearly, let $y(s,a) = \left( r(s,a) + \gamma \sum_{s',a'} p(s'|s,a)\pi(a'|s')Q(s',a') - Q(s,a) \right)/\alpha$. Then, under mild conditions that the interchangeability principle [Dai et al., 2017] is satisfied, and $d^O$ has sufficient support over $\mathcal{S} \times \mathcal{A}$ [Nachum and Dai, 2020], it holds that

$$\max_d \mathbb{E}_{s,a\sim d}[y(s,a)] - D_f(d(s,a) \,||\, d^O(s,a)) \tag{32}$$

$$= \max_d \mathbb{E}_{s,a\sim d^O}\left[ \frac{d(s,a)}{d^O(s,a)}y(s,a) - f\left( \frac{d(s,a)}{d^O(s,a)} \right) \right] \tag{33}$$

$$= \mathbb{E}_{d^O}[f^*(y(s,a))]. \tag{34}$$

We have transformed the problem of computing $J(\pi)$ to solving Eq. (31). Finally, the policy optimization problem $\max_\pi J(\pi)$ is reduced to solving the following min-max optimization problem, which we will refer to as dual-Q:

$$\max_\pi \min_Q \frac{(1-\gamma)}{\alpha}\mathbb{E}_{d_0(s),\pi(a|s)}[Q(s,a)] + \mathbb{E}_{s,a\sim d^O}\left[ f^*((r(s,a) + \gamma \sum_{s',a'} p(s'|s,a)\pi(a'|s')Q(s',a') - Q(s,a))/\alpha) \right]. \tag{35}$$

Table 4 lists the corresponding convex conjugates $f^*$ for common $f$-divergences.

In the case of deterministic policy and deterministic dynamics, the above-obtained optimization takes a simpler form:

$$\max_{\pi(a|s)} \min_{Q(s,a)} \frac{(1-\gamma)}{\alpha} \mathbb{E}_{\rho_0(s)}[Q(s,\pi(s))] + \mathbb{E}_{s,a \sim d^O}\left[f^*((r(s,a) + \gamma Q(s',\pi(s')) - Q(s,a))/\alpha)\right]$$

(36)

Now, we have seen how we can transform a regularized RL problem into its `dual-Q` form which uses Lagrange variables in the form of state-action functions. Interestingly, we can go further to transform the regularized RL problem into Lagrange variables (V) that only depend on the state, and in doing so we also get rid of the two-player nature (min-max optimization) in the `dual-Q`.

### B.1.4 DERIVING `DUAL-V`

One important constraint we have not discussed so far is that the variable $d$ we are optimizing must be nonnegative. This constraint is not needed for `primal-Q`, as for the inner problem (2), the solution is uniquely determined by the constraints. Nonetheless, it is important we consider this constraint for `primal-V` and derive the correct dual problem.

In `primal-V`, we formulate the visitation constraints to depend solely on states rather than state-action pairs. Note that doing this does not change the solution $\pi^*$ for the regularized RL problem (Eq (18)). We consider $\alpha = 1$ for the sake of exposition. Interested readers can derive the result for $\alpha \neq 1$ as in the `dual-Q` case above. Recall the formulation of `primal-V`:

$$\max_{d \geqslant 0} \mathbb{E}_{d(s,a)}[r(s,a)] - D_f(d(s,a) \,||\, d^O(s,a))$$

$$\text{s.t} \quad \sum_{a \in \mathcal{A}} d(s,a) = (1-\gamma)d_0(s) + \gamma \sum_{s',a'} d(s',a')p(s|s',a').$$

(37)

As before, we construct the Lagrangian dual to this problem. Note that our constraints now solely depend on $s$.

$$\min_{V(s)} \max_{d \geqslant 0} \mathbb{E}_{s,a \sim d(s,a)}[r(s,a)] - D_f(d(s,a) \,||\, d^O(s,a))$$

$$+ \sum_s V(s)\left((1-\gamma)d_0(s) + \gamma \sum_{s',a'} d(s',a')p(s|s',a') - \sum_{a \in \mathcal{A}} d(s,a)\right)$$

(38)

Using similar algebraic manipulations we used to obtain `dual-Q` in Section B.1.3, we have :

$$\min_{V(s)} \max_{d(s,a) \geqslant 0} \mathbb{E}_{s,a \sim d(s,a)}[r(s,a)] - D_f(d(s,a) \,||\, d^O(s,a))$$

$$+ \mathbb{E}_{s,a \sim d}\left[r(s,a) + \gamma \sum_{s'} p(s'|s,a)V(s') - V(s)\right] - D_f(d(s,a) \,||\, d^O(s,a))$$

(39)

$$= \min_{V(s)} \max_{d(s,a) \geqslant 0} (1-\gamma)\mathbb{E}_{d_0(s)}[V(s)]$$

$$+ \mathbb{E}_{s,a \sim d}\left[r(s,a) + \gamma \sum_{s'} p(s'|s,a)V(s') - V(s)\right] - D_f(d(s,a) \,||\, d^O(s,a))$$

(40)

$$= \min_{V(s)} \max_{d(s,a) \geqslant 0} (1-\gamma)\mathbb{E}_{d_0(s)}[V(s)]$$

$$+ \mathbb{E}_{s,a \sim d^O}\left[\frac{d(s,a)}{d^O(s,a)}\left(r(s,a) + \gamma \sum_{s'} p(s'|s,a)V(s') - V(s)\right)\right] - \mathbb{E}_{s,a \sim d^O}\left[f\left(\frac{d(s,a)}{d^O(s,a)}\right)\right]$$

(41)

Let $w(s,a) = \frac{d(s,a)}{d^O(s,a)}$ and $\delta_V(s,a) = r(s,a) + \gamma \sum_{s'} p(s'|s,a)V(s') - V(s)$ denote the TD error. The last equation becomes

$$\min_{V(s)} \max_{w(s,a) \geqslant 0} (1-\gamma)\mathbb{E}_{d_0(s)}[V(s)] + \mathbb{E}_{s,a \sim d^O}[w(s,a)(\delta_V(s,a))] - \mathbb{E}_{s,a \sim d^O}[f(w(s,a))].$$

(42)

We now direct the attention to the inner maximization problem and derive a closed-form solution for it. Consider the Lagrangian dual problem of it:

$$\min_{\lambda \geqslant 0} \max_{w(s,a)} \mathbb{E}_{s,a \sim d^O}[w(s,a)(\delta_V(s,a))] - \mathbb{E}_{s,a \sim d^O}[f(w(s,a))] + \sum_{s,a} \lambda(s,a)w(s,a)$$

(43)

where the parameters $\lambda(s,a)$ for all $s \in S$ and $a \in A$ are the Lagrange multipliers. Since strong duality holds, we can use the KKT constraints to find the optimal solutions $w^*(s,a)$ and $\lambda^*(s,a)$:

1. **Primal feasibility** $w^*(s,a) \geqslant 0 \ \forall \ s,a$

2. **Dual feasibility** $\lambda^*(s,a) \geqslant 0 \ \forall \ s,a$

3. **Stationarity** $d^O(s,a)(-f'(w^*(s,a)) + \delta_V(s,a)) + \lambda^*(s,a) = 0 \ \forall \ s,a$

4. **Complementary Slackness** $w^*(s,a)\lambda^*(s,a) = 0 \ \forall \ s,a$

Using stationarity we have the following:
$$d^O(s,a)f'(w^*(s,a)) = d^O(s,a)\delta_V(s,a) + \lambda^*(s,a) \ \forall \ s,a \tag{44}$$
Now using complementary slackness only two cases are possible $w^*(s,a) \geqslant 0$ or $\lambda^*(s,a) \geqslant 0$. Combining both cases we arrive at the following solution for this constrained optimization:
$$w^*(s,a) = \max\left(0, {f'}^{-1}(\delta_V(s,a))\right) \tag{45}$$
We refer to the resulting function after plugging the solution for $w^*$ back in Eq. (42) and refer to the closed form solution for $d$ in second and third term as $f_p^*$.
$$f_p^*(\delta_V(s,a)) = w^*(s,a)(\delta_V(s,a)) - f(w^*(s,a)) \tag{46}$$
Plugging in $w^*(s,a)$ from Eq. (45) to Eq. (46), we get:
$$f_p^*(\delta_V(s,a)) = \max\left(0, {f'}^{-1}(\delta_V(s,a))\right)(\delta_V(s,a)) - f\left(\max\left(0, {f'}^{-1}(\delta_V(s,a))\right)\right) \tag{47}$$

Note that we get the original conjugate $f^*$ back if we do not consider the nonnegativity constraints:
$$f^*(s,a) = {f'}^{-1}(\delta_V(s,a))(\delta_V(s,a)) - f({f'}^{-1}(\delta_V(s,a))). \tag{48}$$
Finally, we have the following optimization to solve for `dual-V` when considering the nonnegativity constraints:

$$\texttt{dual-V: } \min_{V(s)}(1-\gamma)\mathbb{E}_{s\sim d_0}[V(s)] + \mathbb{E}_{(s,a)\sim d^O}\left[f_p^*(\delta_V(s,a))\right]$$

Some works e.g. SMODICE [Ma et al., 2022], ignore the nonnegativity constraints and use the corresponding `dual-V` formulation

$$\texttt{dual-V (w/o nonneg. constraints): } \min_V(1-\gamma)\mathbb{E}_{s\sim d_0}[V(s)] + \mathbb{E}_{(s,a)\sim d^O}[f^*(\delta_V(s,a)].$$

### B.1.5 DISCUSSION ON DUAL FORMULATIONS

In summary, we have two dual formulations for regularized policy learning:

$$\begin{aligned}\texttt{dual-Q: } &\max_\pi \min_Q(1-\gamma)\mathbb{E}_{d_0(s),\pi(a|s)}[Q(s,a)] \\ &+ \mathbb{E}_{s,a\sim d^O}[f^*\left(r(s,a) + \gamma\sum_{s'}p(s'|s,a)\pi(a'|s')Q(s',a') - Q(s,a)\right)]\end{aligned}$$

and

$$\texttt{dual-V: } \min_{V(s)}(1-\gamma)\mathbb{E}_{s\sim d_0}[V(s)] + \mathbb{E}_{(s,a)\sim d^O}\left[f_p^*(\delta_V(s,a))\right]$$

The above derivations for dual of primal RL formulation - `dual-Q` and `dual-V` brings out some important observations

- `dual-Q` and `dual-V` present off-policy policy optimization solutions for regularized RL problems which requires sampling transitions only from the off-policy distribution the policy state-action visitation is being regularized against. The gradient with respect to policy $\pi$ when $d$ is optimized in `dual-Q` can be shown to be equivalent to the on-policy policy gradient under a regularized Q-function (see Section 5.1 from [Nachum and Dai, 2020]).

- The above property allows us to solve not only RL problems but also imitation problems by setting the reward function to be zero everywhere and $d^O$ to be the expert dataset, and also offline RL problems where we want to maximize reward with the constraint

that our state-action visitation should not deviate too much from the replay buffer ($d^O =$ replay-buffer).

- `dual-V` formulation presents a way to solve the RL problem using a single optimization rather than a min-max optimization of the `primal-Q` or standard RL formulation. `dual-V` implicitly subsumes greedy policy maximization.

### B.1.6   HOW TO RECOVER THE OPTIMAL POLICY IN DUAL-V?

In the above derivations for `dual-Q` and `dual-V` we leveraged the fact that the closed form solution for optimizing Eq. (16) w.r.t $d$ is known. The value of $d^*$ for which Eq. (32) is maximized can be found by setting the gradient to zero (stationary point) leading to:

$$\frac{d^*(s,a)}{d^O(s,a)} = \max\left(0, (f')^{-1}\left(\frac{y(s,a)}{\alpha}\right)\right) \tag{49}$$

This ratio can be utilized in two different ways to recover the optimal policy:

**Method 1: Maximum likelihood on expert visitation distribution**

Policy learning can be written as maximizing the likelihood of optimal actions under the optimal state-action visitation:

$$\max \mathbb{E}_{s,a\sim d^*}[\log \pi_\theta(a|s)] \tag{50}$$

Using importance sampling we can rewrite the optimization above in a form suitable for optimization:

$$\max_\theta \mathbb{E}_{s,a\sim d^O}\left[\frac{d^*(s,a)}{d^O(s,a)}\log \pi_\theta(a|s)\right] = \max_\theta \mathbb{E}_{s,a\sim d^O}[w^*(s,a)\log \pi_\theta(a|s)] \tag{51}$$

This way of policy learning is similar to weighted behavior cloning or advantage-weighted regression, but suffers from the issue that policy is not optimized at state-actions where the offline dataset $d^O$ has no coverage but $d^* > 0$.

**Method 2: Reverse KL matching on offline data distribution (Information Projection)**

To allow the policy to be optimized at all that states in the offline dataset + actions outside the dataset we consider an alternate objective:

$$\min_\theta D_{\mathrm{KL}}(d^O(s)\pi_\theta(a|s) \,||\, d^O(s)\pi^*(a|s)) \tag{52}$$

The objective can be expanded as follows:

$$\min_\theta D_{\mathrm{KL}}(d^O(s)\pi_\theta(a|s) \,||\, d^O(s)\pi^*(a|s)) \tag{53}$$

$$= \min_\theta \mathbb{E}_{s\sim d^O(s),a\sim\pi_\theta}\left[\log\frac{\pi_\theta(a|s)}{\pi^*(a|s)}\right] \tag{54}$$

$$= \min_\theta \mathbb{E}_{s\sim d^O(s),a\sim\pi_\theta}\left[\log\frac{\pi_\theta(a|s)d^*(s)d^O(s)\pi^o(a|s)}{\pi^*(a|s)d^*(s)d^O(s)\pi^o(a|s)}\right] \tag{55}$$

$$= \min_\theta \mathbb{E}_{s\sim d^O(s),a\sim\pi_\theta}\left[\log\frac{\pi_\theta(a|s)}{\pi^o(a|s)} - \log(w^*(s,a)) + \log\frac{d^*(s)}{d^O(s)}\right] \tag{56}$$

$$= \min_\theta \mathbb{E}_{s\sim d^O(s),a\sim\pi_\theta}[\log(\pi_\theta(a|s)) - \log(\pi^o(a|s)) - \log(w^*(s,a))] \tag{57}$$

This method recovers the optimal policy at the states present in the dataset but has the added complexity of learning another policy $\pi^o(a|s)$. One way of obtaining $\pi^o(a|s)$ is by behavior cloning the replay buffer.

### B.1.7   SEMI-GRADIENT AND FULL-GRADIENT

RL algorithms often learn via Bellman backups which minimize an error of the form $L(\theta) = \mathbb{E}_{(s,a,s')\sim\mathcal{D}}[(Q_\theta(s,a) - (r(s,a) + \gamma\mathbb{E}_{a'\sim\pi}[Q_\theta(s',a')])^2]$. Full stochastic gradient differentiates through the entire objective, whereas semi-gradient methods do not differentiate through the $Q_\theta(s',a')$ term in the bootstrapping target and is generally implemented through a stop-gradient operator. Note that the semi-gradient update still changes the value of the bootstrapping target when used at the next iteration as $\theta$ is updated. Semi-gradient optimization is a common choice in deep RL and often enables significantly faster learning [Sutton and Barto, 2018].

## C  A UNIFIED PERSPECTIVE ON RL AND IL ALGORITHMS THROUGH DUALITY

Figure 4 shows an alternate viewpoint on the landscape of dualRL methods and highlights the gaps in algorithms that we are able to successfully address in this work. Now we discuss in detail how recent algorithms can be unified via duality viewpoint for both RL and IL.

Figure 4: We show that a number of prior methods can be understood as a special case of the dual RL framework. Based on this framework, we also propose new methods addressing the shortcomings of previous works (boxed in green).

### C.1  DUAL CONNECTIONS TO REINFORCEMENT LEARNING

We begin by showing reducing popular offline RL class of methods: pessimistic value learning (CQL [Kumar et al., 2020], ATAC [Cheng et al., 2022]) and implicit policy improvement (XQL [Garg et al., 2021]) to the `dual-Q` and `dual-V` framework respectively. Then, we show how the `dual-V` framework under a semi-gradient update rule leads to a family of offline RL algorithms that do not sample OOD actions.

**Proposition 4.** *CQL is an instance of* `dual-Q` *under the semi-gradient update rule, where the f-divergence is the Pearson $\chi^2$ divergence, and $d^O$ is the offline visitation distribution.*

*Proof.* We show that CQL [Kumar et al., 2020] and ATAC [Cheng et al., 2022], popular offline RL methods are a special case of `dual-Q` for offline RL. Consider the $\chi^2$ $f$-divergence with the generator function $f = (t-1)^2$. The dual function $f^*$ is given by $f^* = (\frac{t^2}{4} + t)$. With this $f$-divergence the `dual-Q` optimization can be simplified as:

$$\frac{(1-\gamma)}{\alpha}\mathbb{E}_{d_0,\pi(a|s)}[Q(s,a)] + \mathbb{E}_{s,a\sim d^O}\left[\frac{y(s,a,r,s')^2}{4\alpha^2} + \frac{y(s,a,r,s')}{\alpha}\right] \tag{58}$$

$$= \frac{(1-\gamma)}{\alpha}\mathbb{E}_{d_0,\pi(a|s)}[Q(s,a)] + \mathbb{E}_{s,a\sim d^O}\left[\frac{y(s,a,r,s')}{\alpha}\right] + \mathbb{E}_{s,a\sim d^O}\left[\frac{y(s,a,r,s')^2}{4\alpha^2}\right] \tag{59}$$

Let's simplify the first two terms:

$$\frac{1}{\alpha}\left[(1-\gamma)\mathbb{E}_{d_0,\pi(a|s)}[Q(s,a)] + \mathbb{E}_{s,a\sim d^O}\left[r(s,a) + \gamma\sum_{s',a'}p(s'|s,a)\pi(a'|s')Q(s',a') - Q(s,a)\right]\right] \tag{60}$$

$$= \frac{1}{\alpha}\left[(1-\gamma)\mathbb{E}_{d_0,\pi(a|s)}[Q(s,a)] + \mathbb{E}_{s,a\sim d^O}\left[\gamma\sum_{s',a'}p(s'|s,a)\pi(a'|s')Q(s',a')\right] - \mathbb{E}_{s,a\sim d^O}[Q(s,a)] + \cancel{\mathbb{E}_{s,a\sim d^O}[r(s,a)]}\right] \tag{61}$$

$$= \frac{1}{\alpha}\left[(1-\gamma)\sum_{s,a}d_0(s)\pi(a|s)Q(s,a) + \gamma\sum_{s,a}d^O(s,a)\sum_{s'}p(s'|s,a)\pi(a'|s')Q(s',a') - \mathbb{E}_{s,a\sim d^O}[Q(s,a)]\right] \tag{62}$$

$$= \frac{1}{\alpha} \left[ (1-\gamma) \sum_{s,a} d_0(s)\pi(a|s)Q(s,a) + \gamma \langle d^O, P^\pi Q \rangle - \mathbb{E}_{s,a \sim d^O}[Q(s,a)] \right] \tag{63}$$

$$= \frac{1}{\alpha} \left[ (1-\gamma) \sum_{s,a} d_0(s)\pi(a|s)Q(s,a) + \gamma \langle P_*^\pi d^O, Q \rangle - \mathbb{E}_{s,a \sim d^O}[Q(s,a)] \right] \tag{64}$$

$$= \frac{1}{\alpha} \left[ (1-\gamma) \sum_{s,a} d_0(s)\pi(a|s)Q(s,a) + \gamma \sum_{s,a} \pi(a|s)Q(s,a) \sum_{s',a'} p(s|s',a')d(s',a') - \mathbb{E}_{s,a \sim d^O}[Q(s,a)] \right] \tag{65}$$

$$= \frac{1}{\alpha} \left[ \sum_{s,a} (d_0(s) + \gamma \sum_{s'a,'} p(s|s',a')d(s',a'))\pi(a|s)Q(s,a) - \mathbb{E}_{s,a \sim d^O}[Q(s,a)] + \mathbb{E}_{s,a \sim d^O}[r(s,a)] \right] \tag{66}$$

$$= \frac{1}{\alpha} \left[ \sum_{s,a} d^O(s)\pi(a|s)Q(s,a) - \mathbb{E}_{s,a \sim d^O}[Q(s,a)] + \mathbb{E}_{s,a \sim d^O}[r(s,a)] \right] \tag{67}$$

$$= \frac{1}{\alpha} \left[ \mathbb{E}_{s \sim d^O, a \sim \pi}[Q(s,a)] - \mathbb{E}_{s,a \sim d^O}[Q(s,a)] \right] \tag{68}$$

where $P^\pi$ denotes the policy transition operator, $P_*^\pi$ denotes the adjoint policy transition operator. Removing constant terms (Eq. (61)) with respect to optimization variables we end up with the following form for `dual-Q`:

$$\frac{1}{\alpha} \left[ \underbrace{\mathbb{E}_{s \sim d^O, a \sim \pi}[Q(s,a)]}_{\text{reduce Q at OOD actions}} - \underbrace{\mathbb{E}_{s,a \sim d^O}[Q(s,a)]}_{\text{increase Q at in-distribution actions}} \right] + \underbrace{\mathbb{E}_{s,a \sim d^O} \left[ \frac{y(s,a,r,s')^2}{4\alpha^2} \right]}_{\text{minimize Bellman Error}} \tag{69}$$

Hence the `dual-Q` optimization reduces to:

$$\max_\pi \min_Q \alpha \left[ \mathbb{E}_{s \sim d^O, a \sim \pi}[Q(s,a)] - \mathbb{E}_{s,a \sim d^O}[Q(s,a)] \right] + \mathbb{E}_{s,a \sim d^O} \left[ \frac{y(s,a,r,s')^2}{4} \right] \tag{70}$$

Our proposition assumes semi-gradient update, i.e the gradients are not backpropagated through the bootstrapping target $Q(s',\pi(s'))$ when updating $\pi$ or $Q$. The bootstrapping target is regularly updated with the most recent parameters. Thus, maximization with respect to policy just amounts to maximizing the first term $\mathbb{E}_{s \sim d^O, a \sim \pi}[Q(s,a)]$. This update equation matches the unregularized CQL objective (Equation 3 in [Kumar et al., 2020]) and the ATAC objective (Equation 1 in [Cheng et al., 2022] when $\beta = 0.25$). One of they key differences between CQL and ATAC is the use of optimization strategy – CQL uses Gradient Descent Ascent whereas ATAC uses a Stackelberg formulation. □

**Proposition 2.** *XQL is an instance of* `dual-V` *under the semi-gradient update rule, where the $f$-divergence is the reverse Kullback-Liebler divergence, and $d^O$ is the offline visitation distribution.*

*Proof.* We show that the Extreme Q-Learning [Garg et al., 2023] framework for offline and online RL is a special case of the dual framework, specifically the `dual-V` using the semi-gradient update rule.

Consider setting the $f$-divergence to be the KL divergence in the `dual-V` framework, the regularization distribution and the initial state distribution to be the replay buffer distribution ($d^O = d^R$ and $d_0 = d^R$). The conjugate of the generating function for KL divergence is given by $f^*(t) = e^{t-1}$.

$$\min_{V(s)} (1-\gamma)\mathbb{E}_{d_0(s)}[V(s)] + \mathbb{E}_{s,a \sim d^R} \left[ f^* \left( \left[ r(s,a) + \gamma \sum_{s'} p(s'|s,a)V(s') - V(s) \right]/\alpha \right) \right] \tag{71}$$

$$\min_{V(s)}(1-\gamma)\mathbb{E}_{d_0(s)}[V(s)] + \mathbb{E}_{s,a\sim d^S}\left[\exp\left(\left(\left[r(s,a)+\gamma\sum_{s'}p(s'|s,a)V(s')-V(s)\right]\right)/\alpha-1\right)\right]$$ (72)

A popular approach for stable optimization in temporal difference learning is the semi-gradient update rule which has been studied in previous works [Sutton and Barto, 2018]. In this update strategy, we fix the targets for the temporal difference backup. The target in the above optimization is given by:

$$\bar{Q}(s,a) = r(s,a) + \gamma\sum_{s'}p(s'|s,a)V(s')$$ (73)

The update equation for V is now given by:

$$\min_{V(s)}(1-\gamma)\mathbb{E}_{d_0(s)}[V(s)] + \mathbb{E}_{s,a\sim d^R}\left[\exp\left(\left(\left[\bar{Q}(s,a)-V(s)\right]\right)/\alpha-1\right)\right]$$ (74)

where hat denotes the `stop-gradient` operation. We approximate this target by using mean-squared regression with the single sample unbiased estimate as follows:

$$\min_{Q}\mathbb{E}_{s,a,s'\sim d^R}\left[(Q(s,a)-(r(s,a)+V(s')))^2\right]$$ (75)

The procedure (alternating Eq. (74) and Eq. (75)) is now equivalent to the Extreme-Q learning and is a special case of the `dual-V` framework. □

### C.1.1 $f$-DVL: A FAMILY OF IMPLICIT POLICY IMPROVEMENT ALGORITHMS FOR RL

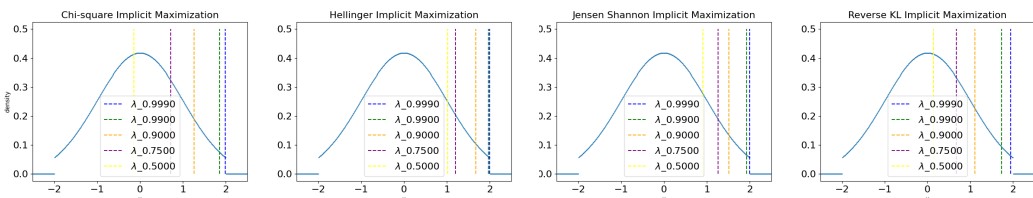

Figure 5: Illustration of a family of implicit maximizers corresponding to different $f$-divergences. The underlying data distribution is a truncated Gaussian `TN` with mean 0, variance 1 and a truncation range $(-2, 2)$. We sample 10000 data points from `TN` and compute the solution $v_\lambda$ of Problem (15). As $\lambda \to 1$, the solution $v_\lambda$ becomes a more accurate estimation for the supremum of the random variable $x$.

**Proposition 3.** *Let $x$ be a real-valued random variable such that $\Pr(x > x^*) = 0$. Let $v_\lambda$ be the solution of Problem (15). It holds that $v_{\lambda_1} \leqslant v_{\lambda_2}$, $\forall 0 < \lambda_1 < \lambda_2 < 1$. Further, $\lim_{\lambda\to 1} v_\lambda = x^*$.*

*Proof.* The behavior of dual-V (Equation 5) when $f$ is reverse KL to serve as an implicit maximizer was established in [Garg et al., 2023]. In this section we consider other divergences from Table 4 under the rewriting of dual-V in terms of temperature $\lambda$ and a surrogate extention for $f_p^*$ (defined below). We analyze the behavior for the following optimization of interest.

$$\min_{v}(1-\lambda)\mathbb{E}_{x\sim D}[v] + \lambda\mathbb{E}_{x\sim D}\left[\tilde{f}_p^*(x-v)\right]$$ (76)

$f_p^*(t)$ is given by (using the definition in Eq. (47):

$$f_p^*(t) = -f\left(\max(f'^{-1}(t),0)\right) + t\max\left(f'^{-1}(t),0\right)$$ (77)

Accordingly, the function $f_p^*$ admits two different behaviors given by:

$$f_p^* = \begin{cases} -f(f'^{-1}(t)) + tf'^{-1}(t) = f^*(t), & \text{if } f'^{-1}(t) > 0 \\ -f(0), & \text{otherwise} \end{cases}$$ (78)

where $f^*$ is the convex conjugate of $f$-divergence. We consider all $f$-divergences for which $f^*$ is strictly increasing in $\mathbb{R}^+$, and note that TV divergence will need special treatment as $(f')^{-1}$ is not well defined. Some properties of note are $f'$ and $(f')^{-1}$ is non-decreasing, $f(0^+) \geqslant 0$ and $f^*(x) \geqslant 0 \ \forall x \geqslant 0$ for the divergences we consider. A key limitation of formulating an optimization objective with forms of $f$-divergences is their domain restriction of $\mathbb{R}^+$. First, we note as a result of restriction of $f$ to $\mathbb{R}^+$ that $f' : \mathbb{R}^+ \to [l, \infty)$ for some $l \in \mathbb{R}$ and as a consequence $(f')^{-1} : [l, \infty) \to \mathbb{R}^+$. Since our objective function depends on $(f')^{-1}(x)$ being well defined

on $\mathbb{R}$, We consider an extension that preserves the non-decreasing property of $(f')^{-1}$ such that $(f')^{-1}(x) = 0$ for $x \in (-\infty, l]$. We also know from above that for $x >= l$ , $(f')^{-1}(x) >= 0$ as $(f')^{-1}$ is non-decreasing. We define the surrogate $\bar{f}_p^*$ to be a particular extension for $f_p^*$ with $l = 0$. Similar extensions can be found in prior work [Picard-Weibel and Guedj, 2022a, Goldfeld].

We analyze the second term in Eq. (76). It can be expanded as follows:

$$\lambda \int_{x:(f')^{-1}(x-v)>0} p(x)f^*(x-v)dx - \lambda \int_{x:(f')^{-1}(x-v)\leqslant 0} f(0)p(x)dx \tag{79}$$

From the properties of $f$, we use the fact that $(f')^{-1}(x-v) > 0$ when $x - v > 0$ or equivalently $x > v$.

$$\lambda \int_{x>v} p(x)f^*(x-v)dx - \lambda \int_{x\leqslant v} f(0)p(x)dx \tag{80}$$

The first term in the above equation decreases monotonically and the second term increases monotonically (thus the combined terms decrease) as $v$ increases until $v = x^*$ (supremum of the support of the distribution) after which the equation assumes a constant value of $-\lambda f(0)$.

Going back to our original optimization in Eq. (76), the first term decreases monotonically with $v$. As $\lambda \to 1$, the minimization of the second term takes precedence, with increasing $v$ until saturation ($v = x^*$). We can go further to characterize the effect of $\lambda$ on solution $v_\lambda$ of the equation. The solution of the optimization can be written in closed form as (using stationarity):

$$\frac{(1-\lambda)}{\lambda} = \mathbb{E}_{x \sim D}\left[f_p^{*\prime}(x-v)\right] \tag{81}$$

Using the fact that $f_p^{*\prime}$ is non-decreasing, we can show that the right-hand term in the equation above increases/stays the same as $v$ decreases. This in turn implies that for all $\lambda_1, \lambda_2$ such that $\lambda_1 \leqslant \lambda_2$ we have that $v_{\lambda_1} \leqslant v_{\lambda_2}$ .

TV divergence require a special treatment as $(f')^{-1}$ is not defined. We construct $f_p^*$ by noting that $f^*$ for TV exists even if $(f')^{-1}$ does not. A concise proof can be found in Example 8.1 from Goldfeld. Thus, for TV we consider a smooth extension in $\mathbb{R}^+$ by using $f^*(x) = x$. For Squared Hellinger and RKL, $f^*$ is discontinuous. The problem can be addressed by considering random variable $x \sim D$ upper bounded by 1 and $\ln 2$ for Hellinger and RKL respectively. This can be ensured by rescaling rewards so that the maximum reward is $1 - \gamma$ and $(1 - \gamma)\ln(2)$ for hellinger and RKL respectively. Appendix F.3 outlines the derivation of surrogate implicit maximizers that we use in practice.

$\square$

### C.1.2 Connections of CQL to AlgaeDICE and XQL to OptiDICE

Kumar et al. [2020] shows that CQL outperforms a family of behavior-regularized offline RL methods [Fujimoto et al., 2018, Wu et al., 2019, Nair et al., 2020], which solve different forms of `primal-Q` using approximate dynamic programming. The above result indicates that CQL's better performance is likely due to the choice of $f$-divergence and more amenable optimization afforded by the dual formulation. Moreover, the same `dual-Q` formulation has been previously studied for online RL in AlgaeDICE [Nachum et al., 2019], and proposition 4 suggests that CQL is an offline version of AlgaeDICE.

We also highlight that the full-gradient variant of the `dual-V` framework for offline RL has been studied extensively in OptiDICE [Lee et al., 2021] and proposition 2 highlights that XQL is a special case OptiDICE with a semi-gradient update rule.

### C.2 Dual Connections to Imitation Learning

This section outlines the reduction of a number of algorithms for Imitation Learning to the dual framework. Most prior methods can either take into account expert-only data for imitation whereas the other methods which do imitation from arbitrary offline data are limited by their assumptions and the form of $f$-divergence they optimize for. We walk through explaining how prior methods can be derived through the unified framework and also why they are limited.

### C.2.1 Offline imitation learning with expert data only

We saw in Section 4.1, how using the `dual-Q` framework directly led to a reduction of IQ-Learn [Garg et al., 2021] as part of the dual framework. This was accomplished by simple

setting the reward function to be 0 uniformly and setting the regularization distribution to the expert. Garg et al. [2021] uses this method in the online imitation learning setting as well by incorporating the replay data as additional regularization which we suggest is unprincipled, also pointed out by others [Al-Hafez et al., 2023] (as only expert data samples can be leveraged in the above optimization) and provide a fix in the Section 5. In this section, we show how the same approach can directly lead to another method for learning to imitate from expert-only data avoiding the alternating min-max optimization of IQ-Learn.

**IV-Learn: A new method for offline imitation learning:** Analogous to `dual-Q` (offline imitation), we can leverage the `dual-V` (offline imitation) setting which avoids the min-max optimization given by:

`IV-Learn` or `dual-V` (offline imitation from expert-only data):

$$\min_{V(s)}(1-\gamma)\mathbb{E}_{d_0(s)}[V(s)] + \mathbb{E}_{s,a\sim d^E}[f^*\left(\left[\mathcal{T}_0 V(s,a) - V(s))\right]/\alpha\right)] \tag{82}$$

We propose `dual-V` (offline imitation) to be a new method arising out of this framework which we leave for future exploration. This work primarily focuses on imitation learning from general off-policy data.

**Proofs for this section:**

**Corollary 1.** *IBC [Florence et al., 2022] is an instance of* `dual-Q` *using the full-gradient update rule, where* $r(s,a) = 0 \ \ \forall s \in \mathcal{S}, a \in \mathcal{A},\ d^O = d^E$, *and the* $f$-*divergence is the total variation distance.*

Eq. (6) suggests that intuitively IQ-Learn trains an energy-based model in the form of Q where it pushes down the Q-values for actions predicted by current policy and pushes up the Q-values at the expert state-action pairs. This becomes more clear when the divergence $f$ is chosen to be Total-Variation ($f^* = \mathbb{I}$), IQ-Learn for Total-Variation divergence reduces to:

$$(1-\gamma)\mathbb{E}_{d_0(s),\pi(a|s)}[Q(s,a)] + \mathbb{E}_{s,a\sim d^E}\left[\gamma\sum_{s',a'}p(s'|s,a)\pi(a'|s')Q(s',a') - Q(s,a)\right] \tag{83}$$

$$= \left[(1-\gamma)\mathbb{E}_{d_0(s),\pi(a|s)}[Q(s,a)] + \mathbb{E}_{s,a\sim d^E}\left[\gamma\sum_{s',a'}p(s'|s,a)\pi(a'|s')Q(s',a')\right]\right]$$
$$- \mathbb{E}_{s,a\sim d^E}[Q(s,a)] \tag{84}$$

First, we simplify the initial two terms:

$$(1-\gamma)\mathbb{E}_{d_0(s),\pi(a|s)}[Q(s,a)] + \mathbb{E}_{s,a\sim d^E}\left[\gamma\sum_{s'}p(s'|s,a)\pi(a'|s')Q(s',a')\right] \tag{85}$$

$$= (1-\gamma)\sum_{s,a}d_0(s)\pi(a|s)Q(s,a) + \gamma\sum_{s,a}d^E(s,a)\sum_{s',a'}p(s'|s,a)\pi(a'|s')Q(s',a') \tag{86}$$

$$= (1-\gamma)\sum_{s,a}d_0(s)\pi(a|s)Q(s,a) + \gamma\sum_{s',a'}\sum_{s,a}d^E(s,a)p(s'|s,a)\pi(a'|s')Q(s',a') \tag{87}$$

$$= (1-\gamma)\sum_{s,a}d_0(s)\pi(a|s)Q(s,a) + \gamma\sum_{s',a'}\pi(a'|s')Q(s',a')(\sum_{s,a}d^E(s,a)p(s'|s,a)) \tag{88}$$

$$= (1-\gamma)\sum_{s,a}d_0(s)\pi(a|s)Q(s,a) + \gamma\sum_{s',a'}\pi(a'|s')Q(s',a')(\sum_{s,a}d^E(s,a)p(s'|s,a)) \tag{89}$$

$$= (1-\gamma) \sum_{s,a} d_0(s)\pi(a|s)Q(s,a) + \gamma \sum_{s,a} \pi(a|s)Q(s,a)(\sum_{s',a'} d^E(s',a')p(s|s',a')) \quad (90)$$

$$= \sum_{s,a} (1-\gamma)d_0(s)\pi(a|s)Q(s,a) + \pi(a|s)Q(s,a)(\sum_{s',a'} d^E(s',a')p(s|s',a')) \quad (91)$$

$$= \sum_{s,a} \pi(a|s)Q(s,a)\left[(1-\gamma)d_0(s) + \gamma \sum_{s',a'} d^E(s',a')p(s|s',a')\right] \quad (92)$$

$$= \sum_{s,a} \pi(a|s)Q(s,a)d^E(s) \quad (93)$$

where the last step is due to the steady state property of the MDP (Bellman flow constraint).

Therefore IQ-Learn/`dual-Q` for offline imitation (in the special case of TV divergence) simplifies to (from Eq. (84)):

$$\left[(1-\gamma)\mathbb{E}_{d_0(s),\pi(a|s)}[Q(s,a)] + \mathbb{E}_{s,a\sim d^E}\left[\gamma \sum_{s',a'} p(s'|s,a)\pi(a'|s')Q(s',a')\right]\right] - \mathbb{E}_{s,a\sim d^E}[Q(s,a)]$$
$$(94)$$

$$= \min_{Q} \mathbb{E}_{d_E(s),\pi(a|s)}[Q(s,a)] - \mathbb{E}_{s,a\sim d^E}[Q(s,a)] \quad (95)$$

The update gradient w.r.t for the above optimization matches the gradient update of infoNCE objective in Implicit Behavior Cloning [Florence et al., 2022] with $Q$ as the energy-based model.

### C.3  OFF-POLICY IMITATION LEARNING (UNDER COVERAGE ASSUMPTION)

Directly utilizing the dual-RL framework for imitation has its limitation as we see in the previous section – we cannot leverage off-policy suboptimal data. We first show that it is easy to see why choosing the $f$-divergence to reverse KL makes it possible to get an off-policy objective for imitation learning in the dual framework. We start with the `primal-Q` for imitation learning under the reverse KL-divergence regularization ($r(s,a) = 0$ and $d^O = d^E$):

$$\max_{d(s,a)\geqslant 0,\pi(a|s)} -D_{\mathrm{KL}}(d(s,a) \,||\, d^E(s,a))$$

$$\text{s.t } d(s,a) = (1-\gamma)\rho_0(s).\pi(a|s) + \gamma\pi(a|s)\sum_{s',a'} d(s',a')p(s|s',a'). \quad (96)$$

*Under the assumption that the suboptimal data visitation (denoted by $d^S$) covers the expert visitation ($d^S > 0$ wherever $d^E > 0$)* [Ma et al., 2022], which we refer to as the **coverage assumption**, the reverse KL divergence can be expanded as follows:

$$D_{\mathrm{KL}}(d(s,a) \,||\, d^E(s,a)) = \mathbb{E}_{s,a\sim d(s,a)}\left[\log \frac{d(s,a)}{d^E(s,a)}\right] = \mathbb{E}_{s,a\sim d(s,a)}\left[\log \frac{d(s,a)}{d^E(s,a)}\frac{d^S(s,a)}{d^S(s,a)}\right]$$
$$(97)$$

$$= \mathbb{E}_{s,a\sim d(s,a)}\left[\log \frac{d(s,a)}{d^S(s,a)} + \log \frac{d^S(s,a)}{d^E(s,a)}\right] \quad (98)$$

$$= \mathbb{E}_{s,a\sim d(s,a)}\left[\log \frac{d^S(s,a)}{d^E(s,a)}\right] + D_{\mathrm{KL}}(d(s,a) \,||\, d^S(s,a)). \quad (99)$$

Hence the `primal-Q` can now be written as:

$$\max_{d(s,a)\geqslant 0,\pi(a|s)} \mathbb{E}_{s,a\sim d(s,a)}\left[-\log \frac{d^S(s,a)}{d^E(s,a)}\right] - D_{\mathrm{KL}}(d(s,a) \,||\, d^S(s,a)) \quad (100)$$

$$\text{s.t } d(s,a) = (1-\gamma)\rho_0(s).\pi(a|s) + \gamma \sum_{s',a'} d(s',a')p(s|s',a')\pi(a|s). \quad (101)$$

Now, in the optimization above the first term resembles the reward function and the second term resembles the divergence constraint with a new distribution $d^S(s,a)$ in the original regularized RL primal (Eq. (26)). Hence we can obtain respective `dual-Q` and `dual-V` in the setting for off-policy imitation learning using the reward function as $r^{\mathrm{imit}}(s,a) = -\log \frac{d^S(s,a)}{d^E(s,a)}$ and the new regularization distribution as $d^S(s,a)$. Using $\mathcal{T}_{r^{\mathrm{imit}}}^{\pi}$ and $\mathcal{T}_{r^{\mathrm{imit}}}$ to denote backup operators under a new reward function $r^{\mathrm{imit}}$, we have

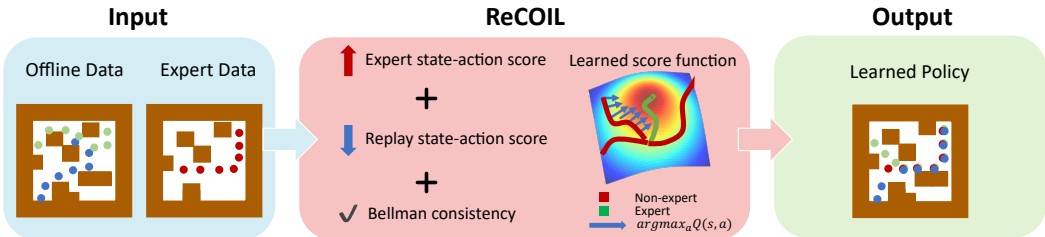

Figure 6: Recipe for ReCOIL: Learn a Bellman consistent EBM - A model which increases the score of expert transitions, and decreases the score of replay transitions while maintaining Bellman consistency throughout.

`dual-Q` for off-policy imitation (coverage assumption) :

$$\max_{\pi(a|s)} \min_{Q(s,a)} (1-\gamma)\mathbb{E}_{\rho_0(s),\pi(a|s)}[Q(s,a)] + \mathbb{E}_{s,a\sim d^S}[f^*(\mathcal{T}_{r^{\text{imit}}}^\pi Q(s,a) - Q(s,a))]. \quad (102)$$

This choice of KL divergence leads us to a reduction of another methods, OPOLO [Zhu et al., 2020] and OPIRL [Hoshino et al., 2022] for off-policy imitation learning to `dualQ` which we formalize in the proposition below:

**Proposition 5.** *OPOLO [Zhu et al., 2020] and OPIRL [Hoshino et al., 2022] are instances of* `dual-Q` *using the semi-gradient update rule, where* $r(s,a) = 0$ $\forall \mathcal{S}, \mathcal{A}$, $d^O = d^E$, *and the* $f$*-divergence set to the reverse KL divergence.*

*Proof.* Proof is sketched in the above section, ie. Eq. (102) is the update equation for OPOLO.

Analogously we have `dual-V` for off-policy imitation (coverage assumption):

$$\min_{V(s)}(1-\gamma)\mathbb{E}_{\rho_0(s)}[V(s)] + \mathbb{E}_{s,a\sim d^S}[f_p^*(\mathcal{T}_{r^{\text{imit}}}V(s,a) - V(s))]. \quad (103)$$

We note that the `dual-V` framework for off-policy imitation learning under coverage assumptions was studied in the imitation learning work SMODICE [Ma et al., 2022].

### C.4 LOGISTIC Q-LEARNING AND P²IL AS DUAL-QV METHODS

Logistic Q-learning and Proximal Point Imitation Learning (P²IL) uses a modified primal for regularized policy optimization:

$$\max_{d \geqslant 0} \mathbb{E}_{d(s,a)}[r(s,a)] - D_f(d(s,a) \,\|\, d^O(s,a)) - H(\mu(s,a)\|\mu^O(s,a))$$

$$\text{s.t } d(s,a) = (1-\gamma)d_0(s) + \pi(a|s)\gamma \sum_{s',a'} \mu(s',a')p(s|s',a'). \quad (104)$$

$$\text{and } d(s,a) = \mu(s,a) \quad (105)$$

where $H(\mu(s,a)\|\mu^O(s,a)) = \sum \mu(s,a) \log \frac{\pi_\mu(a|s)}{\pi_{\mu^O}(a|s)}$ denotes the conditional relative entropy and $\mu^O$ is another offline distribution of state-action transitions potentially the same as $d^O$. The optimization is overparameterized (setting $\mu = d$). This trick was popularized via [Mehta and Meyn, 2009] and leads to unbiased estimators and better downstream data driven algorithms. We call these two methods `dual-QV` as their dual requires estimating both $Q$ and $V$ as shown in [Viano et al., 2022, Bas-Serrano et al., 2021]

## D ReCOIL: OFF-POLICY IMITATION LEARNING WITHOUT THE COVERAGE ASSUMPTION

Understanding the limitations of existing imitation learning methods in the dual framework, we now proceed to derive our proposed method for imitation learning with arbitrary (off-policy) data. The derivation for the `dual-Q` setting is shown below. `dual-V` derivation proceeds analogously.

**Theorem 1.** *(ReCOIL objective) The* `dual-Q` *problem to the mixture distribution matching objective in Eq. 8 is given by:*

$$\max_\pi \min_Q \beta(1-\gamma)\mathbb{E}_{d_0,\pi}[Q(s,a)] + \mathbb{E}_{s,a\sim d_{mix}^{E,S}}[f^*(\mathcal{T}_0^\pi Q(s,a) - Q(s,a))] - (1-\beta)\mathbb{E}_{s,a\sim d^S}[\mathcal{T}_0^\pi Q(s,a) - Q(s,a)] \quad (9)$$

*and recovers the same optimal policy $\pi^*$ as Eq. 8 since strong duality holds from Slater's conditions.*

*Proof.* Let's define two mixture distributions that we are going to leverage to formulate the imitation learning problem: $d_{\text{mix}}^S(s,a) := \beta d(s,a) + (1-\beta)d^S(s,a)$ and $d_{\text{mix}}^{E,S}(s,a) := \beta d^E(s,a) + (1-\beta)d^S(s,a)$. $d_{\text{mix}}^S(s,a)$ is a mixture between the current agent's visitation distribution with suboptimal transition dataset obtained from a mixture of arbitrary policies and $d_{\text{mix}}^{E,S}(s,a)$ is the mixture between the expert's visitation distribution with arbitrary experience from the offline transition dataset. Minimizing the divergence between these visitation distributions still solves the imitation learning problem, i.e $d = d^E$. We again start with the new modified `primal-Q` under this mixture divergence regularization:

$$\max_{d(s,a)\geqslant 0, \pi(a|s)} -D_f(d_{\text{mix}}^S(s,a)(s,a) \mid\mid d_{\text{mix}}^{E,S}(s,a)(s,a))$$

$$\text{s.t } d(s,a) = (1-\gamma)d_0(s).\pi(a|s) + \gamma\pi(a|s)\sum_{s',a'} d(s',a')p(s|s',a').$$

Using the same algebraic machinery of duality as before (Section B.1.3) to get an unconstrained tractable optimization problem, we obtain:

$$\max_{\pi, d\geqslant 0} \min_{Q(s,a)} -D_f(d_{\text{mix}}^S(s,a) \mid\mid d_{\text{mix}}^{E,S}(s,a))$$

$$+ \beta\sum_{s,a} Q(s,a)\left((1-\gamma)d_0(s).\pi(a|s) + \gamma\sum_{s',a'} d(s',a')p(s|s',a')\pi(a|s) - d(s,a)\right) \quad (106)$$

$$= \max_{\pi, d\geqslant 0} \min_{Q(s,a)} \beta(1-\gamma)\mathbb{E}_{d_0(s),\pi(a|s)}[Q(s,a)]$$

$$+ \beta\mathbb{E}_{s,a\sim d}\left[\gamma\sum_{s'} p(s'|s,a)\pi(a'|s')Q(s',a') - Q(s,a)\right] - D_f(d_{\text{mix}}^S(s,a) \mid\mid d_{\text{mix}}^{E,S}(s,a)) \quad (107)$$

$$= \max_{\pi, d\geqslant 0} \min_{Q(s,a)} \beta(1-\gamma)\mathbb{E}_{d_0(s),\pi(a|s)}[Q(s,a)]$$

$$+ \beta\mathbb{E}_{s,a\sim d}\left[\gamma\sum_{s'} p(s'|s,a)\pi(a'|s')Q(s',a') - Q(s,a)\right]$$

$$+ (1-\beta)\mathbb{E}_{s,a\sim d^S}\left[\gamma\sum_{s'} p(s'|s,a)\pi(a'|s')Q(s',a') - Q(s,a)\right]$$

$$- (1-\beta)\mathbb{E}_{s,a\sim d^S}\left[\gamma\sum_{s'} p(s'|s,a)\pi(a'|s')Q(s',a') - Q(s,a)\right] - D_f(d_{\text{mix}}^S(s,a) \mid\mid d_{\text{mix}}^{E,S}(s,a))$$

$$(108)$$

Strong duality allows us to swap the order of $\max_d$ and $\min_Q$ in order to arrive at the following result:

---

**Imitation from Arbitrary data (dualQ)**

$$= \max_{\pi(a|s)} \min_{Q(s,a)} \max_{d\geqslant 0} \beta(1-\gamma)\mathbb{E}_{d_0(s),\pi(a|s)}[Q(s,a)]$$

$$+ \mathbb{E}_{s,a\sim d_{\text{mix}}^S(s,a)}\left[\gamma\sum_{s'} p(s'|s,a)\pi(a'|s')Q(s',a') - Q(s,a)\right] - D_f(d_{\text{mix}}^S(s,a) \mid\mid d_{\text{mix}}^{E,S}(s,a))$$

$$- (1-\beta)\mathbb{E}_{s,a\sim d^S}\left[\gamma\sum_{s'} p(s'|s,a)\pi(a'|s')Q(s',a') - Q(s,a)\right] \quad (109)$$

---

We can ignore the constraints ($d \geqslant 0$) as the primal-Q is overparameterized and the constraints uniquely determine the distribution $d$. Therefore, ignoring this constraint ($d \geqslant 0$) results in the following dual-optimization for imitation from arbitrary data.

$$\max_{\pi(a|s)} \min_{Q(s,a)} \beta(1-\gamma)\mathbb{E}_{d_0(s),\pi(a|s)}[Q(s,a)]$$

$$+ \mathbb{E}_{s,a \sim d_{\text{mix}}^{E,S}(s,a)} \left[ f^*(\gamma \sum_{s'} p(s'|s,a)\pi(a'|s')Q(s',a') - Q(s,a)) \right]$$

$$- (1-\beta)\mathbb{E}_{s,a \sim d^S} \left[ \gamma \sum_{s'} p(s'|s,a)\pi(a'|s')Q(s',a') - Q(s,a) \right] \tag{110}$$

## D.1 ReCOIL-V

A similar derivation can be done in $V$-space to obtain an analogous result for `ReCOIL-V`, although extra care has to be taken to ensure the non-negativity constraints similar to proof for in section B.1.4.

`primal-V` $\quad \max_{d(s,a) \geqslant 0} -D_f(d_{\text{mix}}^S(s,a) \,||\, d_{\text{mix}}^{E,S}(s,a))$

$$\text{s.t } \sum_{a \in \mathcal{A}} d(s,a) = (1-\gamma)d_0(s) + \gamma \sum_{(s',a') \in \mathcal{S} \times \mathcal{A}} d(s',a')p(s|s',a'), \; \forall s \in \mathcal{S}. \tag{111}$$

The dual of `primal-V` form for mixture distribution matching is given by:

`ReCOIL-V` $\min_V \beta(1-\gamma)\mathbb{E}_{s \sim d_0}[V(s)] + \mathbb{E}_{(s,a) \sim d_{\text{mix}}^{E,S}}\left[f_p^*(\mathcal{T}_0 V(s,a) - V(s))\right] - (1-\beta)\mathbb{E}_{(s,a) \sim d^S}[\mathcal{T}_0 V(s,a) - V(s)].$

$$\tag{112}$$

$\square$

## D.2 Suboptimality Bound for ReCOIL-V

**Theorem 2** (Suboptimality Bound for Offline `ReCOIL`). *Let $S^J$ denote the joint support of $d^S$ and $d^E$. Let $r(s,a) = V(s) - \mathcal{T}_0 V(s,a)$ be the pseudo-reward implied by `ReCOIL` and $R_{\max} = \max_{s,a} r(s,a)$. Let $D_\delta = \left\{d \,|\, \text{Pr}_d\left((s,a) \in S^J\right) \geqslant 1 - \delta\right\}$ be the set of visitation distributions that have $1 - \delta$ coverage of $S^J$. Let $\pi_\delta^*$ be the best policy over all policies whose visitation distribution is in $D_\delta$. Let $g(d,V) = (1-\gamma)\mathbb{E}_{d_0(s)}[V(s)] + \mathbb{E}_d[\mathcal{T}_0 V(s,a) - V(s)] - D_f(d(s,a) \,||\, d^E(s,a))$ be the imitation learning objective, and $h(V) = \max_{d \in D_\delta} g(d,V)$. Suppose that we can solve `ReCOIL` with the constraint $d \in D_\delta$, $h$ is $\kappa$-strongly convex in $V$ and $\beta \to 1$, then the output policy $\hat{\pi}$ satisfies that $J(\pi_\delta^*) - J(\hat{\pi}) \leqslant \frac{4}{1-\gamma}\sqrt{2\delta R_{\max}/\kappa}$.*

We provide a suboptimality bound by analyzing `ReCOIL-V` in this section. An important note is that we consider the setting $\beta \to 1$, which implies that we study the behavior of the optimization when $\beta$ is a number close to 1 and not exactly 1. This allows us to incorporate suboptimal data in off-policy imitation learning setting.

Recall that `ReCOIL-V` admits a `dual-V` form (112). When deriving `dual-V`, there is one step (Eq. (41)) where we assumed the importance sampling is exact, i.e.,

$$\mathbb{E}_{(s,a) \sim d}[\mathcal{T} V(s,a) - V(s)] = \mathbb{E}_{(s,a) \sim d^O}\left[\frac{d(s,a)}{d^O(s,a)}(\mathcal{T} V(s,a) - V(s))\right]. \tag{113}$$

However, this assumption does not hold in general and is not practical, because $d^O$ and $d$ might have different support. The gap between the two terms greatly affects the performance of dual RL approaches. We shall bound the approximation error introduced by importance sampling for `ReCOIL-V` in Section D.2.1, and then bound the suboptimality of the learned policy in Section D.2.2, under mild conditions. This analysis also results in the suboptimality bound of `IV-Learn` and `IQ-Learn` methods.

Let $S^J$ denote the joint support of $d^S$ and $d^E$. Let $r(s,a) = V(s) - \gamma\mathcal{T}_0 V(s,a)$ be the pseudo-reward implied by `ReCOIL` and $R_{\max} = \max_{s,a} |r(s,a)|$. Let $D_\delta = \left\{d \,|\, \text{Pr}_d\left((s,a) \in S^J\right) \geqslant 1 - \delta\right\}$ be the set of visitation distributions that have $1 - \delta$ coverage of $S^J$, where $\text{Pr}_d\left((s,a) \in S^J\right)$ is the probably that $(s,a)$ lies in $S^J$ when sampling $(s,a)$ from $d$.

We make the following assumptions for our proof:

**Assumption 1** We consider imitation learning under the constraint $d \in D_\delta$. This is similar to pessimism assumption when learning from fixed datasets in offline RL Levine et al. [2020].

**Assumption 2** The hyperparameter $\beta$ for defining $d_{\text{mix}}^S(s,a)$ and $d_{\text{mix}}^{E,S}(s,a)$ goes to 1: $\beta \to 1$.

**Assumption 3** The function $h(V)$ defined in Section D.2.2 is $\kappa$-strongly convex.

For Assumption 1, `ReCOIL-V` is able to find a policy under the visitation constraint as a result of a combination of implicit maximization, which prevents overestimation and thus choosing OOD action, and weighted behavior cloning (Advantage-weighted regression), which keeps the output policy close to the dataset policy.

### D.2.1 Approximation Error of the Imitation Learning Objective

The imitation learning problem can be written in the Lagrangian form of `primal-V` where $r(s, a) = 0$ everywhere:

$$\min_V \max_{d \in D_\delta} (1 - \gamma)\mathbb{E}_{d_0(s)}[V(s)] + \mathbb{E}_d[\mathcal{T}_0 V(s, a) - V(s)] - D_f(d(s, a) \,||\, d^E(s, a)), \quad (114)$$

where we have a constraint $d \in D_\delta$ due to Assumption 1. `ReCOIL-V` optimizes a surrogate objective of Problem (114). To derive `ReCOIL-V`, consider the corresponding `primal-V` in its Lagrangian form

$$\min_V \max_{d \in D_\delta} (1 - \gamma)\mathbb{E}_{d_0(s)}[V(s)] + \mathbb{E}_d[\mathcal{T}_0 V(s, a) - V(s)] - D_f(d_{\text{mix}}^S(s, a) \,||\, d_{\text{mix}}^{E,S}(s, a)). \quad (115)$$

Rewriting the second term, we obtain

$$\min_V \max_{d \in D_\delta} (1 - \gamma)\mathbb{E}_{d_0(s)}[V(s)] + \frac{1}{\beta}\mathbb{E}_{s,a \sim d_{\text{mix}}^S(s,a)}[\mathcal{T}_0 V(s, a) - V(s)]$$

$$- D_f(d_{\text{mix}}^S(s, a) \,||\, d_{\text{mix}}^{E,S}(s, a)) - \frac{1 - \beta}{\beta}\mathbb{E}_{d^S}[\mathcal{T}_0 V(s, a) - V(s)]. \quad (116)$$

Now we *approximate* the second term via importance sampling, which leads to

$$\min_V \max_{d \in D_\delta} (1 - \gamma)\mathbb{E}_{d_0(s)}[V(s)] + \frac{1}{\beta}\mathbb{E}_{s,a \sim d_{\text{mix}}^{E,S}(s,a)}\left[\frac{d_{\text{mix}}^S(s, a)}{d_{\text{mix}}^{E,S}(s, a)}(\mathcal{T}_0 V(s, a) - V(s))\right]$$

$$- \mathbb{E}_{d_{\text{mix}}^{E,S}(s,a)}\left[f\left(\frac{d_{\text{mix}}^S(s, a)}{d_{\text{mix}}^{E,S}(s, a)}\right)\right] - \frac{1 - \beta}{\beta}\mathbb{E}_{d^S}[\mathcal{T}_0 V(s, a) - V(s)]. \quad (117)$$

By expanding $d_{\text{mix}}^S(s, a) = \beta d(s, a) + (1 - \beta)d^S(s, a)$, we obtain

$$\min_V \max_{d \in D_\delta} (1 - \gamma)\mathbb{E}_{d_0(s)}[V(s)] + \frac{1}{\beta}\mathbb{E}_{s,a \sim d_{\text{mix}}^{E,S}(s,a)}\left[\frac{d_{\text{mix}}^S(s, a)}{d_{\text{mix}}^{E,S}(s, a)}(\mathcal{T}_0 V(s, a) - V(s))\right]$$

$$- \mathbb{E}_{d_{\text{mix}}^{E,S}(s,a)}\left[f\left(\frac{d_{\text{mix}}^S(s, a)}{d_{\text{mix}}^{E,S}(s, a)}\right)\right] - \frac{1 - \beta}{\beta}\mathbb{E}_{d^S}[\mathcal{T}_0 V(s, a) - V(s)], \quad (118)$$

This can be further simplified to

$$\min_V \max_{d \in D_\delta} (1 - \gamma)\mathbb{E}_{d_0(s)}[V(s)] + \mathbb{E}_{s,a \sim d_{\text{mix}}^{E,S}(s,a)}\left[\frac{d(s, a)}{d_{\text{mix}}^{E,S}(s, a)}(\gamma\mathcal{T}_0 V(s, a) - V(s))\right]$$

$$- \mathbb{E}_{d_{\text{mix}}^{E,S}(s,a)}\left[f\left(\frac{d_{\text{mix}}^S(s, a)}{d_{\text{mix}}^{E,S}(s, a)}\right)\right], \quad (119)$$

where we used the fact

$$\mathbb{E}_{s,a \sim d_{\text{mix}}^{E,S}(s,a)}\left[\frac{d^S(s, a)}{d_{\text{mix}}^{E,S}(s, a)}(\mathcal{T}_0 V(s, a) - V(s))\right] = \mathbb{E}_{s,a \sim d^S}[\mathcal{T}_0 V(s, a) - V(s)]$$

as the support of $d_{\text{mix}}^{E,S}(s, a)$ contains the support of $d^S$.

Let $g(d, V)$ and $\hat{g}_{\texttt{ReCOIL}}(d, V)$ be the objective functions of Problem (114) and (119). $g(d, V)$ is the original IL objective we want to solve, and $\hat{g}_{\texttt{ReCOIL}}(d, V)$ is an approximation (with importance sampling) of $g(d, V)$ used by `ReCOIL-V`. To simplify the analysis, we consider the case when mixture ratio $\beta \to 1$ (Assumption 2), so that *the approximation error of the objective function reduces to the approximation error of importance sampling.* That is,

$$|g(d, V) - \hat{g}_{\texttt{ReCOIL}}(d, V)| \to \left|\mathbb{E}_d[\mathcal{T}_0 V(s, a) - V(s)] - \mathbb{E}_{d_{\text{mix}}^{E,S}(s,a)}\left[\frac{d(s,a)}{d_{\text{mix}}^{E,S}(s,a)}(\mathcal{T}_0 V(s, a) - V(s))\right]\right|. \quad (120)$$

For any visitation distribution $d \in D_\delta$, it holds that

$$\left| \mathbb{E}_d[(\mathcal{T}_0 V(s,a) - V(s))] - \mathbb{E}_{d_{\mathrm{mix}}^{E,S}(s,a)}\left[ \frac{d(s,a)}{d_{\mathrm{mix}}^{E,S}(s,a)} (\mathcal{T}_0 V(s,a) - V(s)) \right] \right|$$

$$\leqslant \mathbb{E}_{s,a \in S^d \setminus S^J}[|\mathcal{T}_0 V(s,a) - V(s)|] \leqslant \max \delta |\mathcal{T}_0 V(s,a) - V(s)| \leqslant \delta R_{\max}, \quad (121)$$

where $S^d$ is the support of $d$, and the second inequality follows from the definition of $D_\delta$. As a consequence, we can bound the approximation error

$$\epsilon_{\mathtt{ReCOIL}} = \max_{d \in D_\delta, V} \left| g(d, V) - \lim_{\beta \to 1} \widehat{g}_{\mathtt{ReCOIL}}(d, V) \right| \leqslant \delta R_{\max}. \quad (122)$$

Similarly, one can show that for $\mathtt{IV\text{-}Learn}$, we have

$$|g(d, V) - \widehat{g}_{\mathtt{IVLearn}}(d, V)| \to \left| \mathbb{E}_d[\mathcal{T}_0 V(s,a) - V(s)] - \mathbb{E}_{s,a \sim d^E}\left[ \frac{d(s,a)}{d^E(s,a)} \right] (\mathcal{T}_0 V(s,a) - V(s)) \right|. \quad (123)$$

Let $S^E$ be the support of $d^E$. Unlike $\mathtt{ReCOIL\text{-}V}$, the objective of $\mathtt{IVLearn}$ suffers from the following worst-case estimation error

$$\left| \mathbb{E}_d[(\mathcal{T}_0 V(s,a) - V(s))] - \mathbb{E}_{d^E}\left[ \frac{d(s,a)}{d^E(s,a)} (\mathcal{T}_0 V(s,a) - V(s)) \right] \right|$$

$$\leqslant \mathbb{E}_{(s,a) \in S^d \setminus S^E}[|\mathcal{T}_0 V(s,a) - V(s)|] \leqslant \max |\mathcal{T}_0 V(s,a) - V(s)| \leqslant R_{\max}, \quad (124)$$

and consequently

$$\epsilon_{\mathtt{IVLearn}} = \max_{d \in D_\delta, V} \left| g(d, V) - \lim_{\beta \to 1} \widehat{g}_{\mathtt{IVLearn}}(d, V) \right| \leqslant R_{\max}. \quad (125)$$

We note that the same approximation error bounds hold similarly for $\mathtt{IQLearn}$ as that of $\mathtt{IVLearn}$. Thus $\mathtt{ReCOIL}$ has a smaller upper bound for the approximation error than $\mathtt{IQLearn}$ which we will see in the next sections leads to a better performance guarantee than $\mathtt{IQLearn}$.

### D.2.2    PERFORMANCE BOUND OF THE LEARNED POLICY

Recall that $\epsilon_{\mathtt{ReCOIL}}$ denotes the approximation error of the objective function by $\mathtt{ReCOIL\text{-}V}$:

$$\epsilon_{\mathtt{ReCOIL}} = \max_{d \in D_\delta, V} \left| g(d, V) - \lim_{\beta \to 1} \widehat{g}(d, V) \right|. \quad (126)$$

Let $h(V) = \max_{d \in D_\delta} g(d, V)$ and $\widehat{h}(V) = \max_{d \in D_\delta} \lim_{\beta \to 1} \widehat{g}(d, V)$. It directly follows from Eq. (126) that

$$|\widehat{h}(V) - h(V)| \leqslant 2\epsilon_{\mathtt{ReCOIL}}, \ \forall V. \quad (127)$$

We note that $\max_d g(d, V)$ (without the $d \in D_\delta$ constraint) is the standard $\mathtt{dual\text{-}V}$ form for imitation learning, but $h(V)$ here is defined as the same optimization under a constrained set $d \in D_\delta$.

Let $\widehat{V} = \arg\min_V \widehat{h}(V)$ and $V^* = \arg\min_V h(V)$. We are interested in bounding the gap $h(\widehat{V}) - h(V^*)$. It holds that

$$h(\widehat{V}) - h(V^*) = h(\widehat{V}) - \widehat{h}(\widehat{V}) + \widehat{h}(\widehat{V}) - h(V^*) \quad (128)$$

$$= h(\widehat{V}) - \widehat{h}(\widehat{V}) + \widehat{h}(\widehat{V}) - \widehat{h}(V^*) + \widehat{h}(V^*) - h(V^*) \quad (129)$$

$$\leqslant 2\epsilon_{\mathtt{ReCOIL}} + 0 + 2\epsilon_{\mathtt{ReCOIL}} \quad (130)$$

$$= 4\epsilon_{\mathtt{ReCOIL}}, \quad (131)$$

where the inequality follows from Eq. (127) and the fact $\widehat{V} = \arg\min_V \widehat{h}(V)$.

As a consequence, we have

$$4\epsilon_{\mathtt{ReCOIL}} \geqslant h(\widehat{V}) - h(V^*) \quad (132)$$

$$\geqslant h(V^*) + (V^* - \widehat{V})\nabla h(V^*) + \frac{\kappa}{2}\|V^* - \widehat{V}\|_F^2 - h(V^*) \quad (133)$$

$$= \frac{\kappa}{2}\|V^* - \widehat{V}\|_F^2, \quad (134)$$

where the second inequality comes from the fact that the function $h(V)$ is $\kappa$-strongly convex (Assumption 3) and $\nabla h(V^*) = 0$. It directly follows that

$$\|V^* - \widehat{V}\|_\infty \leqslant \|V^* - \widehat{V}\|_F \leqslant 2\sqrt{\frac{2}{\kappa}\epsilon_{\mathtt{ReCOIL}}}. \quad (135)$$

Let $\pi_\delta^*$ be the policy that acts greedily with value function $V^*$, which is an optimal policy over all policies whose visitation distribution is within $D_\delta$. Let $\widehat{\pi}$ denote the policy that acts greedily with value function $\widehat{V}$, i.e., the output policy of $\mathtt{ReCOIL\text{-}V}$. We then use the results in Singh and Yee

[1994] to bound the performance gap between $\pi_\delta^*$ and $\widehat{\pi}$:

$$J^{\pi_\delta^*} - J^{\widehat{\pi}} \leqslant \frac{4}{1-\gamma}\sqrt{\frac{2\epsilon_{\texttt{ReCOIL}}}{\kappa}} \leqslant \frac{4}{1-\gamma}\sqrt{\frac{2\delta R_{\max}}{\kappa}}. \tag{136}$$

The above results demonstrate that `ReCOIL` is able to leverage suboptimal data with an approximate in-distribution policy improvement and results in a policy close to the best policy with visitation almost in-support of the dataset.

## D.3    ReCOIL with $\chi^2$ divergence

In this section, we derive the objective for `ReCOIL` under the chosen $\chi^2$ divergence. Starting with the core ReCOIL objective for Q-update:

$$\max_\pi \min_Q \beta(1-\gamma)\mathbb{E}_{d_0,\pi}[Q(s,a)] + \mathbb{E}_{s,a\sim d_{\text{mix}}^{E,S}}[f^*(\mathcal{T}_0^\pi Q(s,a) - Q(s,a))] - (1-\beta)\mathbb{E}_{s,a\sim d^S}[\mathcal{T}_0^\pi Q(s,a) - Q(s,a)]$$
$$\tag{137}$$

Under the $\chi^2$ divergence ($f^* = x^2/4 + x$), we can simplify the Recoil objective as follows. Let $Q(s',\pi) = \mathbb{E}_{a'\sim\pi(s')}[Q(s',a')]$, then:

$$\max_\pi \min_Q \beta(1-\gamma)\mathbb{E}_{d_0,\pi}[Q(s,a)] + 0.25\mathbb{E}_{s,a\sim d_{\text{mix}}^{E,S}}[(\gamma Q(s',\pi) - Q(s,a))^2] + \beta\mathbb{E}_{s,a\sim d_E}[(\gamma Q(s',\pi) - Q(s,a))]$$
$$+ (1-\beta)\mathbb{E}_{s,a\sim d_S}[(\gamma Q(s',\pi) - Q(s,a))] - (1-\beta)\mathbb{E}_{s,a\sim d_S}[(\gamma Q(s',\pi) - Q(s,a))] \tag{138}$$

The last term of the ReCOIL objective cancels, simplifying to:

$$\max_\pi \min_Q \beta(1-\gamma)\mathbb{E}_{d_0,\pi}[Q(s,a)] + 0.25\mathbb{E}_{s,a\sim d_{\text{mix}}^{E,S}}[(\gamma Q(s',\pi) - Q(s,a))^2] + \beta\mathbb{E}_{s,a\sim d_E}[(\gamma Q(s',\pi) - Q(s,a))]$$
$$\tag{139}$$

Finally, rearranging terms we get:

$$\max_\pi \min_Q \beta(1-\gamma)\mathbb{E}_{d_0}[Q(s,\pi)] + \gamma\beta\mathbb{E}_{s\sim d_E}[(Q(s',\pi)] - \beta\mathbb{E}_{s,a\sim d_E}[Q(s,a)]) + 0.25\mathbb{E}_{s,a\sim d_{\text{mix}}^{E,S}}[(\gamma Q(s',\pi) - Q(s,a))^2]$$
$$\tag{140}$$

Equivalently:

$$\max_\pi \min_Q (1-\gamma)\mathbb{E}_{d_0}[Q(s,\pi)] + \gamma\mathbb{E}_{s\sim d_E}[(Q(s',\pi)] - \mathbb{E}_{s,a\sim d_E}[Q(s,a)]) + \frac{0.25}{\beta}\mathbb{E}_{s,a\sim d_{\text{mix}}^{E,S}}[(\gamma Q(s',\pi) - Q(s,a))^2]$$
$$\tag{141}$$

Substituting the initial distribution as the dataset distribution $d_0 = d^S$ (similar to [Kostrikov et al., 2019] and common practice in off-policy RL), and combining the first two terms which decrease $Q$ values at a mixture of dataset states(offline and expert) under the current policy we obtain the intuitive definition of ReCOIL from the paper which indicates to increase Q at expert-state actions and decrease Q at dataset states under current policy. But this can lead to unbounded Q functions. Finally, Eq. 141 in the practical algorithm (Algorithm 1) implements $\max_\pi$ by performing an implicit maximization.

## E    Taking dual-RL from offline to online setting

**Imitation Learning**  Problem (7) naturally extends to online IL, as the suboptimal data does not need to be static—it can be the replay buffer during online training. The corresponding algorithms generalize as well, since their key component is estimating the $Q^\pi$ function using *off-policy* data. It is worth noting that $d^S$ is dynamically changing for online IL. In contrast, Eq. (6) cannot be extended to online IL. Garg et al. [2021] uses IQLearn in the online setting where they add additional regularization using bellman backups on $d^S$. Our results suggest this to be unprincipled (also pointed out by Al-Hafez et al. [2023]), as only expert data samples can be leveraged in this formulation.

**Reinforcement Learning**  Again, all the above-discussed offline methods naturally extend to online settings [Kostrikov et al., 2021, Garg et al., 2023, Nakamoto et al., 2023], as their off-policy nature extends beyond the offline setup. Our analysis still holds, where the regularization distribution $d^O$ becomes the visitation distribution of the replay buffer $d^R$. It is worth noting that $d^R$ is dynamically changing over the course of training.

## F  IMPLEMENTATION AND EXPERIMENT DETAILS

### F.1  OFFLINE IL: ReCOIL ALGORITHM AND IMPLEMENTATION DETAILS

The algorithm for ReCOIL can be found in Algorithm 1. We base the ReCOIL implementation on the official implementation of XQL [Garg et al., 2023] and IQL [Kostrikov et al., 2021]. Our network architecture mimics theirs and uses the same data preprocessing techniques.

In our set of environments, we keep the same hyper-parameters (except $\lambda$ - parameter that intuitively controls the pessimism or the upper expectile of $Q$ function) across tasks - locomotion, adroit manipulation, and kitchen manipulation. For each environment, the values of $\lambda$ are searched between [2.5,5,10]. We keep a constant batch size of 256 across all environments. For all tasks we average mean returns over 10 evaluation trajectories and 7 random seeds. We add Layer Normalization [Lei Ba et al., 2016] to the value networks for all environments. Full hyper-parameters we used for experiments are given in Table 5. Although there might be better alternatives for implicit maximization, we found the implicit maximizer from [Garg et al., 2023] to be especially performant in the imitation setting. For policy update, using Advantage weighted regression, we use the temperature $\alpha$ to be 3 for MuJoCo locomotion environments and to be 0.5 for kitchen environments. The resembles prior work [Kostrikov et al., 2021].

**Numerical Stability:**  In practice a naive implementation of ReCOIL update for $Q_\phi$ in equation 11 suffers from numerical instability to learning $Q$-functions that are unbounded and since our objective maximizes $Q$-values at expert distribution, the $Q$ values can be arbitrarily large without any grounding. The equation for $Q$-update from ReCOIL is given by:

$$\mathcal{L}(\phi) = \beta(\mathbb{E}_{d^S, \pi(a|s)}[Q_\phi(s,a)] - \mathbb{E}_{d^E(s,a)}[Q_\phi(s,a)]) + \mathbb{E}_{s,a \sim d^{E,S}_{\text{mix}}(s,a)}\big[(\gamma V_\psi(s') - Q_\phi(s,a))^2\big],$$
(142)

To avoid this numerical instability we make a small modification to the objective, that upper bounds the $Q$-function regression target as follows:

$$\mathcal{L}(\phi) = \beta(\mathbb{E}_{d^S, \pi(a|s)}[Q_\phi(s,a)] - \mathbb{E}_{d^E(s,a)}\big[(Q_\phi(s,a) - Q_{max})^2\big] + \mathbb{E}_{s,a \sim d^{E,S}_{\text{mix}}(s,a)}\big[(\gamma V_\psi(s') - Q_\phi(s,a))^2\big],$$
(143)

Such modifications are inspired by [Sikchi et al., 2022a, Al-Hafez et al., 2023] which have found that bounding targets can make learning significantly more stable.

Hyperparameters for our proposed off-policy imitation learning method `ReCOIL` are shown in Table 5.

| Hyperparameter | Value |
|---|---|
| Policy learning rate | 3e-4 |
| Value learning rate | 3e-4 |
| $f$-divergence | $\chi^2$ |
| max-clip (loss clipping) | 7 |
| MLP layers | (256,256) |
| LR decay schedule | cosine |
| $Q_{max}$ | 200 |

Table 5: Hyperparameters for `ReCOIL`.

### F.2  OFFLINE IMITATION LEARNING EXPERIMENTS

**Environments:**  For the offline imitation learning experiments we focus on 10 locomotion and manipulation environments from the MuJoCo physics engine [Todorov et al., 2012]. These environments include Hopper, Walker2d, HalfCheetah, Ant, Kitchen, Pen, Door, Hammer, and Relocate. The MuJoCo environments used in this work are licensed under CC BY 4.0 and the datasets used from D4RL are also licensed under Apache 2.0.

**Suboptimal Datasets:**  For the offline imitation learning task, we utilize offline datasets consisting of environment interactions from the D4RL framework [Fu et al., 2020]. Specifically, we construct suboptimal datasets following the composition approach introduced in SMODICE [Ma et al., 2022]. The suboptimal datasets, denoted as 'random+expert', 'random+few-expert', 'medium+expert', and 'medium+few-expert' combine expert trajectories with low-quality trajectories obtained from the

"random-v2" and "medium-v2" datasets, respectively. For locomotion tasks, the 'x+expert' dataset (where x is 'random' or 'medium') contains a mixture of some number of expert trajectories ($\leqslant 200$) and $\approx 1$ million transitions from the "x" dataset. The 'x+few-expert' dataset is similar to 'x+expert,' but with only 30 expert trajectories included. For manipulation environments we consider only 30 expert trajectories mixed with the complete 'x' dataset of transitions obtained from D4RL.

**Expert Dataset:** To enable imitation learning, an offline expert dataset is required. In this work, we use 1 expert trajectory obtained from the "expert-v2" dataset for each respective environment.

**Baselines:** To benchmark and analyze the performance of our proposed methods for offline imitation learning with suboptimal data, we consider four representative baselines in this work: SMODICE [Ma et al., 2022], RCE [Eysenbach et al., 2021], ORIL [Zolna et al., 2020], and IQLearn [Garg et al., 2021]. We exclude DEMODICE [Kim et al., 2022b] from the comparison, as SMODICE has been shown to be competitive [Ma et al., 2022]. SMODICE is an imitation learning method based on the dual framework, assuming a restrictive coverage. ORIL adapts the generative adversarial imitation learning (GAIL) [Ho and Ermon, 2016] algorithm to the offline setting, employing an offline RL algorithm for policy optimization. The RCE baseline combines RCE, an online example-based RL method proposed by Eysenbach et al. [2021]. RCE also uses a recursive discriminator to test the proximity of the policy visitations to successful examples. [Eysenbach et al., 2021], with TD3-BC [Fujimoto and Gu, 2021]. Both ORIL and RCE utilize a state-action based discriminator similar to SMODICE, and TD3-BC serves as the offline RL algorithm. All the compared approaches only have access to the expert state-action trajectory.

The open-source implementations of the baselines SMODICE, RCE, and ORIL provided by the authors [Ma et al., 2022] are employed in our experiments. We use the hyperparameters provided by the authors, which are consistent with those used in the original SMODICE paper [Ma et al., 2022], for all the MuJoCo locomotion and manipulation environments.

### F.3 CALCULATION OF $f_p^*$ FOR $f$-DVL UNDER PRACTICAL CONSIDERATIONS

The main practical consideration when optimizing $f_p^*$ is that the function is not well defined in $\mathbb{R}$. We extend the domain of $f_p^*$ from a semi-closed interval $[l, \infty)$ for certain $l \in \mathbb{R}$ to the set of real numbers $\mathbb{R}$. Such extension and the behavior of $f_p^*$ is described in the proof of proposition 3, but we will discuss it in more detail in this section.

Let us start from Eq.47. Here, the domain of $f_p^*$ is the same as $f'^{-1}$. Recall that $f$ only admits a domain of $\mathbb{R}^+ = [0, \infty)$. As a consequence, the function $f'^{-1}$ has a limited domain $[l, \infty)$ for certain $l \in \mathbb{R}$ (To see this, first note that $f'$ is non-decreasing as $f$ is convex; further, since the domain of $f$ is bounded from below, the range of $f'$ is also bounded from below.). The behavior of $f_p^* : [l, \infty) \mapsto \mathbb{R}_+$ is then described as:

$$f_p^*(x) = \begin{cases} f^*(x), & \text{if } f'^{-1}(x) > 0 \\ C = -f(0), & \text{otherwise} \end{cases}$$

Next, we extend the domain of $f_p^*$ to $\mathbb{R}$. We use $\bar{f}_p^*$ to denote the extended function. A natural choice is to take Eq 78 and extend it to $\mathbb{R}$. We also note that similar extensions can be found in prior work [Goldfeld, Picard-Weibel and Guedj, 2022b].

$\chi^2$ **divergence** In practice, using the definition of $f^*$ for $\chi^2$, we use a smoother surrogate objective that still maintains the property of implicit maximization for Equation 76 and concisely write:

$$\bar{f}_p^* = \max(C, x^2/4 + x). \tag{144}$$

**Total Variation** For the special case of total variation divergence, note that the convex conjugate $f^*(y)$ exists and is given by $f^*(y) = y$ if $y \in [-\frac{1}{2}, \frac{1}{2}]$ otherwise $\infty$, even if $f'^{-1}$ does not exist. A concise proof can be found in Example 8.1 from [Goldfeld]. The reason is basically that a closed-form solution for convex conjugate does not exist as equation 45 in our paper no longer follows ($f'$ is not invertible). We recover $f_p^*$ for total-variation divergence using the definition in Eq 78 similar to $\chi^2$, as follows:

$$f_p^*(x) = \begin{cases} x, & \text{if } 0.5 > x > 0 \\ \infty, & \text{if } x > 0.5 \\ C = -f(0), & \text{otherwise} \end{cases} \tag{145}$$

In practice we use a smooth extension of $f_p^*$ for TV divergence given by $\max(-f(0), x)$

---

**Algorithm 3:** $f$-`DVL` (Under Stochastic Dynamics)

1: Initialize $Q_\phi$, $V_\theta$, and $\pi_\psi$, temperature $\alpha$, weight $\lambda$
2: Let $\mathcal{D} = (s, a, r, s')$ be data from $\pi_\mathcal{D}$ (offline) or replay buffer (online)
3: **for** $t = 1..T$ iterations **do**
4:     Train $Q_\phi$ using $\min_\phi \mathcal{L}(\phi)$:

$$\mathcal{L}(\phi) = \mathbb{E}_{s,a,s' \sim \mathcal{D}}\big[(Q_\phi(s,a) - (r(s,a) + V(s')))^2\big]$$

5:     Train $V_\theta$ using $\min_\theta \mathcal{J}(\theta)$

$$\mathcal{J}(\theta) = \begin{cases} (1-\lambda)\mathbb{E}_{s,a \sim D}[V_\theta(s)] + \lambda\mathbb{E}_{s,a \sim D}\big[\max(\bar{Q}_\phi(s,a) - V_\theta(s), 0)\big] & \text{TV} \\ (1-\lambda)\mathbb{E}_{s,a \sim D}[V_\theta(s)] + \lambda\mathbb{E}_{s,a \sim D}\big[\max((\bar{Q}_\phi(s,a) - V_\theta(s)) + 0.5(\bar{Q}_\phi(s,a) - V_\theta(s))^2, 0)\big] & \chi^2 \\ (1-\lambda)\mathbb{E}_{s,a \sim D}[V_\theta(s)] + \lambda\mathbb{E}_{s,a \sim D}\big[\exp(([\bar{Q}_\phi(s,a) - V_\theta(s)]) - 1)\big] & \text{RKL/XQL} \end{cases}$$

6:     Update $\pi_\psi$ via $\max_\psi \mathcal{M}(\psi)$:

$$\mathcal{M}(\psi) = \mathbb{E}_{s,a \sim \mathcal{D}}\big[e^{\alpha(Q_\phi(s,a) - V_\theta(s))} \log \pi_\psi(s|a)\big]. \tag{148}$$

7: **end for**

---

**Practical Choice of** $C$   While Eq 78 suggests setting $C$ dependent on the corresponding $f$-divergence, we found a single value of $C = 0$ to be a robust choice across our experiments. A comparison between using $f_p^* = \{-f(0)$ if $x < -4, x^2/4 + x$ otherwise$\}$) and $f_p^* = \max(0, x^2/4 + x)$ for $\chi^2$ divergence can be found in Table 6 below. Choosing $C = 0$ instead of $-f(0)$ led to performance improvements.

| Dataset | $f$-**DVL** ($\chi^2$, $f_p^* = \max(0, x^2/4 + x)$) | $f$-**DVL (TV)** | $f$-**DVL** ($\chi^2$, $f_p^* = \{-f(0)$ if $x < -4, x^2/4 + x$ **otherwise**$\}$) ) |
|---|---|---|---|
| halfcheetah-medium-v2 | **47.7** | 47.5 | 46.19 |
| hopper-medium-v2 | 63.0 | 64.1 | **78.66** |
| walker2d-medium-v2 | 80.0 | **81.5** | 76.85 |
| halfcheetah-medium-replay-v2 | 42.9 | **44.7** | 42.91 |
| hopper-medium-replay-v2 | 90.7 | **98.0** | 97.73 |
| walker2d-medium-replay-v2 | 52.1 | 68.7 | **73.5** |
| halfcheetah-medium-expert-v2 | 89.3 | **91.2** | 89.3 |
| hopper-medium-expert-v2 | **105.8** | 93.3 | 94.5 |
| walker2d-medium-expert-v2 | **110.1** | 109.6 | 106.54 |
| kitchen-complete-v0 | **67.5** | 65.71 | 67.14 |
| kitchen-partial-v0 | 58.8 | **70.0** | 48.2 |
| kitchen-mixed-v0 | **53.75** | 52.5 | 52.4 |

Table 6: The normalized return of offline RL methods on D4RL tasks. Shows comparison of setting the cutoff constant for $f_p^*$ to be $C = -f(0)$ vs $C = 0$

### F.4   Online and Offline RL: $f$-`DVL` Algorithm and implementation details

**Rewriting of `dual-V` using temperature parameter $\lambda$ instead of $\alpha$:** We found rewriting dualV using temperature parameter $\lambda$ instead of $\alpha$ to be particularly useful in reducing the number of hyperparameters to tune in order to obtain strong learning performance. We replace the temperature parameter from $\alpha$ to $\lambda$. Notice that our initial `dual-V` formulation used the temperature parameter $\alpha$ as follows:

$$\texttt{dual-V} \quad \min_V (1-\gamma)\mathbb{E}_{s \sim d_0}[V(s)] + \alpha\mathbb{E}_{(s,a) \sim d^O}\big[f_p^*\left([\mathcal{T}V(s,a) - V(s)]/\alpha\right)\big], \tag{146}$$

The temperature parameter $\alpha$ captures the tradeoff between the first term which seeks to minimize V vs the second term which seeks to maximize V and set it to the maximum value possible when taking various actions from that state onwards. Depending on different $f$ generator functions we would require tuning this parameter as it has a non-linear dependence on the entire optimization problem through the function $f$. Instead we consider a simpler objective, that we observe to empirically reduce hyperparameter tuning significantly by trading off linear between the first term and the second term using parameter $\lambda$. This modification is used in all of our experiments for RL and IL.

$$\texttt{dual-V (rewritten)} \quad \min_V (1-\lambda)\mathbb{E}_{s \sim d_0}[V(s)] + \lambda\mathbb{E}_{(s,a) \sim d^O}\big[f_p^*\left([\mathcal{T}V(s,a) - V(s)]\right)\big], \tag{147}$$

**Offline RL**: Algorithm F.4 gives the algorithm for $f$-`DVL`. This section provides additional offline RL experiments along with complete hyper-parameter and implementation details. Figure 14 shows learning curves for all the environments. $f$-`DVL` exhibits as fast convergence as XQL but avoids the numerical instability of XQL with one hyperparameter across each set of environments. We base our implementation of $f$-`DVL` off the official implementation of XQL [Garg et al., 2023] and IQL from

| Env | Lambda $\lambda$ | Batch Size | v_updates |
|---|---|---|---|
| halfcheetah-medium-v2 | 0.7 | 256 | 1 |
| hopper-medium-v2 | 0.7 | 256 | 1 |
| walker2d-medium-v2 | 0.7 | 256 | 1 |
| halfcheetah-medium-replay-v2 | 0.7 | 256 | 1 |
| hopper-medium-replay-v2 | 0.7 | 256 | 1 |
| walker2d-medium-replay-v2 | 0.7 | 256 | 1 |
| halfcheetah-medium-expert-v2 | 0.7 | 256 | 1 |
| hopper-medium-expert-v2 | 0.7 | 256 | 1 |
| walker2d-medium-expert-v2 | 0.7 | 256 | 1 |
| antmaze-umaze-v0 | 0.8 | 256 | 1 |
| antmaze-umaze-diverse-v0 | 0.8 | 256 | 1 |
| antmaze-medium-play-v0 | 0.8 | 256 | 1 |
| antmaze-medium-diverse-v0 | 0.8 | 256 | 1 |
| antmaze-large-play-v0 | 0.8 | 256 | 1 |
| antmaze-large-diverse-v0 | 0.8 | 256 | 1 |
| kitchen-complete-v0 | 0.8 | 256 | 1 |
| kitchen-partial-v0 | 0.8 | 256 | 1 |
| kitchen-mixed-v0 | 0.8 | 256 | 1 |
| pen-human-v0 | 0.8 | 256 | 1 |
| hammer-human-v0 | 0.8 | 256 | 1 |
| door-human-v0 | 0.8 | 256 | 1 |
| relocate-human-v0 | 0.8 | 256 | 1 |
| pen-cloned-v0 | 0.8 | 256 | 1 |
| hammer-cloned-v0 | 0.8 | 256 | 1 |
| door-human-v0 | 0.8 | 256 | 1 |
| relocate-human-v0 | 0.8 | 256 | 1 |

Table 7: Offline RL Hyperparameters used for $f$-DVL. Lambda $\lambda$ is the value that controls the strength of the implicit maximizer. V-updates gives the number of value updates per Q updates.

Kostrikov et al. [2021]. Our network architecture mimics theirs and uses the same data preprocessing techniques.

In our set of environments, we keep the same hyper-parameter across sets of tasks - locomotion, adroit manipulation, kitchen-manipulation, and antmaze. Contrary to XQL, we find no need to use tricks like gradient clipping to stabilize learning. For each set of environment, the values of $\lambda$ were tuned via hyper-parameter sweeps over a fixed set of values $[0.65, 0.7, 0.75, 0.8, 0.9]$. We keep a constant batch size of 256 across all environments. For MuJoCo locomotion tasks we average mean returns over 10 evaluation trajectories and 7 random seeds. For the AntMaze tasks, we average over 1000 evaluation trajectories. We add Layer Normalization [Lei Ba et al., 2016] to the value networks for all environments. For policy update, using Advantage weighted regression, we use the temperature $\alpha$ to be 3 for MuJoCo locomotion environments and to be 0.5 for kitchen environments. The resembles prior work [Kostrikov et al., 2021]. Full hyper-parameters we used for experiments are given in Table 7.

### F.5 Online RL Experiments with $f$-DVL

**Online RL**: We base the implementation of SAC on pytorch_sac and XQL [Garg et al., 2023]. Like in offline experiments, hyper-parameters were left as default except for $\lambda$, which we tuned between $[0.6, 0.7, 0.8]$ and found a single value to work best across all environments. This was in contrast to XQL's finding which required per environment different hyperparameter. Also, as opposed to XQL we required no clipping of the loss function. We test our method on 7 random seeds for each environment.

**Compute** We ran all our experiments on a machine with AMD EPYC 7J13 64-Core Processor and NVIDIA A100 with a GPU memory consumption of <1000 MB per experiment. Our offline RL and IL experiments for locomotion tasks take 10-20 min and the online IL experiments took around 5-6 hours for 1 million timesteps.

## G Additional Experimental Results

### G.1 Why Dual-RL
#### Methods are a Better Alternative to Traditional Off-Policy Algorithms

Our experimental evaluation aims to illustrate the benefits of the dual RL framework and analyze our proposed method for off-policy imitation learning. In the RL setting, we first present a case study on the failure of ADP-based methods like SAC [Haarnoja et al., 2018] to make the most when bootstrapped with additional (helpful) data. This setting is what motivates the use of off-policy

| Hyperparameter | Value |
|---|---|
| Policy updates $n_{pol}$ | 1 |
| Policy learning rate | 3e-4 |
| Value learning rate | 3e-4 |
| MLP layers | (256,256) |
| LR decay schedule | cosine |

Table 8: Common hyperparameters for $f$-`DVL`.

| Hyperparameter | Value |
|---|---|
| Batch Size | 1024 |
| Learning Rate | 0.0001 |
| Critic Freq | 1 |
| Actor Freq | 1 |
| Actor and Critic Arch | 1024, 1024 |
| Buffer Size | 1,000,000 |
| Actor Noise | Auto-tuned |
| Target Noise | – |

Table 9: Hyperparameters for SAC.

algorithms in the first place and is invaluable in domains like robotics Uchendu et al. [2022], Nair et al. [2020]. Our results validate the benefit of utilizing the dual RL framework for off-policy learning.

**The limitations of classical off-policy algorithms:** In this section, we test the sensitivity of an ADP method (SAC [Haarnoja et al., 2018]) vs dual-RL methods in the case when we initialize the replay buffer of both styles of off-policy algorithms with expert or human demonstrated trajectories. At the beginning of training, each learning agent is provided with expert or human-demonstrated trajectories for completing the task. We add 1000 transitions from this dataset to the replay buffer for the off-policy algorithm to bootstrap from. SAC is able to leverage this helpful data and shows improved performance in Hopper-v2, where the action dimension is small. As the action dimension increases, the instability of SAC becomes more apparent (see SAC+off policy data and SACfD plots in Figure 7). We hypothesize that this failure in the online RL setting is primarily due to the training instabilities caused by TD-backups resulting in overestimation in regions where the agent's current policy does not visit. In Figure 8, we observe that overestimation indeed happens in environments with larger action dimensions and these overestimations take longer to get corrected and in the process destabilize the training.

Figure 7 shows that the dual-RL method (AlgaeDICE) is able to leverage off-policy data to increase learning performance without any signs of destabilization. This can be attributed to the distribution correction estimation property of dual RL methods which updates the current policy using the corrected on-policy policy visitation [Nachum et al., 2019]. Note, that we set the temperature $\alpha$ to a low value (0.001) to disentangle the effect of pessimism which is an alternate way to avoid overestimation.

### G.2 TRAINING CURVES FOR RECOIL ON MUJOCO TASKS

We show learning curves for `ReCOIL` in Figure 9 for locomotion tasks and Figure 10 for manipulation tasks below. `ReCOIL` training curves are reasonably stable while also being performant, especially in the manipulation setting where other methods completely fail.

### G.3 DOES RECOIL ALLOW FOR BETTER ESTIMATION OF AGENT VISITATION DISTRIBUTION?

In this section, we consider the experiment of visitation estimation for a prespecified policy given expert data and suboptimal/replay data. Our experiments are tabular, so we can have an accurate estimate of the visitation of the policy by running rollouts in the MDP. We call this estimate the ground-truth policy visitation. We will estimate the accuracy of dual-RL methods to estimate visitation density by measuring MSE error against the ground truth agent/policy visitation. The property of Dual-RL methods to implicitly estimate density ratios (eg. Eq 49) has been studied

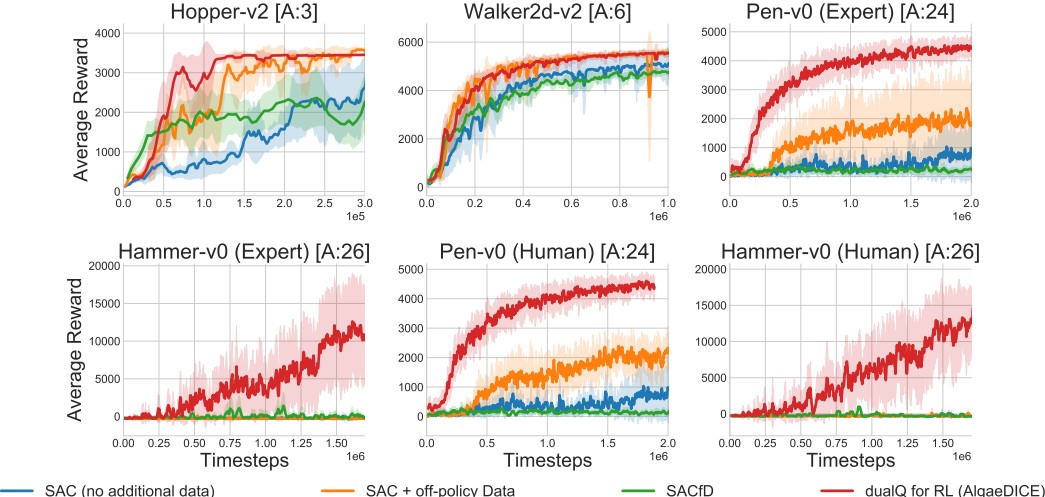

Figure 7: Despite the promise of off-policy methods, current methods based on ADP such as SAC fail when the dimension of action space, denoted by A, increases even when helpful data is added to their replay buffer. On other hand, dual-Q methods are able to leverage off-policy data to increase their learning performance

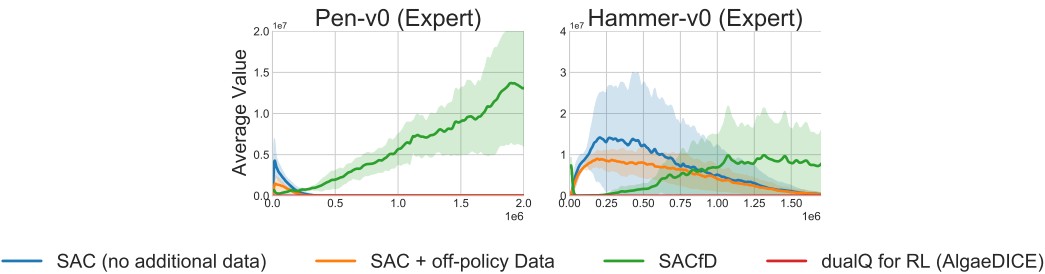

Figure 8: SAC and SACfD suffer from overestimation when off-policy data is added to the replay buffer. We hypothesize this to cause instabilities during training while dualQ has no overestimation.

before [Nachum and Dai, 2020]. We explain below how to extract visitation density ratios for any policy $\pi$ with ReCOIL. First, we discuss the setup for the experiment.

We consider two settings in our experiments: (1) a 2-timestep MDP where agent states from state $s_0$ and transitions to one of the states $\{s_1, s_2, s_3, s_4, s_5\}$ (from left to right in Figure 1a) which are absorbing. In this setting, the replay buffer perfectly covers the unknown ground truth agent visitation. (2) a 2-D gridworld (Figure 11) where the agent can move cardinally and the replay buffer distribution does not cover the unknown ground truth agent visitation. In both environments we have access to a policy at training time – the task is to estimate this policy's visitation using the offline dataset of expert and suboptimal quality. Both figures also demonstrate the ground truth policy visitation that is used to compute the mean-squared evaluation loss and test the quality of our predictions.

We consider the proposed ReCOIL method and investigate its ability to estimate distribution ratios correctly. We consider the inner optimization for ReCOIL-Q:

$$\min_{Q(s,a)} \beta(1-\gamma)\mathbb{E}_{d_0(s),\pi(a|s)}[Q(s,a)] + \mathbb{E}_{s,a \sim d_{mix}^{E,S}}\left[f^*(\gamma \sum_{s'} p(s'|s,a)\pi(a'|s')Q(s',a') - Q(s,a))\right]$$

$$- (1-\beta)\mathbb{E}_{s,a \sim d^S}\left[\gamma \sum_{s'} p(s'|s,a)\pi(a'|s')Q(s',a') - Q(s,a)\right] \tag{149}$$

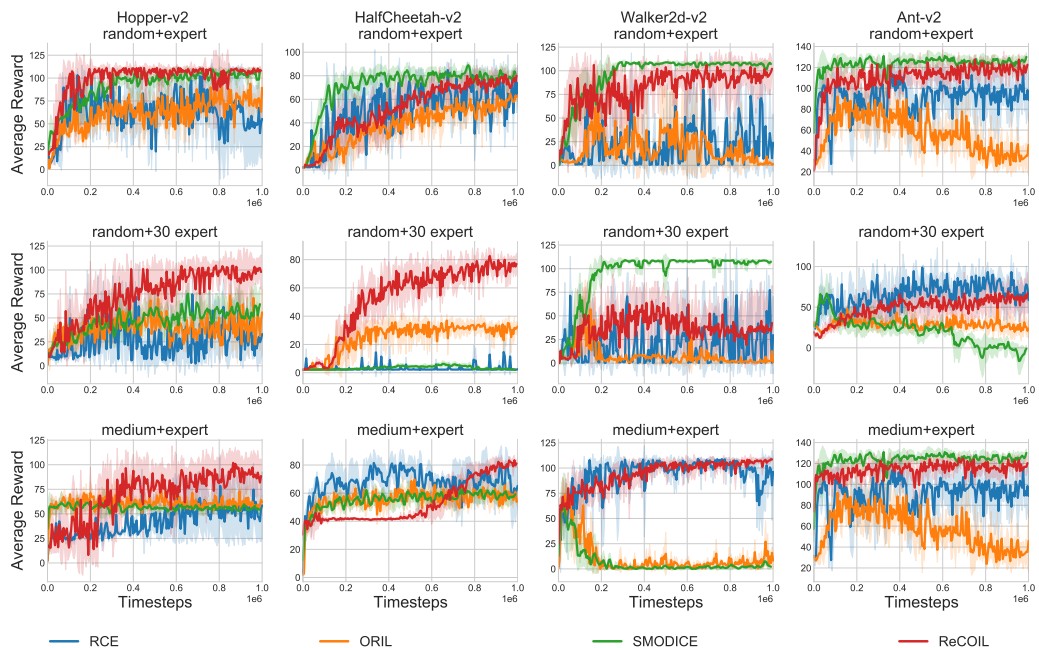

Figure 9: Learning curves for ReCOIL showing that it outperforms baselines in the setting of learning to imitate from diverse offline data. The results are averaged over 7 seeds

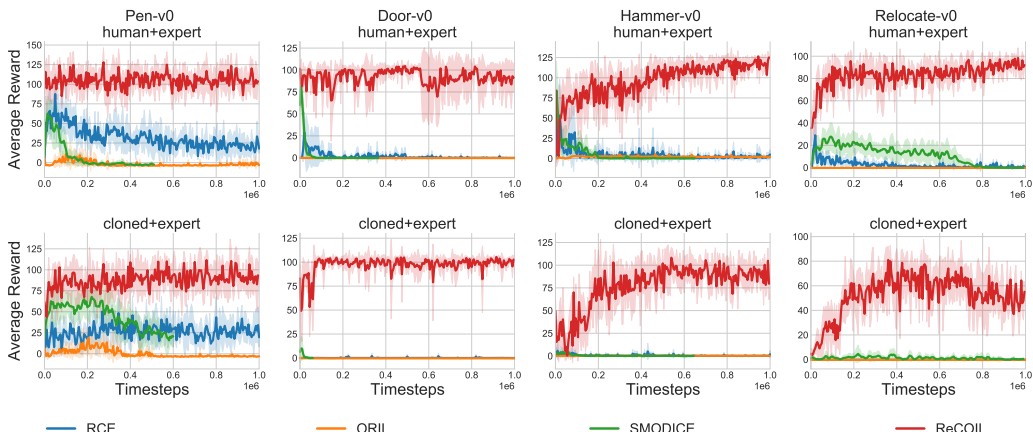

Figure 10: Learning curves for ReCOIL showing that it outperforms baselines in the setting of learning to imitate from diverse offline data. The results are averaged over 7 seeds

The following holds for the inner optimization for `ReCOIL-Q` when $Q$ is optimized:

$$f^{*'}(\gamma \sum_{s'} p(s'|s,a)\pi(a'|s')Q(s',a') - Q(s,a)) = \frac{\beta d^\pi(s,a) + (1-\beta)d^S(s,a)}{\beta d^E(s,a) + (1-\beta)d^S(s,a)} \qquad (150)$$

Thus, given the visitation distribution of the replay buffer $d^R$, expert $d^E$ and the policy $\pi$, the inner optimization implicitly learns the distribution ratio in Eq 150, allowing us to infer agent visitation $d^\pi$.

Our results (Figure 1a and 11) demonstrate that in the perfect coverage setting, `ReCOIL` is able to infer the agent policy visitation perfectly, and in the case of imperfect coverage is able to significantly outperform other methods (IQLearn and SMODICE). Note that we modify SMODICE with a $Q$ objective instead of $V$ objective to incorporate state-action expert data rather than relying on state-only expert data. IQLearn does not leverage replay data information, relying only on expert data to infer agent visitation. SMODICE's reward function $-\log \frac{d^S(s,a)}{d^E(s,a)}$, arising from its coverage

assumption is ill-defined in parts of state space where the expert has no support leading to poor downstream density ratio estimation.

Our results on the 2-D gridworld environment that demonstrate the failures of a method that either do not utilize all available suboptimal data (IQ-Learn) or relies on a coverage assumption (SMODICE). We saw that ReCOIL is able to perfectly infer the agent's visitation when the replay buffer covers agent ground truth visitation perfectly (Fig 1a) and here we see that ReCOIL is able to outperform baselines when the replay buffer has imperfect coverage over the agent's ground truth visitation (Fig 11). In this task, the agent starts at (0,0) which is the top-left corner. The agent can only move in cardinal directions with deterministic dynamics. The agent has access to two sources of off-policy data - expert visitation and replay visitation. The problem is to estimate the agent's visitation distribution given access to the agent's policy using all the available transition data. IQLearn and SMODICE predict an agent's visitation that wildly differs from Agent's ground truth visitation distribution. While ReCOIL is not perfect as the coverage of the offline data is limited, we can estimate some visitation which is qualitatively very similar to the agent's ground truth visitation.

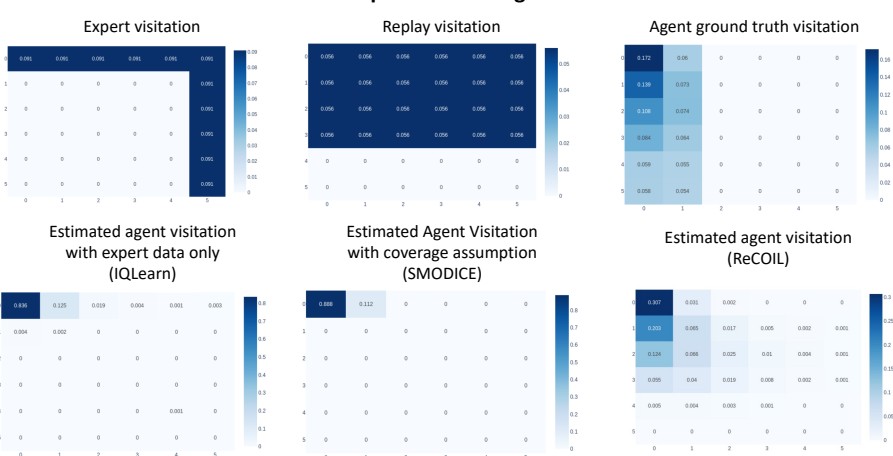

Figure 11: Replay buffer consists of data that visits near the initial state (0,0), a setting commonly observed when training RL agents. We estimate the agent's policy visitation and observe ReCOIL to outperform both methods which rely on expert data only or use the replay data with coverage assumption

### G.4 ReCOIL: Qualitative Comparison with a Baseline

In Figure 12, we investigate qualitatively why other baselines fail where ReCOIL succeeds in high-dimensional tasks. A surprising finding is that the baseline we consider 'SMODICE' almost learns to imitate. It follows nearly the same actions as an expert but makes small mistakes along the way - eg. 'gripping the hammer too loose' or 'picking up the ball at a slightly wrong location'. SMODICE is unable to recover from such mistakes and ends up having low performance. ReCOIL, on the other hand, learns a performant task-solving policy from the same data.

### G.5 Evaluation of $f$-DVL for Online RL

Fig 13 shows that $f$-DVL is competitive to performant off-policy RL methods in the online RL benchmarks.

### G.6 Training Curves for $f$-DVL on MuJoCo Tasks (Offline)

Figure 14 shows the learning curves during training for $f$-DVL. $f$-DVL is able to leverage low-order conjugate $f$-divergences to give offline RL algorithms that more stable compared to XQL. XQL frequently crashes in the antmaze environment.

### G.7 $f$-DVL: Complete Offline RL Results

Table 10 and Table 11 show complete results for benchmarking $f$-DVL on MuJoCo D4RL environments. Here we also show the author-reported results for XQL and the reproduced results (XQL(r)) using the metric of taking the average of the last iterate performance across seeds.

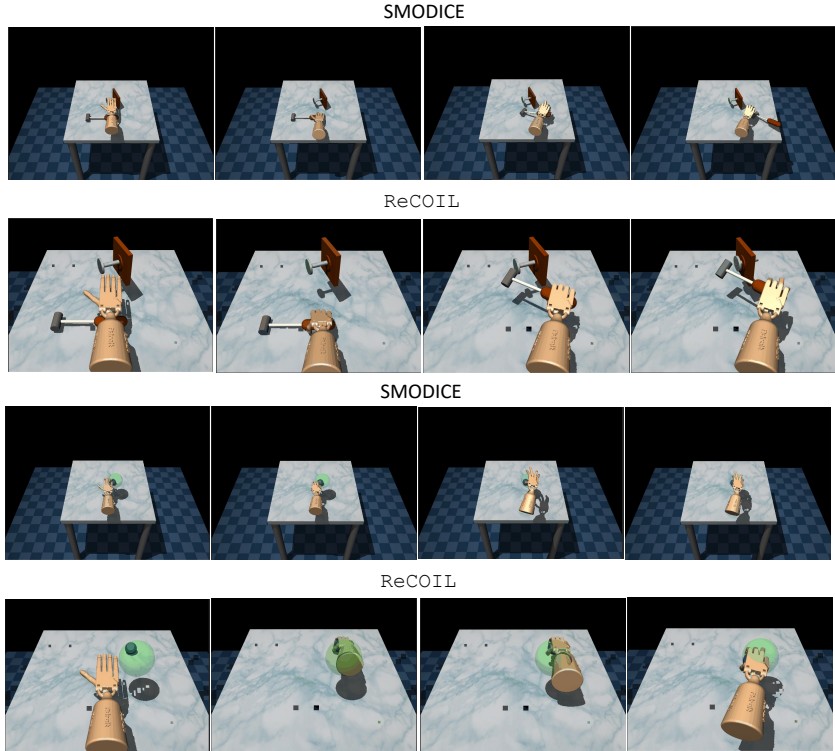

Figure 12: Errors compound in imitation learning and recovery is of crucial importance. Figure demonstrate how SMODICE 'almost' imitates, figures out roughly what actions to take but does not realise once it has made a mistake. In Hammer environment, it grips the hammer too loose causing it to get thrown away and for relocate picks up just beside the ball missing the original task the expert intended to solve.

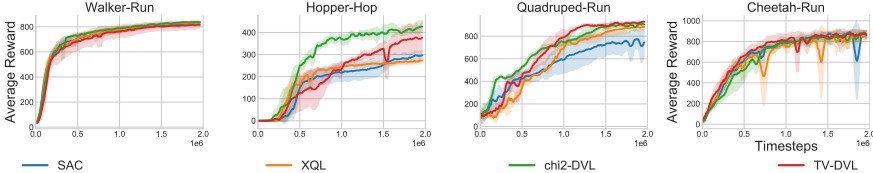

Figure 13: Online RL: $f$-DVL is competitive to SAC and XQL, particularly for Hopper-Hop and Quadruped-Run tasks.

### G.8    SENSITIVITY OF $f$-DVL (OFFLINE) WITH VARYING $\lambda$ ON MUJOCO TASKS

We ablate the temperature parameter, $\lambda$ for offline RL experiments using $f$-DVL in Figure 16 and Figure 15. The temperature $\lambda$ controls the strength of KL penalization between the learned policy and the dataset behavior policy, and a small $\lambda$ is beneficial for datasets with lots of random noisy actions. In contrast, a high $\lambda$ favors more expert-like datasets. We observe that significantly less hyperparameter tuning is required compared to XQL as a single temperature value works well across a broad range of experiments.

### G.9    SENSITIVITY OF $f$-DVL (ONLINE) WITH VARYING $\lambda$ ON MUJOCO TASKS

We ablate the temperature parameter $\lambda$ for online RL experiments using $f$-DVL in Figure 18 (chi-square) and Figure 17 (TV). We observe that significantly less hyperparameter tuning is required compared to XQL as a single temperature value works well across a broad range of experiments.

### G.10    RECOVERING REWARD FUNCTIONS FROM RECOIL

We study the quality of reward functions recovered from ReCOIL using the hopper-medium-expert and Walker2d-medium-expert datasets and the setup described in Section 7.1. For all trajectories in this dataset, we calculate the ground truth return (sum of rewards) and the predicted cumulative

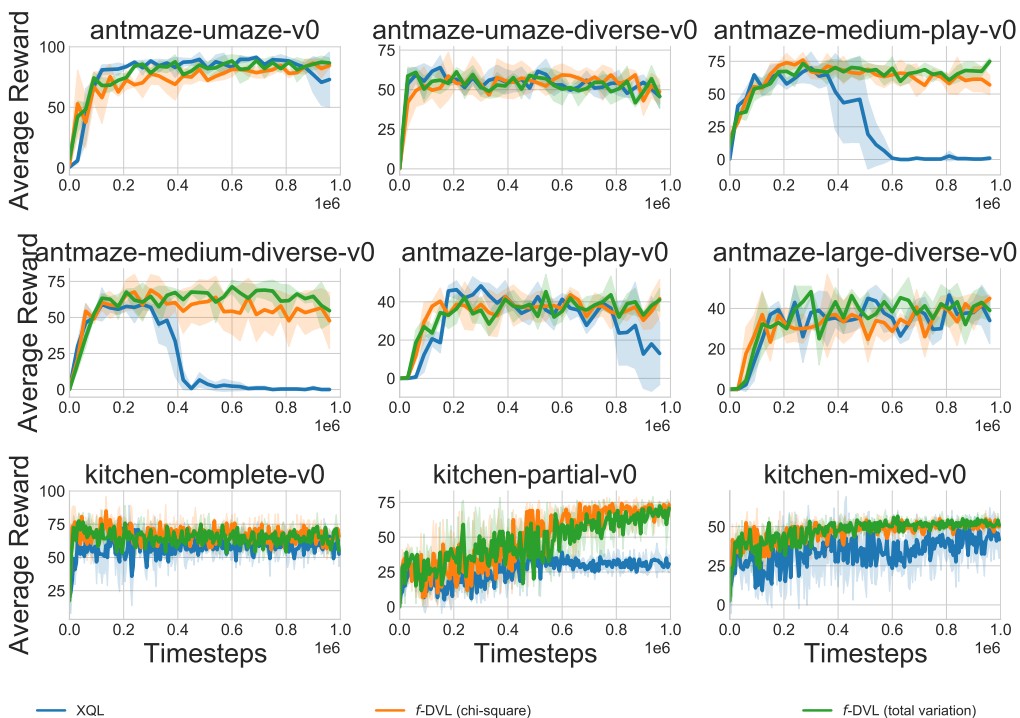

Figure 14: Learning curves for $f$-DVL showing that it is able to leverage low-order conjugate $f$-divergences to give offline RL algorithms that more stable compared to XQL. The results are averaged over 7 seeds

Table 10: Averaged normalized scores on MuJoCo locomotion and Ant Maze tasks. XQL(r) denotes the reproduced results with author's implementation. We do not highlight XQL due to an incorrect evaluation strategy used in the work. Standard deviations for our proposed offline RL methods can be found below.

| | Dataset | BC | 10%BC | DT | TD3+BC | CQL | IQL | XQL | XQL(r) | $f$-DVL $\chi^2$ | $f$-DVL TV |
|---|---|---|---|---|---|---|---|---|---|---|---|
| **Gym** | halfcheetah-medium-v2 | 42.6 | 42.5 | 42.6 | **48.3** | 44.0 | **47.4** | 47.7 | 47.4 | **47.7**±0.23 | 47.5±0.20 |
| | hopper-medium-v2 | 52.9 | 56.9 | 67.6 | 59.3 | 58.5 | 66.3 | 71.1 | **68.5** | 63.0±4.64 | 64.1±3.46 |
| | walker2d-medium-v2 | 75.3 | 75.0 | 74.0 | **83.7** | 72.5 | 78.3 | 81.5 | 81.4 | 80.0±6.75 | 81.5±3.47 |
| | halfcheetah-medium-replay-v2 | 36.6 | 40.6 | 36.6 | **44.6** | 45.5 | 44.2 | 44.8 | 44.1 | 42.9±1.79 | **44.7**±0.44 |
| | hopper-medium-replay-v2 | 18.1 | 75.9 | 82.7 | 60.9 | 95.0 | 94.7 | 97.3 | 95.1 | 90.7±6.13 | **98.0**±4.62 |
| | walker2d-medium-replay-v2 | 26.0 | 62.5 | 66.6 | **81.8** | 77.2 | 73.9 | 75.9 | 58.0 | 52.1±12.15 | 68.7±7.20 |
| | halfcheetah-medium-expert-v2 | 55.2 | **92.9** | 86.8 | 90.7 | 91.6 | 86.7 | 89.8 | 90.8 | 89.3±2.42 | 91.2±2.27 |
| | hopper-medium-expert-v2 | 52.5 | **110.9** | 107.6 | 98.0 | 105.4 | 91.5 | 107.1 | 94.0 | 105.8±5.79 | 93.3±14.04 |
| | walker2d-medium-expert-v2 | 107.5 | 109.0 | 108.1 | **110.1** | 108.8 | **109.6** | 110.1 | **110.1** | **110.1**±0.29 | 109.6±1.46 |
| **AntMaze** | antmaze-umaze-v0 | 54.6 | 62.8 | 59.2 | 78.6 | 74.0 | **87.5** | 87.2 | 47.7 | 83.7±5.90 | **87.7**±3.07 |
| | antmaze-umaze-diverse-v0 | 45.6 | 50.2 | 53.0 | 71.4 | **84.0** | 62.2 | 69.17 | 51.7 | 50.4±2.44 | 48.4±9.95 |
| | antmaze-medium-play-v0 | 0.0 | 5.4 | 0.0 | 10.6 | 61.2 | **71.2** | 73.5 | 31.2 | 56.7±13.82 | **71.0**±5.90 |
| | antmaze-medium-diverse-v0 | 0.0 | 9.8 | 0.0 | 3.0 | 53.7 | **70.0** | 67.8 | 0.0 | 48.2±8.85 | 60.2±7.99 |
| | antmaze-large-play-v0 | 0.0 | 0.0 | 0.0 | 0.2 | 15.8 | 39.6 | 41 | 10.7 | 36.0±5.82 | **41.7**±9.43 |
| | antmaze-large-diverse-v0 | 0.0 | 6.0 | 0.0 | 0.0 | 14.9 | **47.5** | 47.3 | 31.28 | 44.5±7.66 | 39.3±11.84 |
| **Franka** | kitchen-complete-v0 | 65.0 | - | - | - | 43.8 | 62.5 | 72.5 | 56.7 | **67.5**±6.68 | 61.3±7.95 |
| | kitchen-partial-v0 | 38.0 | - | - | - | 49.8 | 46.3 | 73.8 | 48.6 | 58.8±9.60 | **70.0**±1.82 |
| | kitchen-mixed-v0 | 51.5 | - | - | - | 51.0 | 51.0 | 54.6 | 40.4 | **53.75**±5.32 | 52.5±5.15 |

reward using `ReCOIL`. The scatter plot in figure 19 shows the correlation between predicted rewards. We note that `ReCOIL` is an IRL method and suffers from the reward ambiguity problems as rest of the IRL methods— we can only expect a reward function that induces an optimal policy whose visitation is close to an expert and cannot guarantee that we recover the expert's exact reward function. To test the quality of rewards functions output by IRL methods, Pearson correlation is not the accurate metric and metrics like EPIC [Gleave et al., 2020] might be used instead.

Table 11: Evaluation on Adroit tasks from D4RL.XQL-C (r) denotes the reproduced results with author's implementation.

| Dataset | BC | BRAC-p | BEAR | Onestep RL | CQL | IQL | XQL | XQL(r) | $f$-DVL ($\chi^2$) | $f$-DVL (TV) |
|---|---|---|---|---|---|---|---|---|---|---|
| pen-human-v0 | 63.9 | 8.1 | -1.0 | - | 37.5 | 71.5 | 85.5 | 63.5 | 67.1 | 64.1 |
| hammer-human-v0 | 1.2 | 0.3 | 0.3 | - | 4.4 | 1.4 | 2.2 | 1.4 | 2.6 | 1.8 |
| door-human-v0 | 2 | -0.3 | -0.3 | - | 9.9 | 4.3 | 11.5 | 6.63 | 5.7 | 6.77 |
| relocate-human-v0 | 0.1 | -0.3 | -0.3 | - | 0.2 | 0.1 | 0.17 | 0.2 | 0.37 | 0.12 |
| pen-cloned-v0 | 37 | 1.6 | 26.5 | 60.0 | 39.2 | 37.3 | 38.6 | 25.25 | 36.1 | 38.1 |
| hammer-cloned-v0 | 0.6 | 0.3 | 0.3 | 2.1 | 2.1 | 2.1 | 4.3 | 1.58 | 1.64 | 1.65 |
| door-cloned-v0 | 0.0 | -0.1 | -0.1 | 0.4 | 0.4 | 1.6 | 5.9 | 0.69 | 0.45 | 0.87 |
| relocate-cloned-v0 | -0.3 | -0.3 | -0.3 | -0.1 | -0.1 | -0.2 | -0.2 | -0.24 | -0.24 | -0.24 |

Figure 15: Offline RL: Ablating the temperature parameter for $f$-DVL (Total variation). The plot shows the effect of temperature parameters on learning performance.

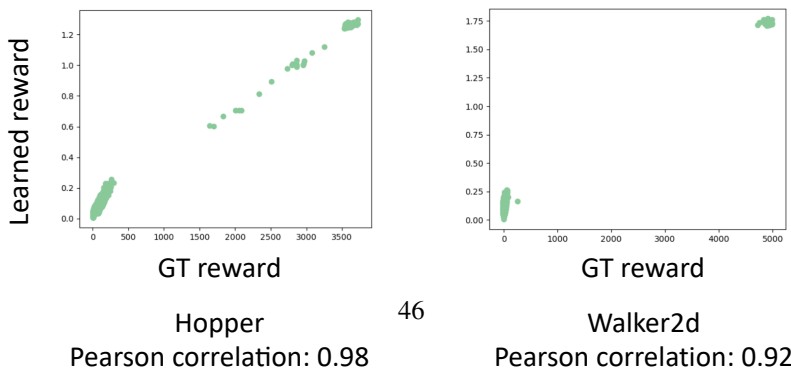

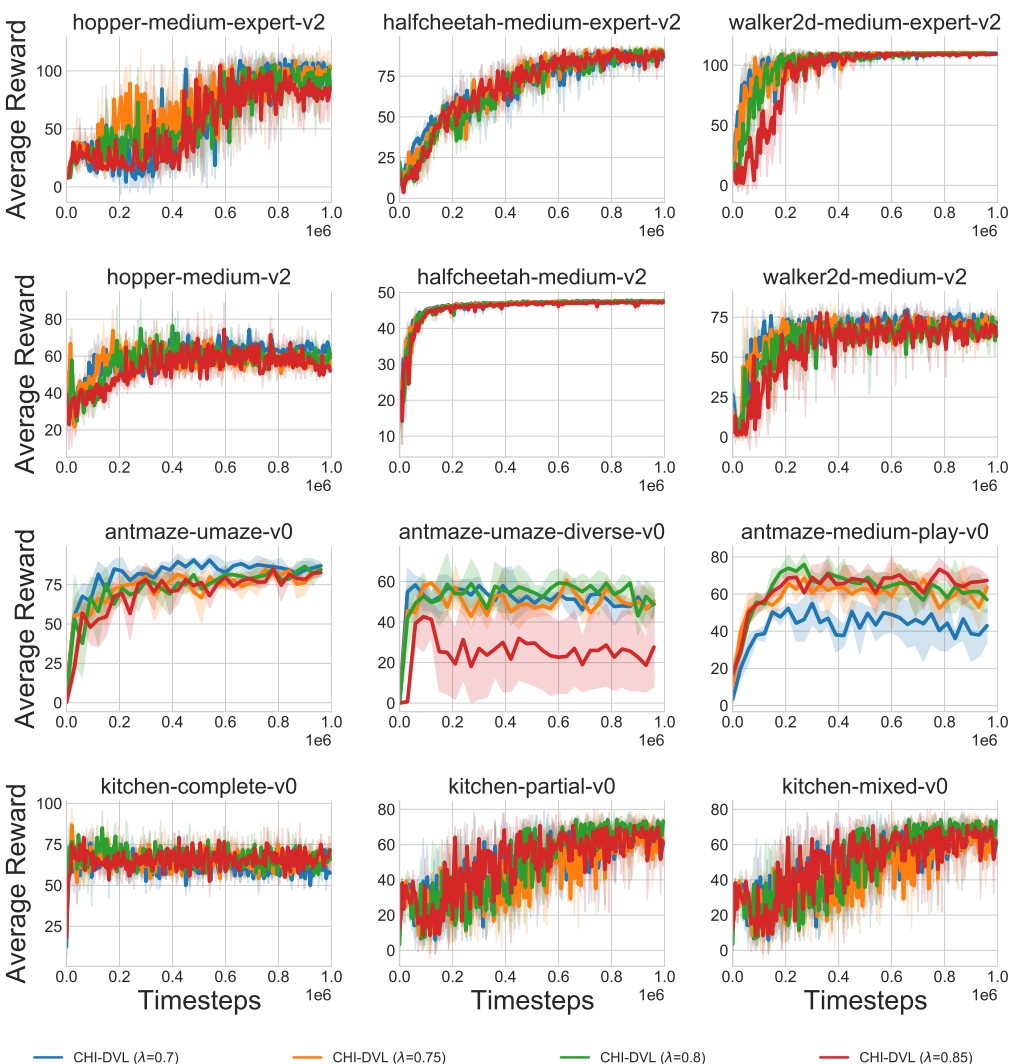

Figure 16: Offline RL: Ablating the temperature parameter for $f$-DVL (Chi-square). The plot shows the effect of temperature parameters on learning performance.

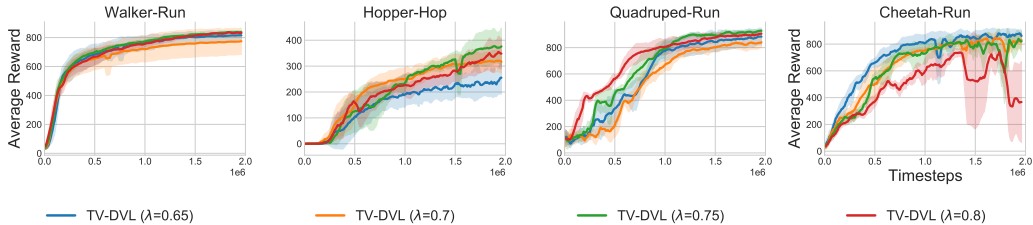

Figure 17: Online RL: Ablating the temperature parameter for $f$-DVL (Total variation). The plot shows the effect of temperature parameters on learning performance.

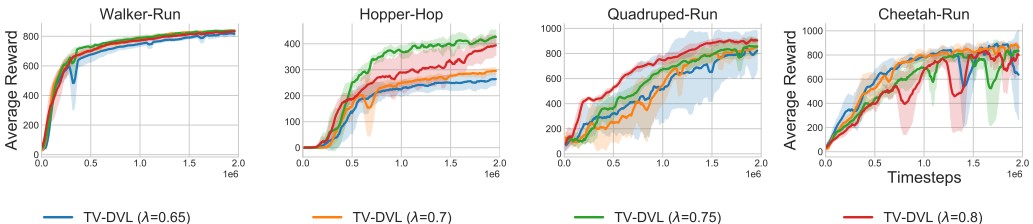

Figure 18: Online RL: Ablating the temperature parameter for $f$-DVL (Chi-square). The plot shows the effect of temperature parameters on learning performance.

