# OpenReview forum: "Dual RL: Unification and New Methods for Reinforcement and Imitation Learning"
_ICLR.cc/2024/Conference — ICLR 2024 spotlight_

### Official Review · Reviewer_aLZ7 · 2023-10-30

**Soundness:** 4 excellent
**Presentation:** 3 good
**Contribution:** 4 excellent
**Rating:** 10
**Confidence:** 4

**Summary:**

This paper unifies a number of recent methods in offline RL and IL through the lens of regularized policy learning and Lagrange duality. Specifically, the paper views the regularized policy learning problem as a convex optimization problem with equality constraints on the state-action visitation distribution. Then, deriving the dual version of two equivalent primal problems, each named dual-Q and dual-V, the authors show that many recent offline IL and RL methods such as IQLearn and XQL, are specific instance of those problems with a particular choice of the f-divergence as the regularizer. Based on this analysis, the authors propose two algorithms for fixing shortcomings of previous algorithms. For offline IL, ReCOIL is presented as a new dual-Q algorithm that can leverage suboptimal demonstration data and that does not require the coverage assumption needed in prior work. For offline RL, f-DVL is proposed as a new dual-V algorithm as an extension of XQL, which results in more stable training by choosing a different f-divergence. The proposed methods are evaluated on the locomotion and manipulation tasks from the D4RL benchmark, and baseline comparisons show superior performance of ReCOIL and f-DVL on various tasks.

**Strengths:**

The major novel contribution of the paper is the theory which shows that various existing offline IL and RL methods can be derived from a Lagrange dual formulation of a single regularized policy learning problem. This itself has a huge consequence. For instance, showing that pessimistic value learning such as CQL is essentially a dual RL problem brings about a new insight into analyzing their properties and deficiencies. The self-contained review as well as derivations of key theoretical results in Appendix C has a core value in the paper, where the math is relatively easy to follow with the basic knowledge of convex optimization. This analysis itself paves a new way to developing many practical algorithms for both offline IL and RL, and has a significant potential impact on future research in this domain.

**Weaknesses:**

On the other hand, I have found a handful of math errors throughout the paper, especially in Appendix C, which critically degrades the quality of such a theory-driven paper. While many of them can be simple typos, some may actually require careful consideration. I will list them below.

* ~~For the state-action visitation distribution $d(s,a)$ to satisfy the Bellman-flow constraint in the form of equation (2), I think that the definition of $d$ has to involve the $(1 - \gamma)$ correction term: $d(s, a) = (1 - \gamma) \pi(a \mid s) \sum_{t=0}^{\infty} \gamma^t P(s_t = s \mid \pi)$.~~

* ~~In equation (9) on p.6 and p.28, the conjugate of the f-divergence should be just $f^*$, not $f^*_p$ because the positivity constraint on the visitation distribution is not necessary in the dual-Q formulation.~~

* ~~In equation (29) you have the term $\gamma \sum_{s'} p(s' \mid s, a) \pi(a' \mid s') Q(s', a')$. I believe that the sum must be taken over both $s'$ and $a'$.~~

 * There is a typo in equation (35). The second term inside $f^*$ should be $\gamma \sum_{s', a'} p(s' \mid s, a) \pi(a' \mid s') Q(s', a')$, not $\gamma \sum_{s'} p(s' \mid s, a) \pi(a \mid s') Q(s', a')$.

~~* There are two typos in equation (38). First, the expectation in the first term needs to be over both $(s, a) \sim d(s, a)$, not $s \sim d(s, a)$. Second, I believe that the last term has to be $\sum_a d(s, a)$, not just $d(s, a)$.~~

* ~~The Lagrangian stationarity condition as stated on p.20 does not seem to be consistent with equation (43). Taking the derivative of the Lagrangian with respect to the primal variables, I suppose that the stationarity is $d^o(s, a) \delta_V(s, a) -d^o(s, a) f'(w^*(s, a)) + \lambda^*(s, a) = 0 ~\forall s \in S, a \in A$. Also, there seems to be an implicit assumption that $d^o(s, a)$ is non-zero everywhere, and the degenerate case of $d^o(s, a) = 0$ is not discussed carefully. It seems that the Lagrangian stationarity is not informative enough to uniquely determine the relationship between $f'(w^*(s, a))$ and $\delta_V(s, a)$ in the degenerate case.~~

* ~~In equation (110), $\alpha$ should be replaced by $\beta$.~~


Regarding the empirical results in Section 7, the quality of presentation has much room for improvement, as described below.
* It is unclear what the left-most figure in Figure 1(a) represents and how it is related to the squared distribution gap plot right next to it.

* In Table 2, it is unclear how the authors determined to highlight some of the entries with blue bold texts as an indication of the most-performant policy. For instance, In the hopper task with the medium few-expert dataset, the RCE seems to have the highest mean but not highlighted in blue.

* ~~Similarity in Table 3, it is unclear what rules were applied to highlight entries in blue. For example, IQL on the antmaze-large-diverse-v0 dataset had a larger return than f-DVL, but both IQL and f-DVL are highlighted.~~

* ~~Moreover, I do not see a huge performance gap between f-DVL and prior methods, and it is hard to judge whether the proposed RL methods empirically outperform existing ones (partially because there is no standard deviation information in the table). In fact, a recent study [1] finds that just reporting the mean and the standard deviation of different RL policies is insufficient, and suggests reporting confidence intervals or performance profiles as more reliable performance measures. The authors should consider leveraging some statistical tools like [1] in comparing policies, especially in situations like Table 3 where many methods have similar average returns.~~

[1] Agarwal, Rishabh, Max Schwarzer, Pablo Samuel Castro, Aaron C. Courville, and Marc Bellemare. "Deep reinforcement learning at the edge of the statistical precipice." Advances in neural information processing systems 34 (2021): 29304-29320.

**Questions:**

* ~~I do not see how one can directly derive equation (14) from equation (5), as a rewriting of dual-V with the temperature parameter $\lambda$. Can you elaborate on the derivation? I believe that the discount factor $\gamma$ is determined by the problem specification and is not a hyperparameter that can be tuned arbitrarily.~~

* ~~In equations (32) to (34), can you elaborate on why the interchangeability principle holds in this specific case? In other words, can you explain why $\max$ and $\mathbb{E}$ can be exchanged?~~

* ~~All the analysis presented in the paper implicitly assumes discrete state and action spaces (which I think is good for readers in the RL community to follow the math). Do the results naturally extend to continuous spaces? I would be surprised if they did not, but I bet we need more sophisticated tools from real-analysis and probability theory to prove the same results.~~

---

> ### Author Response · Authors · 2023-11-16
> **Response to Reviewer aLZ7 (1/2)**
>
> We thank the reviewer for the detailed and thoughtful review. We have uploaded a revision with all the suggested typos corrected. The changes are marked in red. Please find the clarification to the queries below:
>
> 1. **There seems to be an implicit assumption that $d^O$ is non-zero everywhere**
>
> We had earlier discussed the assumption of sufficient coverage of $d^O$ over $\mathcal{S}\times \mathcal{A}$ above Equation 32, but make this more clear in the revised paper. The degeneracy without this assumption is expected as a result of using importance sampling (requires coverage on importance sampled distribution) as well as $f$-divergences which may be ill-defined without enough support. We discuss a potential solution for this in Theorem 2 (see Assumption 1 in theorem 2) where we constrain the search set of policies within the support of offline dataset (similar to the idea of pessimism). In this case, the best we can expect to do is return a policy with highest performance within the dataset.
>
>
> 2. **It is unclear what the left-most figure in Figure 1(a) represents and how it is related to the squared distribution gap plot right next to it.**
>
> We have updated the caption in the revised paper as well as included a detailed explanation of the experiment in Appendix H.3. Please let us know if you need more clarification.
>
>
> 3. **In Table 2, it is unclear how the authors determined to highlight some of the entries with blue bold texts as an indication of the most-performant policy.**
>
> We have updated the highlighting to be more accurate and have highlighted all offline imitation algorithms that lie within the standard deviation of the algorithm with the highest mean performance. The revised paper mentions this in the caption now.
>
> 4. **Similarity in Table 3, it is unclear what rules were applied to highlight entries in blue.**
>
> We have updated the highlighting to mark all offline RL algorithms within one point performance of the top-performing algorithm. Note that in the offline RL experiments, the results extend the performance reported by prior works and the standard deviation is unavailable. Comparing using standard deviation would require rerunning all the baselines, which we believe will take more than the discussion time limit and compute capacity available to us at the moment. We have updated the revised paper to mention this in the caption now.
>
>
> 5. **Moreover, I do not see a huge performance gap between f-DVL and prior methods, and it is hard to judge whether the proposed RL methods empirically outperform existing ones.**
>
> The locomotion benchmark in D4RL for offline RL is simpler with performance to certain extent saturated, compared to the Antmaze or Kitchen environments. A detailed discussion on the properties of these environment and datasets can be found in [2]. We notice that $f$-DVL outperforms in 6/9 tasks in antmaze and kitchen environments which pose a harder challenge of stitching suboptimal trajectories compared to recently proposed IQL[3], XQL[4] and DT[5]. We believe that these improvements over prior work are significant by the standards used in prior offline RL papers.
>
> One of the contributions through $f$-DVL is also to provide another axis of improvement in a stable learning alternative to XQL. Indeed, using statistical metrics like IQM would be more suitable but unfortunately prior works in offline RL(XQL, IQL, etc) have used mean as an estimate, and computing IQL would require rerunning all the baselines which would take more time and more compute budget than we have available.
>
>
> 6. **I do not see how one can directly derive equation (14) from equation (5), as a rewriting of dual-V with the temperature parameter.**
>
>  Note that Equation 14 and Equation 5 are not equivalent - we consider a simpler objective for optimization with identical properties. Indeed the discount factor in $\hat{Q}$ (Eq. 14) remains unchanged and $\lambda$ is used as a substitute for the temperature parameter ($\alpha$) of the second term (multiplying by $\frac{1-\gamma}{1-\lambda}$, results in exactly the same first term as Equation 5 and the weight on second term $\frac{\lambda(1-\gamma)}{1-\lambda}$ acts as a tunable temperature parameter).  A more detailed discussion about this conversion is in Section G.4 in the appendix.
>
> 7. **In equations (32) to (34), can you elaborate on why the interchangeability principle holds in this specific case?**
>
> Consider the unconstrained optimization in Equation 33 w.r.t $d$. The inner optimization objective is an upper semicontinuous concave function. The idea is simply that for every $s,a$ in the expectation one can pointwise maximize $d$, and the maximizer is guaranteed to exist by assumption of concavity and upper semicontinuity. So taking the maximum inside or outside the expectation will lead to same maximizer per $s,a$. We refer the reviewer to Lemma 1 in [1] for a formal proof.

---

> > ### Author Response · Authors · 2023-11-16
> > **Response to Reviewer aLZ7 (2/2)**
> >
> > 8. **The analysis presented in the paper implicitly assumes discrete state and action spaces (which I think is good for readers in the RL community to follow the math). Do the results naturally extend to continuous spaces?**
> >
> > We agree with the reviewer and conjecture that the analysis presented in this work should extend to continuous state-action spaces, but the analysis is out of scope for the paper. Currently, we have validated that the algorithms designed through our analysis extend to continuous spaces by rigourous evaluation on continuous locomotion and manipulation tasks.
> >
> > ------------------------------
> >
> > Please let us know if our clarifications resolve your questions and concerns. We would be happy to expand the discussion.
> >
> >
> > ------------------------------
> >
> > References:
> > [1]: Dai, Bo, et al. "Learning from conditional distributions via dual embeddings." Artificial Intelligence and Statistics. PMLR, 2017.
> > [2]: Fu, Justin, et al. "D4rl: Datasets for deep data-driven reinforcement learning." arXiv preprint arXiv:2004.07219 (2020).
> > [3]: Kostrikov, Ilya, Ashvin Nair, and Sergey Levine. "Offline reinforcement learning with implicit q-learning." arXiv preprint arXiv:2110.06169 (2021).
> > [4]: Garg, Divyansh, et al. "Extreme q-learning: Maxent RL without entropy." arXiv preprint arXiv:2301.02328 (2023).
> > [5]: Chen, Lili, et al. "Decision transformer: Reinforcement learning via sequence modeling." Advances in neural information processing systems 34 (2021): 15084-15097.

---

> > > ### Comment · Reviewer_aLZ7 · 2023-11-20
> > > **Response to Authors**
> > >
> > > Thank you for making revisions and providing detailed comments. Most of my concerns are addressed, with only a few questions remaining.
> > >
> > > 1. Equation (35) remains unchanged in the revised manuscript. Could you please look at my comments in the original review again and confirm whether (35) is correct as is? My concern is that $a'$ is not marginalized out in the expectation.
> > >
> > > 2. You mentioned in the response that the caption in Table 2 was updated to mention the highlighting rule, but I do not see it. Can you double-check?
> > >
> > > 3. Figure 1(a) Left is not totally clear yet. What does the bottom row represent? What does "Truth" mean and which "policy" is plotted here?

---

> > > > ### Author Response · Authors · 2023-11-23
> > > > **Response to Reviewer aLZ7**
> > > >
> > > > We thank the reviewer for helping us improve the quality of the manuscript.
> > > >
> > > > 1. **Equation (35) remains unchanged in the revised manuscript. Could you please look at my comments in the original review again and confirm whether (35) is correct as is?**
> > > >
> > > > We made the changes in the equations above but missed the final equation. Thanks for pointing this out and is corrected now in the revised paper.
> > > >
> > > > 2. **You mentioned in the response that the caption in Table 2 was updated to mention the highlighting rule, but I do not see it. Can you double-check?**
> > > >
> > > > We indeed missed this and have updated it in the paper.
> > > >
> > > > 3. **Figure 1(a) Left is not totally clear yet. What does the bottom row represent? What does "Truth" mean and which "policy" is plotted here?**
> > > >
> > > > The experiment here is: Given a prespecified policy $\pi$ and offline data in the form of suboptimal/replay $d^S$ data and expert data $d^E$, how accurately can we infer the visitation of policy $\pi$ given by $d^\pi$? Since the environment is tabular we can simulate policy $\pi$ and come up with the ground truth visitation of the policy. We use this as the metric against which we compare dual-RL methods using an MSE loss. Note that as indicated by the ground truth policy visitation on states 0,1,2,3,4,5 (left to right), the policy simply takes action to transition to states 1 and 2 with equal probability from state 0 and for all other states takes action to remain in same state.  We had to defer a detailed explanation of this experiment to Appendix H.3 due to space constraints. We have made changes to the caption as well as in Appendix H.3 to increase clarity.
> > > >
> > > > Please let us know if any other outstanding questions or concerns remain and we would be happy to expand the discussion.

---

### Official Review · Reviewer_JKJA · 2023-11-01

**Soundness:** 2 fair
**Presentation:** 3 good
**Contribution:** 3 good
**Rating:** 6
**Confidence:** 4

**Summary:**

The paper proposes a framework called dual RL based on the LP formulation of RL. The author shows dual RL can be used to derive offline RL and offline IL algorithms, providing a unification of existing approaches. Based on this insight, they propose new offline IL and RL algorithms (ReCOIL and f-DVL), which remove existing assumptions. These algorithms are tested in the D4RL benchmarks empirically and achieve competent performance compared with other baselines.

**Strengths:**

The proposed dual RL framework based on the duality of a regularized RL problem is quite clean. The authors provide good background information and detailed derivations, which makes the paper self-contained and easy to follow. The dual RL framework makes some connections of existing works and suggests potential generalization (which the authors adopt) to design new algorithms. The empirical results are promising. Overall, this is an insightful paper.

**Weaknesses:**

There are some technical parts, which I think are not fully correct (see below). If they're indeed wrong, then the contribution of this paper would be compromised. The current writing is also quite dense as well, though the details are provided in the appendix. It would be better if the main paper is more readable. Lastly, in some sense, this paper lacks novelty. The unifying perspective and connection are interesting, but at the end the resulting algorithms can be viewed as minor tweaks from existing ones. Nonetheless, I think this last point on novelty is minor.

**Questions:**

1. Proposition 4: The authors write that CQL can be cast as a dual-Q problem with a right choice of f-divergence (Pearson). However, in the proof of Proposition 4 in Appendix D.1, Eq (70) actually corresponds to the objective of ATAC [Cheng et al. 2022], not CQL. See Eq (1) in [Cheng et al. 2022], which is exactly the same Eq (70) here. To my understanding, the main difference between CQL and ATAC is that ATAC takes a maxmin formulation, whereas CQL doesn't, though they have a similar objective for learning Q. The dual-Q formulation here is a maxmin problem. The authors should update Proposition 4 to make the connection to ATAC, which would be more appropriate and precise.

2. Bug in the proof of Theorem 1. There is a minor bug in the proof of Theorem 1. In the block starting "Imitation from Arbitrary data (dualQ)" after Eq (108), max_d is moved inside min_Q, whereas max_d is outside min_Q in Eq (108). So Eq (109) is not just a summary, and that "we summarize the result of the derivation so far" can be misleading. But this can be easily fixed though, by applying strong duality before moving forward. Another minor suggestion: I think it would be good to write out the proof Eq (112) for completeness.

3. Validity Eq (10). In the proof, the author assume d_0 = d_E. I think that this is a fairly strong assumption, as it implies that the initial state at the test time would be drawn from the same distribution of the expert. This is rarely the case in practice, and in my experience it's not a common assumption in offline or off-policy RL. The authors should highlight this assumption in the main paper, or the authors should present Eq (140) as the main result. A minor bug in the proof is the derivation misses a 1/4 factor in front of the square loss.

4. Bug in Proof of Proposition 3. There're a few issues in the derivation, and one is major.
    a. f', (f')^-1 are not strictly increasing. it's only non-decreasing. e.g. see f(t) = | t-1|.
    b. why is f*(x) = -f(0) for x<0?
    c. why is f(0+)>0?
    d. You write (f')^{-1}(t) >0 when t>0 and 0 otherwise. What does "0 otherwise" mean?
    e. (Major issue). From Eq (79) to Eq (80), it doesn't use "(f')^{-1}(t) >0 when t>0" (though the paper writes so). It uses the other way (f')^{-1}(t) >0  --> t>0 and likewise (f')^{-1}(t) <0  --> t<0. But they are not true. See e.g. f(t) = | t-1| again. This error step prevents me from validating the remaining part.

5. The experiment design of offline IL can be a bit confusing. It says the agent is given 1 expert demonstration. But the suboptimal transition dataset actually contains 200 expert demonstrations, though I understand they're not marked as expert demonstrations to the algorithm. I think a more convincing experiment is to completely remove the expert demonstrations in the suboptimal data. Table 2 writes 1000 expert transitions. Is that the same as one expert trajectory?

6. I don't know how to interpret Fig 1. Can you explain more?

Minor:

1. This sentence is confusing to read. "Unfortunately, one of the most successful classes of offline RL methods .. has evaded connections to regularized policy optimization. Proposition 2 shows, perhaps surprisingly, that XQL can be cast as a dual of regularized policy learning, concretely as a dual-V problem."

2. I don't understand the connection between the sentence "To prevent extrapolation error in the offline setting, we rely on an implicit maximizer [Garg et al., 2023] that estimates the maximum over the Q-function conservatively with in-distribution samples." and Eq (12)

Reference:
Cheng, Ching-An, et al. "Adversarially trained actor critic for offline reinforcement learning." International Conference on Machine Learning. PMLR, 2022.

---

> ### Author Response · Authors · 2023-11-16
> **Response to Reviewer JKJA (1/2)**
>
> We thank the reviewer for taking the time to provide a thoughtful and thorough review of the paper. We uploaded the revised paper with the changes highlighted in red. Please find the clarification to the queries below:
>
> 1. **Proposition 4: The authors write that CQL can be cast as a dual-Q problem with a right choice of f-divergence (Pearson). However, in the proof of Proposition 4 in Appendix D.1, Eq (70) actually corresponds to the objective of ATAC [Cheng et al. 2022], not CQL.**
>
> We appreciate reviewer's suggestion of expanding the family of algorithms unified by dual-RL by including ATAC[1]. We have updated Proposition 4 and it's proof to point out the connection to ATAC.
>
> We believe Conservative Q-learning (CQL) and ATAC share some similarities in their objectives, which makes it hard to discard CQL from this family.  The reviewer mentions that CQL does not formulate their objective as a min-max game. However, we believe this is not completely correct. Directing the reviewer's attention to Equation 3 in CQL, a min-max game is proposed between an action-proposal distribution $\mu$ and the Q-function. The authors point out that $\mu$ can be taken as the last policy iterate (also used in their implementation for high-dimensional action spaces) making it particularly close to the objective coming out of dual-RL. A straightforward difference between CQL and ATAC is the reliance on Gradient-Descent Ascent vs the Stackelberg game strategy of updating the two-player game. We have included both ATAC and CQL in Proposition 4 in the revised paper.
>
>
> 2. **Eq (109) is not just a summary, and that "we summarize the result of the derivation so far" can be misleading. But this can be easily fixed though, by applying strong duality before moving forward.**
>
> Thanks for pointing this out. We have made the correction.
>
>
>
> 3. **Validity Eq (10). In the proof, the author assume $d_0 = d_E$. I think that this is a fairly strong assumption, as it implies that the initial state at the test time would be drawn from the same distribution of the expert.**
>
> We think there might be a misunderstanding here. We do not assume $d_0=d^E$ but rather assume $d_0=d^S$. We clarify that Eq.10 is a simplified objective and the full derivation under $\chi^2$ can be found in Appendix E.3. In the full derivation we downweigh the $Q$-function on states obtained from mixture distribution of offline data ($1-\gamma$) and expert data ($\gamma$), and the actions obtained from current policy. In practice, the simplified objective contains the offline data term only as our offline data is a superset of expert data and $\gamma$ is close to 1.
>
> The offline distribution $d^S$ is a wider distribution encompassing the starting state $d_0$ distribution. This strategy allows us to learn a near-optimal policy from a wider initial state distribution and should not be detrimental to function representation with sufficient capacity. A similar strategy is common in (eg. [2], off-policy RL [3],[4] ).
>
> We have corrected Eq.10 with the missing constant of 1/4 and also updated the sentence before Eq.10 to say that we are presenting a simplified objective here.
>
> 4. **Bug in Proof of Proposition 3.**
>
>
> We apologize for the phrasing of this section in the appendix leading to confusion. We have rewritten the paragraph to increase clarity. To summarize answers to your concerns briefly:
>
> - We have corrected that $f'$, $(f')^{-1}$ are not strictly increasing (only non-decreasing), but note that we do not need them to be strictly increasing for the following proof.
> -  A common issue when dealing with $f$-divergences is the limited domain of $\mathbb{R}^+$ on which $f$ divergences are defined and subsequently limits the domain and range of $f'$ or ${f'}^{-1}$. For us to be able to create an optimization objective defined on $\mathbb{R}$, we utilize a surrogate extension of ${f'}^{-1}$ on $\mathbb{R}$. This was initially expanded upon in the practical algorithm section but we have now made this change in Section 6 as well.
> - $(f')^{-1}$ for TV divergence (which the reviewer points out) is not well defined and $f^*$ for RKL and Sq. Hellinger divergences are discontinuous on $\mathbb{R}^+$. These divergences require special treatment and have expanded this discussion in the proof of Proposition 3. This treatment was earlier discussed in our proposed practical objective for $f$-DVL in Section 6.

---

> > ### Author Response · Authors · 2023-11-16
> > **Response to Reviewer JKJA (2/2)**
> >
> > 5. **The experiment design of offline IL can be a bit confusing. It says the agent is given 1 expert demonstration. But the suboptimal transition dataset actually contains 200 expert demonstrations, though I understand they're not marked as expert demonstrations to the algorithm. I think a more convincing experiment is to completely remove the expert demonstrations in the suboptimal data.**
> >
> > The suboptimal dataset we consider either comes from random or medium (for locomotion)/ cloned-human (for manipulation), it might be unreasonable to expect any imitation learning method to achieve meaningful performance on these tasks with 1-expert trajectory as even RL with known dense reward function has struggled to obtain near-expert behavior from these datasets (eg. Table 1 in [3]). Second, the experimental design for imitation learning used here is not a contribution of our work rather is an evaluation benchmark established by prior works [5,6]. We note that in our work, we extended this benchmark to include manipulation tasks for which we collected our own data following a similar protocol outlined in [5,6].
> >
> > 6. **Table 2 writes 1000 expert transitions. Is that the same as one expert trajectory?**
> >
> > Yes, thanks for pointing this out. We have corrected it to say one expert trajectory.
> >
> > 7. **I don't know how to interpret Fig 1. Can you explain more?**
> >
> > We have updated the caption in the revised paper as well as included a detailed explanation of the experiment in Appendix H.3. Figure 1 aims to test the ability of ReCOIL objective to learn distribution ratios. For a fixed policy, the inner optimization objective of Equation 9 captures the distribution ratio between mixture distributions of (current policy visitation+offline data visitation) and (expert visitation+offline data visitation). In Figure 1, we consider a 1-D MDP with simple dynamics. Given an expert visitation, replay visitation and a policy, we aim to test how accurately can we predict the given policy's visitation. Figure 1(a) also provides the ground truth visitation of the policy, and we plot MSE error with ground truth visitation in the right plot.
> >
> >
> > 8. **This sentence is confusing to read. "Unfortunately, one of the most successful classes of offline RL methods..."**
> >
> > We have rephrased this sentence. Thanks for pointing this out.
> >
> >
> > 9. **I don't understand the connection between the sentence "To prevent extrapolation error in the offline setting, we rely on an implicit maximizer [Garg et al., 2023] that estimates the maximum over the Q-function conservatively with in-distribution samples." and Eq (12)**
> >
> > In the offline setting, optimizing Equation 10 results in value function overestimation. This phenomenon arises because policy maximization can choose actions that are not present in the dataset, whose values are incorrectly overestimated. Now, these overestimated values gets bootstrapped as a result of Bellman backup in the last term of Equation 10 resulting in value function blowup. To solve this issue, we subsitute the unconstrained policy maximization step with an in-sample maximizer -- only samples that are seen in datset are used to conservatively (under)estimate the maximum.
> >
> > -----------------
> >
> > Please let us know if our clarifications resolves your questions and concerns. We would be happy to expand the discussion.
> >
> >
> > -----------------
> >
> > References:
> > [1] Cheng, Ching-An, et al. "Adversarially trained actor critic for offline reinforcement learning." International Conference on Machine Learning. PMLR, 2022.
> > [2] Kostrikov, Ilya, Ofir Nachum, and Jonathan Tompson. "Imitation learning via off-policy distribution matching." arXiv preprint arXiv:1912.05032 (2019).
> > [3] Kumar, Aviral, et al. "Conservative q-learning for offline reinforcement learning." Advances in Neural Information Processing Systems 33 (2020): 1179-1191.
> > [4] Garg, Divyansh, et al. "Extreme q-learning: Maxent RL without entropy." arXiv preprint arXiv:2301.02328 (2023).
> > [5] Ma, Yecheng, et al. "Versatile offline imitation from observations and examples via regularized state-occupancy matching." International Conference on Machine Learning. PMLR, 2022.
> > [6] Kim, Geon-Hyeong, et al. "Demodice: Offline imitation learning with supplementary imperfect demonstrations." International Conference on Learning Representations. 2021.

---

> > > ### Author Response · Authors · 2023-11-23
> > > **Follow up on rebuttal**
> > >
> > > We thank the reviewer for taking the time to provide a thoughtful review and are motivated that they found the work insightful, clean, and detailed.
> > >
> > > We believe we have responded to the questions and concerns in full, but if something is missing please let us know and we would be happy to add it. If we have addressed your concerns we would appreciate it if the reviewer could reassess our work in light of these clarifications.

---

### Official Review · Reviewer_7FxJ · 2023-11-10

**Soundness:** 3 good
**Presentation:** 3 good
**Contribution:** 3 good
**Rating:** 8
**Confidence:** 4

**Summary:**

The paper presents a new approach in offline IL, adopting the f-MAX objective function instead of the commonly utilized SMODICE objective. Unlike f-MAX, which employs GAIL for optimization, this work leverages the DICE technique to derive a dual form and subsequently applies an IQL-style algorithm for resolution. Additionally, the framework exhibits versatility, with potential applications in both the online and offline RL settings.

**Strengths:**

1. The paper offers a thorough and insightful analysis of recent developments within the dual-RL paradigm.
2. The proposed framework demonstrates adaptability across IL, offline RL, and online RL settings.
3. By eliminating the discriminator in offline IL, the paper potentially enhances stability and reliability in this domain.

**Weaknesses:**

1. The paper's structure appears ill-suited for a conference format, dedicating 3.5 pages to prior knowledge while relegating significant analyses and experimental results (pertaining to online and offline RL) to the Appendix.
2. There are instances of overclaims and problematic expressions that require rectification.

**Questions:**

Your work is commendable, but improvements in presentation and accuracy are needed. My specific questions and suggestions are as follows:

1. XQL's instability and heavy reliance on hyperparameters are noticeable, with subpar performance in both online and offline RL under unsuitable conditions. As an alternative example, it might be more suitable to reference a more general and effective implicit policy improvement algorithm, like IQL[1]. I suggest you use a more general and effective implicit policy improvement algorithm, IQL, as an example.

2. In the last paragraph of Section 4.1, it is not suitable to say, "This insight also allows us to cast IL method OPOLO ....." as the objective function here originates from SMODICE. Referencing "this objective function" or similar expressions would be more appropriate.

3. In the first paragraph of Section 4.2, it is strange to use "Unfortunately" just because XQL is not the policy regularization style.

4. Figure 1 (a) is quite hard to follow. Please refine the caption to explain what the floats and the colors mean.

5. The first paragraph of Section 7.1 lacks supportive evidence for this claim. Incorporating illustrative figures and evaluations would substantiate this statement.

6. On page 36, typo in "We base the implementation of SAC off pytorch_sac ..."

7. On page 37, I disagree with your claim that "Our experiments with the popular off-policy method SAC [Haarnoja et al., 2018] reveal its brittleness to off-policy data". SAC is not brittle to off-policy data. Filling the replay buffer arbitrarily with expert data at the beginning is not fair for an ADP-style algorithm. Please rewrite this paragraph.

8. On page 40, the statement "f-DVL leads to noticeable improvements even in the online RL benchmarks" appears overstated:

   * The baselines are too few.

   * The performance improvement is not noticeable in 75% tasks you conducted experiments in.

   If you would like to claim noticeable,  please include more baselines (PPO[2], TD3[3], AlgeaDICE[4], and some other more recent algorithms, e.g., RRS[5], BAC[6], REDQ[7]) and more benchmark tasks. And please broaden the discussion of related works in the online RL setting beyond SAC.

9. Some crucial experimental outcomes, currently in Appendix H, should be integrated into the main paper for better accessibility and coherence, especially those results referenced and analyzed within the main paper.

10. Better not to claim "New" in the title directly. Just saying "A unified framework" is ok.

By addressing these points, your work could gain greater clarity and impact. And then, I would like to raise the score.

[1] Kostrikov I, Nair A, Levine S. Offline Reinforcement Learning with Implicit Q-Learning[C]//International Conference on Learning Representations. 2021.

[2] John Schulman, Filip Wolski, Prafulla Dhariwal, Alec Radford, and Oleg Klimov. Proximal policy optimization algorithms. arXiv preprint arXiv:1707.06347, 2017.

[3] S Fujimoto, H van Hoof, and D Meger. Addressing function approximation error in actor-critic methods. Proceedings of Machine Learning Research, 80:1587–1596, 2018.

[4] Nachum O, Dai B, Kostrikov I, et al. Algaedice: Policy gradient from arbitrary experience[J]. arXiv preprint arXiv:1912.02074, 2019.

[5] Hao Sun, Lei Han, Rui Yang, Xiaoteng Ma, Jian Guo, and Bolei Zhou. Optimistic curiosity exploration and conservative exploitation with linear reward shaping. In Advances in Neural Information Processing Systems, 2022.

[6] Ji T, Luo Y, Sun F, et al. Seizing Serendipity: Exploiting the Value of Past Success in Off-Policy Actor-Critic[J]. arXiv preprint arXiv:2306.02865, 2023.

[7] Chen X, Wang C, Zhou Z, et al. Randomized Ensembled Double Q-Learning: Learning Fast Without a Model[C]//International Conference on Learning Representations. 2021.

---

> ### Author Response · Authors · 2023-11-16
> **Response to Reviewer 7FxJ**
>
> We thank the reviewer for taking the time to provide a thoughtful and thorough review of the paper.
>
> We have made the changes for all the suggested typos in the revision. The changes are highlighted in red. We appreciate the reviewer's attention for pointing these out. We provide clarification to reviewers queries below:
>
> 1. **Figure 1 (a) is quite hard to follow. Please refine the caption to explain what the floats and the colors mean.**
>
> We have elaborated on the experiment in Figure 1 (a) in the caption of the revised paper as well as added a detailed explanation of the experiment setup in Appendix H.3.
>
> 2. **The first paragraph of Section 7.1 lacks supportive evidence for this claim. Incorporating illustrative figures and evaluations would substantiate this statement.**
>
> We want to request clarification on this question. It seems like Section 7.1 is the experimental setup section and perhaps the reviewer meant some other section.
>
> 3.  **Filling the replay buffer arbitrarily with expert data at the beginning is not fair for an ADP-style algorithm. Please rewrite this paragraph.**
>
> In this section of the appendix, our intention was to compare popular ADP-based off-policy methods vs dual-RL/DICE style off-policy methods. Any off-policy algorithm, by definition, should be able to make use of any kind of data obtained from the environment -- expert data is a special case. We have reworded this section to make our claims more accurate since as the reviewer points out considering one special case is not enough to say the the algorithm is brittle.
>
>
> 4. **On page 40, the statement "f-DVL leads to noticeable improvements even in the online RL benchmarks" appears overstated. And please broaden the discussion of related works in the online RL setting beyond SAC.**
>
> We have corrected our claim to say that we are competitive with SAC and XQL in online RL experimeents. In this work, our focus was limited to offline setting and we leave a more in-depth analysis of online RL algorithms for future work. We have added the suggested recent online RL methods in the related work section.
>
>
> 5. **Some crucial experimental outcomes, currently in Appendix H, should be integrated into the main paper for better accessibility and coherence, especially those results referenced and analyzed within the main paper.**
>
>  We appreciate the reviewer's suggestion. We acknowledge that our appendix contains a number of interesting experiments. In this work, we were constrained by the page limit to strike a balance between facilitating understanding, demonstrating our contributions concisely, and providing empirical evidence to support the theoretical contributions. Currently, we take $\approx 0.6$ page for related works and $\approx 1.4$ page for preliminary. We believe introduction should not be considered a part of prior work and a similar format can be found in a number of influential ICLR works [1,2,3]. We would be happy to receive suggestions on what the reviewer thinks might be traded off in favor of more experiments.
>
>
> 6. **Better not to claim "New" in the title directly. Just saying "A unified framework" is ok.**
>
> We clarify that the use of the word 'New' was to signify that we leveraged the existing framework of duality in RL to propose new and improved algorithms for offline RL and IL and not to signify the novelty of the duality framework.
>
> -------------------
>
> We appreciate the reviewer's suggestions and are happy to incorporate other suggestions into our submission. Please let us know if there are other issues or concerns we can address. We would be happy to expand this discussion.
>
> -------------------
>
> References:
> [1]: Eyuboglu, Sabri, et al. "Domino: Discovering systematic errors with cross-modal embeddings." arXiv preprint arXiv:2203.14960 (2022).
> [2]: Flennerhag, Sebastian, et al. "Bootstrapped meta-learning." arXiv preprint arXiv:2109.04504 (2021).
> [3]: Ren, Tongzheng, et al. "Latent variable representation for reinforcement learning." arXiv preprint arXiv:2212.08765 (2022).

---

> > ### Comment · Reviewer_7FxJ · 2023-11-16
> > **Discussion regarding to "The first paragraph of Section 7.1"**
> >
> > I wanted to clarify a point I previously mentioned regarding the term "LOW COVERAGE OF EXPERT”, as I realize my initial expression might not be clear. This term could potentially be misleading. The support set of 30 expert demonstrations may not have a clear difference compared to 100 expert demonstrations, but I still think this is a much more difficult setting. Thus I suggest you provide some support for this claim or change the expression here.

---

> ### Comment · Reviewer_7FxJ · 2023-11-16
> **Discussion about putting the interesting experiments in your appendix into main text.**
>
> I wanted to discuss the possibility of moving some of the interesting experiments from your appendix into the main text of your paper.
>
> Please don't worry too much about this comment. I think your work is really good, and this is just a suggestion from a writing perspective. Feel free to disregard it if it doesn't suit your needs, and don't feel any pressure about it.
>
> In terms of shortening some parts of the paper, I feel that the sections on basic MDP definitions, value functions, and Bellman operators are a bit lengthy. Perhaps you could consider condensing them slightly.

---

> > ### Author Response · Authors · 2023-11-17
> > **Response to Reviewer 7FxJ**
> >
> > We thank the reviewer for their kind words and motivating responses. We have uploaded a revision with the changes suggested by the reviewer.
> >
> > 1. We agree the use of the term 'low coverage' could be misleading for an experiment with fewer expert samples. We have corrected to claim that we use fewer samples to simulate a more difficult imitation learning setting.
> >
> > 2. We have condensed the preliminary section and included an additional experiment (Fig 3) comparing the learning stability of dual-RL and SAC-based approaches when additional expert data is added to their replay buffer at the beginning of training.
> >
> >  Please let us know if there are other issues or concerns we can address. We hope the reviewer can reassess our work in light of the writing improvements.

---

> > > ### Comment · Reviewer_7FxJ · 2023-11-18
> > > **Thank the authors for their rebuttal!**
> > >
> > > Thank you for your rebuttal! I would like to raise my score.

---

### Official Review · Reviewer_NyfP · 2023-11-20

**Soundness:** 3 good
**Presentation:** 3 good
**Contribution:** 3 good
**Rating:** 6
**Confidence:** 4

**Summary:**

This paper proposes a unified dual RL framework that connects several recent offline RL and IL methods. New algorithms called ReCOIL (for IL) and f-DVL (for RL) are presented that aim to address limitations of prior approaches. Experiments across locomotion, manipulation and navigation tasks generally demonstrate improved or comparable performance to baselines.

**Strengths:**

- The theoretical dual RL formulation provides a common lens to view recent offline RL/IL methods. This is a useful conceptual contribution.
- The methods are evaluated extensively on a diverse set of tasks using standardized benchmarks. Implementation details are clearly described.
- For offline IL, ReCOIL relaxes assumptions like coverage and avoids instability of density ratio estimation. The results showcase strong performance on imitation tasks.
- The modifications in f-DVL seem to improve training stability compared to XQL, a prior state-of-the-art in offline RL.

**Weaknesses:**

- While f-DVL outperforms XQL, its gains over other offline RL methods like IQL are marginal. The gains are not as significant as claimed over the full spectrum of baselines.
- The presentation of empirical results could be improved by using standardized metrics, showing confidence intervals, and increasing clarity around performance highlights.
- There are open questions around design choices, estimation procedures, and other technical details that warrant clarification.

**Questions:**

Overall, I am moderately positive about this submission. The theoretical framework is clean and impactful. The empirical results are reasonably strong but could use tighter presentation and analysis.

I have the following specific questions:

- In Eq. 10, is the reward assumed to be zero in the Bellman consistency term?
- Can you expand on how ReCOIL estimates the policy visitation and reward for the results in Sec 7.1? Also clarify if Fig 1 and Fig 10 use OPOLO or SMODICE for comparison.
- What do the distributions D and d^O refer to in Eq. 12 and 13? Are they both d_{mix}^{E,S}?

---

> ### Author Response · Authors · 2023-11-20
> **Response to Reviewer NyfP (1/2)**
>
> We thank the reviewer for the thoughtful review. The modifications are marked in red in the paper revision. We address your concerns and questions below.
>
>
> 1. **While f-DVL outperforms XQL, its gains over other offline RL methods like IQL are marginal. The gains are not as significant as claimed over the full spectrum of baselines.**
>
> The locomotion benchmark in D4RL for offline RL is simpler with performance to a certain extent saturated, compared to the Antmaze or Kitchen environments. Our proposed approach is competitive in these domains. A detailed discussion of the properties of these environments and datasets can be found in [1]. We notice that $f$-DVL outperforms in 6/9 tasks in antmaze and kitchen environments which pose a harder challenge of stitching suboptimal trajectories compared to recently proposed IQL[2], XQL[3], and DT[4].  We believe that these improvements over prior work are meaningful by the standards used in prior offline RL papers.
> One of the contributions through $f$-DVL is also to provide another axis of improvement in a stable learning alternative to XQL. We would like to bring attention to the fact that XQL (ICLR 23) is a more recent method than IQL (ICLR 22) and our proposed method drives direct improvement over XQL.
>
> 2. **The presentation of empirical results could be improved by using standardized metrics, showing confidence intervals, and increasing clarity around performance highlights.**
>
> We appreciate the reviewer's suggestions and note that the following changes have been made in our revised paper: a) In the results for imitation learning (ReCOIL) we highlight all methods that lie within the standard deviation of the best-performing method. b) In the offline RL (f-DVL) results we highlight methods that are within 1 performance point of the best-performing method. Additionally, we have also added standard deviations for our method in Table 10. Indeed, using statistical metrics would be more suitable but unfortunately prior works in offline RL like XQL have used mean as an estimate. We extend prior results in our work and computing metrics like Inter-Quartile mean (IQM) or std-dev would require rerunning all the baselines which would take more time and more compute budget than we have available.
>
> Questions:
>
> 1. **In Eq. 10, is the reward assumed to be zero in the Bellman consistency term?**
>
> To clarify for Eq. 10, we are in the setting of imitation learning where we are not provided with task rewards but rather only expert demonstrations. Eq. 10 is a direct consequence of Theorem 1 which provides a principled objective for imitation learning signifying that bellman backup with zero rewards in conjunction with a contrastive loss on Q is sufficient and pricipled for imitation learning.

---

> ### Author Response · Authors · 2023-11-20
> **Response to Reviewer NyfP (2/2)**
>
> 2. **Can you expand on how ReCOIL estimates the policy visitation and reward for the results in Sec 7.1?**
>
>
> We have added a detailed explanation for estimating policy visitation in Appendix H.3 in the revised paper and also edited the caption of Figure 1 to give more clarity. In brief, any dual-RL objective written in the $Q$-form allows us to extract density ratios by considering the inner optimization of Equation 9.
>
>  For reward estimation, the paragraph below Theorem 1 explains how rewards can be inferred from learned Q-function and policy. In brief, given $Q^*$ and $\pi^*$, the rewards can be extracted as $r(s,a)= Q^*(s,a)-\mathcal{T}^{\pi^*}_0(Q^*(s,a))$
>
>  3. **Also clarify if Fig 1 and Fig 10 use OPOLO or SMODICE for comparison.**
>
>  We appreciate the reviewer pointing out this typo, we correctly mentioned in the figures that we used SMODICE but incorrectly in the text that OPOLO was used for comparison. We have made this correction in the text. SMODICE can be understood as an offline counterpart to OPOLO and in our work we focus on the offline setting. A more detailed discussion of this experiment and baselines can be found in Appendix H.3.
>
> 4. **What do the distributions D and $d^O$ refer to in Eq. 12 and 13? Are they both $d_{mix}^{E,S}$?**
>
> We appreciate the reviewer bringing attention to this typo. They are both indeed $d_{mix}^{E,S}$ and we have fixed this in the revised paper.
>
> -----------------
>
>
> Please let us know if our clarifications resolve your questions and concerns. We would be happy to expand the discussion on any remaining specific concerns about design choices, and other technical details . **Since the review is posted near the end of the discussion period we would greatly appreciate if we are made aware of any remaining concerns and issues soon.**
>
>
>
>
> -----------------
> -----------------
>
>
>
> References:
> [1]: Fu, Justin, et al. "D4rl: Datasets for deep data-driven reinforcement learning." arXiv preprint arXiv:2004.07219 (2020).
> [2]: Kostrikov, Ilya, Ashvin Nair, and Sergey Levine. "Offline reinforcement learning with implicit q-learning." arXiv preprint arXiv:2110.06169 (2021).
> [3]: Garg, Divyansh, et al. "Extreme q-learning: Maxent RL without entropy." arXiv preprint arXiv:2301.02328 (2023).
> [4]: Chen, Lili, et al. "Decision transformer: Reinforcement learning via sequence modeling." Advances in neural information processing systems 34 (2021): 15084-15097.

---

> > ### Author Response · Authors · 2023-11-23
> > **Follow up on Rebuttal**
> >
> > We thank the reviewer for taking the time to provide a thoughtful review and are motivated that they found the work to be clean, and impactful with extensive experiments.
> >
> > We believe we have responded to the questions and concerns in full, but if something is missing please let us know and we would be happy to add it. If we have addressed your concerns we would appreciate it if the reviewer could reassess our work in light of these clarifications.

---

### Meta-Review · Area_Chair_f7ZB · 2023-12-05

**Metareview:**

The paper uses Lagrangian duality to propose a framework that subsumes several existing methods for offline RL and offline IL.  They use the framework to derive new methods for both scenarios, which are demonstrated to work well empirically.

Strengths:
- new, interesting framework
- strong empirical results
- training more stable than XQL
- method does not depend on having a discriminator
- the paper offers a new perspective on CQL and ATTAC, which are influential algorithms

Weaknesses:
- no performance bounds for finite dataset sizes. In other words, there is nothing in the theory that how much data to have. In addition, in the batch IL setting, it would be useful to know how much adding more exploration data helps relative to adding expert data.
- evaluation does not use rliable (https://github.com/google-research/rliable) or similar framework to summarize the results.

Note to authors: please fix the (minor) math errors pointed out by reviewer JKJA.

**Justification For Why Not Higher Score:**

The paper is good, but not stellar. See weakness section in the meta-review. My understanding is that the paper characterises well what happens when dataset sizes go to infinity. However, the more interesting setting for batch RL / batch IL is where the dataset is small enough for there to be a significant difference between the available trajectories and the underlying MDP.

**Justification For Why Not Lower Score:**

See strengths section in the meta-review. I think this a really solid contribution which subsumes many influential prior methods.

---

### Decision · Program_Chairs · 2024-01-16

Accept (spotlight)